# Physics of Language Models: Part 1, Learning Hierarchical Language Structures[*]

**Zeyuan Allen-Zhu**                  *zeyuanallenzhu@meta.com*
*FAIR at Meta*

**Yuanzhi Li**                    *Yuanzhi.Li@mbzuai.ac.ae*
*Mohamed bin Zayed University of AI*

**Reviewed on OpenReview:** *https://openreview.net/forum?id=mPQKyzkA1K*

## Abstract

Transformer-based language models are effective but complex, and understanding their inner workings and reasoning mechanisms remains a significant challenge. Previous research has primarily explored how these models handle simple tasks such as name copying or selection; we extend this line of work by investigating how they perform recursive language structure reasoning defined by context-free grammars (CFGs). We introduce a family of synthetic CFGs that produce hierarchical rules, capable of generating long (e.g., hundreds of tokens), locally ambiguous sentences that require dynamic programming to parse. Despite this complexity, we demonstrate that autoregressive language models such as GPT can accurately learn and reason over these CFG-defined hierarchical languages and generate valid continuations. Analyzing model internals in this controlled setting, we reveal that hidden states linearly encode CFG parse structure, and that attention patterns align closely with the information flow of dynamic-programming parsing algorithms.

This paper also presents corollary findings, including: why absolute positional embeddings are inferior to relative and rotary embeddings; why uniform attention alone is surprisingly effective (motivating our follow-up work on Canon layers (Allen-Zhu, 2025a;b)); why encoder-only models (e.g., BERT, DeBERTa) struggle with *deep* structural reasoning on CFGs compared to autoregressive models (e.g., GPT); and why injecting structural or syntactic noise into pretraining data markedly improves robustness to corrupted language prompts.

## 1 Introduction

Transformer-based language models, like GPT (OpenAI, 2023), are powerful but mysterious; many studies attempt to uncover the inner workings of transformers. Perhaps the simplest observation is that attention heads can pair closing brackets with open ones, see the concurrent work and the references therein (Zhang et al., 2023). Others also demonstrate that transformer can store key-value knowledge pairs by storing value in the hidden embedding of keys (see Allen-Zhu and Li (2024) and the references therein).

The seminal work from Anthropic (Elhage et al., 2021; Olsson et al., 2022) focuses on *induction heads*, which are logic operations *on the input level* (such as [A][B]...[A] implies the next token should be [B]). This can be used to interpret how language models perform sequence copying, translation, and some easy forms of pattern matching. They hypothesized that induction heads may exist to "match and copy more abstract and sophisticated linguistic features, rather than precise tokens", yet they acknowledge that they "don't have a

---

[*]The first six papers in the *Physics of Language Models* series were presented as a two-hour tutorial at ICML 2024 in Austria (`youtu.be/yBL7J0kgldU`). A 100-min deep dive into Part 1 is available at `youtu.be/kf_eGgVtOcs`. **Future updates and code releases are available** via `ssrn.com/abstract=5250639` and the project page `physics.allen-zhu.com`. This TMLR-accepted version corresponds to V5 of the paper: V1 (May 24, 2023); V2 added Appendix G (additional data families); V3 polished the writing and revised the title; V4 added Appendix H (more uniform-attention results); and V5 added pseudocode and further clarified the connection to dynamic programming.

| root \|->20 21 | 19\|->18 16 18 | 16\|->15 15 | 13\|->11 12 | 10\|->8 9 9 | 7\|->2 2 1 | *an example sentence* |
|---|---|---|---|---|---|---|
| root \|->20 19 21 | 19\|->17 18 | 16\|->13 15 13 | 13\|->12 11 12 | 10\|->9 7 9 | 7\|->3 2 2 | |
| root \|->21 19 19 | 19\|->18 18 | 16\|->14 13 | 13\|->10 12 11 | 10\|->9 7 9 | 7\|->3 1 2 | 33221312331211312321132231231211121321132231 1311 |
| root \|->20 20 | 20\|->16 16 | 16\|->14 14 | 14\|->10 12 | 11\|->8 8 | 7\|->3 2 | 32233312312111213113311213212133333123221213 1232 |
| | 20\|->16 17 | 17\|->15 14 13 | 14\|->12 10 12 | 11\|->9 7 | 8\|->3 1 1 | 22111121332213113113113111111323123313313331 1331 |
| | 20\|->17 16 18 | 17\|->14 15 | 14\|->12 11 | 11\|->9 7 7 | 8\|->1 2 | 33333223121131112122111121123331233112111331 3333 |
| | 21\|->18 17 | 17\|->15 14 | 14\|->10 12 12 | 12\|->7 9 7 | 8\|->3 3 1 | 33112333313111133331211321131212111333332121 11121 |
| | 21\|->17 16 | 18\|->14 15 13 | 15\|->10 11 11 | 12\|->9 8 | 9\|->1 2 1 | 21322322332213322111322113232331311121322322 3221 |
| | 21\|->16 17 18 | 18\|->15 13 13 | 15\|->11 11 10 | 12\|->8 8 9 | 9\|->3 3 | 21113333112132222133322112121331213313322122 13221 |
| | 21\|->16 18 | 18\|->13 15 | 15\|->10 10 | | 9\|->1 1 | 211213331232233312 |
| | | | 15\|->12 12 11 | | | |

Figure 1: An example CFG used in our experiments. It generates long (e.g., *length 354* in this example) and ambiguous strings. Determining if a string $x$ belongs to the CFG language $x \in L(\mathcal{G})$ typically requires dynamic programming, even when the CFG rules are known.

strong framework for mechanistically understanding" this.

The *interpretability in the wild* paper (Wang et al., 2022) explored many different types of attention heads, including "copy head", "name mover head", "inhibition head", etc. Most notably, they explained how GPT2 predicts the next token "Mary" given prefix "When Mary and John went to the store, John gave a drink to [...]" This requires some logical reasoning by selecting (not naively copying) what is the right name. While this result is very inspiring, there exists very simple rule-based algorithm to achieve the same.[1]

In practice, transformers perform much more complex operations and reasoning, yet, achieving a mechanistic understanding of their internal workings remains a significant challenge. To gain such interpretability on how a transformer performs a certain task, it is often beneficial to have a *well-defined algorithm* for that task; the model's internal representations and computations can then be *compared against* this algorithmic benchmark. However, many "impressive skills" of state-of-the-art language models are for tasks lacking such clear algorithmic solutions. Motivated by this, we ask: *Is there a setting for us to understand* how *language models perform* hard *tasks, involving deep logics / reasoning / computation chains?*

To isolate and rigorously study how models tackle tasks demanding deep reasoning over hierarchical structures, we employ a *controlled* setting using synthetic Context-Free Grammars (CFGs). CFGs, which include terminal (T) and nonterminal (NT) symbols, a root symbol, and production rules, inherently *hierarchically* produce highly-structured expressions. Crucially for our study, parsing such CFG-defined languages—a form of *structured reasoning*—often necessitates textbook-level, yet quite difficult, dynamic programming (DP)—a class of algorithms relevant to complex problem-solving. This CFG/DP paradigm provides a framework to probe for DP-like computational mechanisms when language models tackle these structured tasks.[2] Generally,

- We wish to capture how models reason over *long-range* dependencies via CFG. The simplest example is bracket matching, in ...Y(...)[[...]{...}]{...}X, the next symbol X could depend on Y that was hundreds of tokens before. Another example is coding, where goto N can only be used if N is a valid line number that could be hundreds of lines ago.

- We wish to capture how models reason through *local ambiguity*. A coding grammar (like python) can be parsed using a greedy algorithm without ambiguity, so does bracket matching — once locally seen ...()... we know the two parentheses must be paired together. We *focus on hard CFGs* that require global planning via *dynamic programming* to parse.

Most popular choices of CFGs do not satisfy the two above properties. Notably, the English CFG (e.g., derived from Penn TreeBank) has an average length of 28 tokens (too short), and is not very locally ambiguous (e.g., RB JJ or JJ PP imply their parent must be ADJP). As we show in Appendix G, such CFGs can even be learned using tiny GPT2 models with $\sim$ 100k parameters. Thus, *CFG grammars based on human languages may be* too easy *for our interpretability purpose.*

**For this reason, we design synthetic CFGs.** We give one example in Figure 1 and discuss a family of 7 CFGs with varying difficulties in Section 2 (we have 15 more in the appendix).[3] We *pre-train* GPT-

---

[1] Yet, they also said "to the best of our knowledge, (this is) the most detailed attempt at reverse-engineering a natural end-to-end behavior in a transformer-based language model."

[2] Not to say in the theory community, CFGs are also used to model some rich, recursive structure in languages, including some logics, grammars, formats, expressions, patterns, etc.

[3] A benefit of using synthetic data is to control the difficulty of the data, so that we can observe how transformers learn to solve tasks at different difficulty levels.

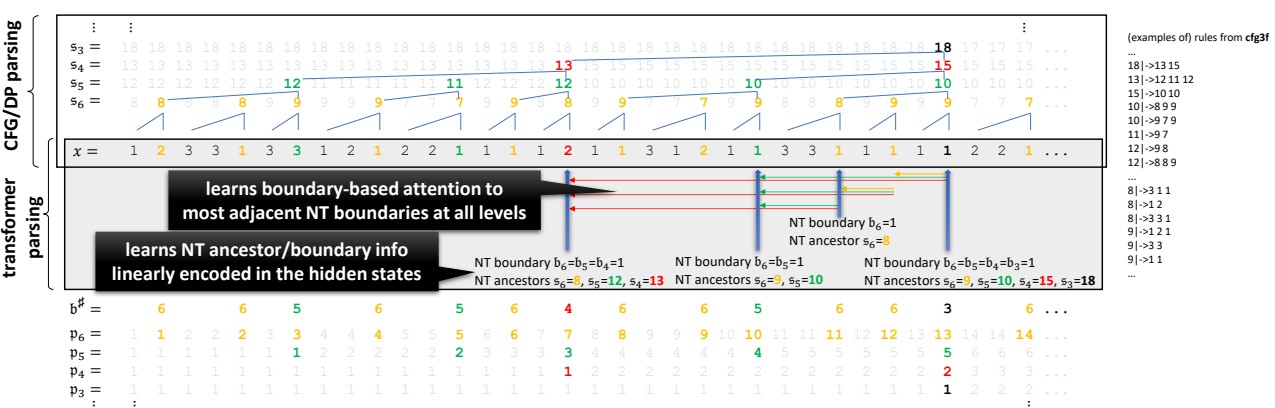

Figure 2: An example string $x$ from $\mathcal{G} = \texttt{cfg3f}$. Though formally defined in Section 2.1, bold symbols in color represent *NT boundaries* which mark the ending positions of the parsed CFG subtrees at various levels $\ell$: we denote by $\mathfrak{b}_\ell(i) = 1$ if $x_i$ is at the NT boundary for level $\ell$. The *NT ancestor* $\mathfrak{s}_\ell(i)$ represents the tree node's label on level $\ell$ for symbol $x_i$. The NT ancestor index $\mathfrak{p}_\ell(i)$ represents that $x_i$ is on the "$\mathfrak{p}_\ell(i)$-th" subtree for level $\ell$ counting from the left.

2 (Radford et al., 2019), denoted by GPT, on a language modeling task using a corpus of strings sampled from such CFGs. We test the model's accuracy and diversity by feeding it prefixes from the CFG (or no prefix, just the starting token) and observing if it can generate completions.

It is perhaps evident from Figure 1 that *even if* the CFG tree is given, *deciding* if a string satisfies it may require scratch paper and half an hour for a person, not to mention learning the CFG from scratch. However, we demonstrate that GPT can learn these CFGs, and using rotary or relative attention is crucial, especially for complex CFGs (**Results 1-3**). More crucially, we examine attention patterns and hidden states to understand the reasoning mechanisms GPT employs to achieve this. Specifically,

- **Results 4-5.** Develop a *multi-head linear probing method* to verify that the model's hidden states linearly encode NT information almost perfectly, a significant finding as pre-training does not expose the CFG structure. (In contrast, encoder models like BERT do not.)

- **Results 6-9.** Introduce methods to visualize and quantify attention patterns, demonstrating that GPT learns position-based and boundary-based attentions, contributing to understanding how it performs hierarchical structure reasoning of CFG regularity and periodicity.

- **Corollary.** GPT models perform structure reasoning on CFGs by mimicking information flow characteristic of dynamic programming. Boundary-based attention allows a token to attend to its closest NT symbols in CFG tree, even when separated by hundreds of tokens. This resembles DP, in which parsing on a sequence $1...i$ needs to be "concatenated" with another sequence $i + 1...j$ to form a solution to a larger problem on $1...j$. See Figure 2+10 for illustrations.

We also explore *implicit CFGs* (Post and Bergsma, 2013), where each T symbol is a bag of tokens, and data is generated by randomly selecting tokens from these bags. Implicit CFGs capture additional structures, such as word categories. We demonstrate that GPT models learn implicit CFGs by encoding the T symbol information (i.e., token bags) directly into their token embedding layers (**Result 10**).

We further examine *model robustness* (Moradi and Samwald, 2021; Tu et al., 2020) using CFGs, assessing the model's ability to auto-correct errors and generate valid CFGs from a corrupted prefix (e.g., randomly flipping 15% of the symbols in the prefix). This capability is crucial as it reflects the model's ability to process real-world data, including those containing grammatical errors. We find that:

- **Result 11.** GPT models, trained on grammatically correct data, exhibit low robustness. However, introducing just a 10% perturbation to the training data significantly improves the model's robustness. This suggests the benefit of using lower-quality data during pre-training.

- **Result 12-13.** When trained with perturbed data, GPT models develop a "mode switch" for toggling between making or not making grammar mistakes. This behavior is observable in real-life completion models like Llama or GPT-3 (davinci003).

While previous works explored synthetic grammars and interpretability (e.g., (Hewitt and Manning, 2019;

Deletang et al., 2023)), our contribution lies in isolating and quantifying dynamic programming-like computation in generative models via CFGs that require global parsing decisions — a regime where local heuristics fail.

## 2 Our Synthetic Context-Free Grammars

A probabilistic context-free grammar (CFG) is a formal system defining a string distribution using production rules. It comprises four components: terminal symbols ($\mathbf{T}$), nonterminal symbols ($\mathbf{NT}$), a root symbol ($root \in \mathbf{NT}$), and production rules ($\mathcal{R}$). We represent a CFG as $\mathcal{G} = (\mathbf{T}, \mathbf{NT}, \mathcal{R})$, with $L(\mathcal{G})$ denoting the string distribution generated by $\mathcal{G}$.

### 2.1 Definition and Notations

We focus on $L$-level CFGs where each level $\ell \in [L]$ corresponds to a set of symbols $\mathbf{NT}_\ell$ with $\mathbf{NT}_\ell \subseteq \mathbf{NT}$ for $\ell < L$, $\mathbf{NT}_L = \mathbf{T}$, and $\mathbf{NT}_1 = \{root\}$. Symbols at different levels are disjoint: $\mathbf{NT}_i \cap \mathbf{NT}_j = \varnothing$ for $i \neq j$. We consider rules of length 2 or 3, denoted as $\mathcal{R} = (\mathcal{R}_1, \ldots, \mathcal{R}_{L-1})$, where each $\mathcal{R}_\ell$ consists of rules in the form:

$$r = (a \mapsto b, c, d) \quad \text{or} \quad r = (a \mapsto b, c) \quad \text{for} \quad a \in \mathbf{NT}_\ell \quad \text{and} \quad b, c, d \in \mathbf{NT}_{\ell+1}$$

Given a non-terminal symbol $a \in \mathbf{NT}$ and any rule $r = (a \mapsto \star)$, we say $a \in r$. For each $a \in \mathbf{NT}$, its associated set of rules is $\mathcal{R}(a) \stackrel{\text{def}}{=} \{r \mid r \in \mathcal{R}_\ell \wedge a \in r\}$, its *degree* is $|\mathcal{R}(a)|$, and the CFG's *size* is $(|\mathbf{NT}_1|, |\mathbf{NT}_2|, \ldots, |\mathbf{NT}_L|)$.

**Generating from CFG.** To generate samples $x$ from $L(\mathcal{G})$, follow these steps:

1. Start with the *root* symbol $\mathbf{NT}_1$.
2. For each layer $\ell < L$, keep a sequence of symbols $s_\ell = (s_{\ell,1}, \cdots, s_{\ell,m_\ell})$.
3. For the next layer, randomly sample a rule $r \in \mathcal{R}(s_{\ell,i})$ for each $s_{\ell,i}$ with uniform probability.[4] Replace $s_{\ell,i}$ with $b, c, d$ if $r = (s_{\ell,i} \mapsto b, c, d)$, or with $b, c$ if $r = (s_{\ell,i} \mapsto b, c)$. Let the resulting sequence be $s_\ell = (s_{\ell+1,1}, \cdots, s_{\ell+1,m_{\ell+1}})$.
4. During generation, when a rule $s_{\ell,i} \mapsto s_{\ell+1,j}, s_{\ell+1,j+1}$ is applied, define the parent $\mathsf{par}_{\ell+1}(j) = \mathsf{par}_{\ell+1}(j+1) \stackrel{\text{def}}{=} i$ (and similarly if the rule of $s_{\ell,i}$ is of length 3).
5. Define *NT ancestor indices* $\mathfrak{p} = (\mathfrak{p}_1(i), \ldots, \mathfrak{p}_L(i))_{i \in [m_L]}$ and *NT ancestor symbols* $\mathfrak{s} = (\mathfrak{s}_1(i), \ldots, \mathfrak{s}_L(i))_{i \in [m_L]}$ as shown in Figure 2:

$$\mathfrak{p}_L(j) \stackrel{\text{def}}{=} j \ , \quad \mathfrak{p}_\ell(j) \stackrel{\text{def}}{=} \mathsf{par}_{\ell+1}(\mathfrak{p}_{\ell+1}(j)) \quad \text{and} \quad \mathfrak{s}_\ell(j) \stackrel{\text{def}}{=} s_{\ell, \mathfrak{p}_\ell(j)}$$

The final string is $x = s_L = (s_{L,1}, \cdots, s_{L,m_L})$ with $x_i = s_{L,i}$ and length $\mathbf{len}(x) = m_L$. We use $(x, \mathfrak{p}, \mathfrak{s}) \sim L(\mathcal{G})$ to represent $x$ with its associated NT ancestor indices and symbols, sampled according to the generation process. We write $x \sim L(\mathcal{G})$ when $\mathfrak{p}$ and $\mathfrak{s}$ are evident from the context.

**Definition 2.1.** *A symbol $x_i$ in a sample $(x, \mathfrak{p}, \mathfrak{s}) \sim L(\mathcal{G})$ is the **NT boundary / NT end** on level $\ell \in [L-1]$ if $\mathfrak{p}_\ell(i) \neq \mathfrak{p}_\ell(i+1)$ or $i = \mathbf{len}(x)$. We denote $\mathfrak{b}_\ell(i) \stackrel{\text{def}}{=} \mathbb{1}_{x_i \text{ is the NT boundary on level } \ell}$ as the **NT-end boundary** indicator function. The* deepest NT-end *of $i$ is — see also Figure 2 —*

$$\mathfrak{b}^\sharp(i) = \min_{\ell \in \{2,3,\ldots,L-1\}} \{\mathfrak{b}_\ell(i) = 1\} \quad \text{or} \perp \text{ if set is empty }.$$

**The cfg3 synthetic CFG family.** We focus on seven synthetic CFGs of depth $L = 7$ detailed in Section A.1. The hard datasets $\mathsf{cfg3b}, \mathsf{cfg3i}, \mathsf{cfg3h}, \mathsf{cfg3g}, \mathsf{cfg3f}$ have sizes $(1, 3, 3, 3, 3, 3, 3)$ and increasing difficulties $\mathsf{cfg3b} < \mathsf{cfg3i} < \mathsf{cfg3h} < \mathsf{cfg3g} < \mathsf{cfg3f}$. The easy datasets $\mathsf{cfg3e1}$ and $\mathsf{cfg3e2}$ have sizes $(1, 3, 9, 27, 81, 27, 9)$ and $(1, 3, 9, 27, 27, 9, 4)$ respectively. The sequences generated by these CFGs are up to $3^6 = 729$ in length. Typically, the learning difficulty of CFGs *inversely scales* with the number of NT/T symbols, assuming other factors remain constant, because having more NT/T symbols makes the language less ambiguous and more easily parsed using greedy (see Figure 4, indeed $\mathsf{cfg3e1}$ and $\mathsf{cfg3e2}$ are much easier

---

[4]For simplicity, we consider the uniform case, eliminating rules with extremely low probability. Such rules complicate the learning of the CFG and the investigation of a transformer's inner workings (e.g., require larger networks and longer training time). Our results do extend to non-uniform cases when the distributions are not heavily unbalanced.

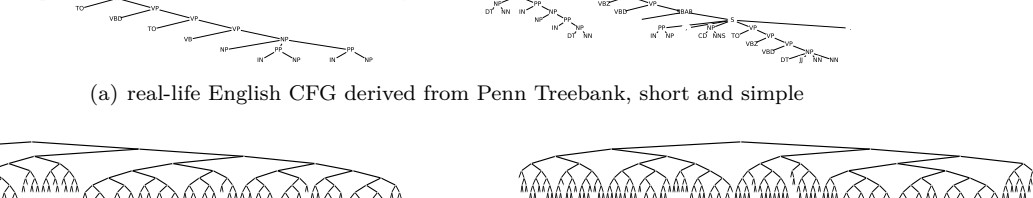

(a) real-life English CFG derived from Penn Treebank, short and simple

(b) a family of max-depth 11 CFGs where rules have length 1 or 2 that GPT can learn, see cfg0 in Appendix G

Figure 3: CFG visual comparisons: *left* is a medium-length sample, and *right* is a 80%-percentile-length sample

to learn and we discuss more in Appendix G). We thus primarily focus on cfg3b, cfg3i, cfg3h, cfg3g, cfg3f.

## 2.2 Why Such CFGs

We use CFG as a proxy to study rich, recursive *structure reasoning* in languages—from logics and grammars to formats and patterns. Those structures are diverse yet strict (e.g., in a CFG describing chapter numbers, Chapter 3.1 can be only followed by Chapter 3.1.1, Chapter 4 or Chapter 3.2, not others). The CFGs we consider are non-trivial, with over $2^{270} > 10^{80}$ strings in cfg3f among a total of over $3^{300} > 10^{140}$ possible strings of length 300 or more (see our entropy estimation later in Figure 4). The probability of a random string belonging to this language is nearly zero, and a random completion of a valid prefix is unlikely to satisfy the CFG. In particular, Figure 31 in the appendix shows that cfg3f cannot be learned by transformers (much) smaller than GPT2-small. In contrast, the English CFG (e.g., derived from Penn TreeBank) can be learned to good accuracy using tiny GPT2 models with $\sim 100k$ parameters — *it is too easy* for our interpretability purpose.

To obtain clean interpretability result and facilitate clearer analysis of learned representations across processing levels, we selected a CFG family with a 'canonical representation' (e.g., layered CFG). This layered structure, while simplified, allows for direct probing of per-level NT symbol encodings and attention patterns at distinct hierarchical depths, aiding our interpretability goals. This *controlled* design allows us to demonstrate a strong correlation between the CFG representation and the hidden states in the learned transformer. We also create additional CFG families to examine 'not-so-canonical' CFG trees, with results deferred to Appendix G (an example in Figure 3). *We do not claim* our results encompass all CFGs; our chosen CFGs are already challenging for a transformer to learn and can lead to clean hierarchical interpretability results.

## 3 Results 1-3: Transformer Can Learn Such CFGs

Before we analyze *how* transformers perform structure reasoning on such CFGs, we have to first verify that they *at least can* learn such CFGs. In this section, we generate a large corpus $\{x^{(i)}\}_{i \in [N]}$ from a synthetic CFG language $L(\mathcal{G})$ in Section 2.1, and pretrain a (decoder-only) transformer model $F$ on this corpus, treating each terminal symbol as a separate token, using an auto-regressive task (see Appendix A.3 for details). We then evaluate how well the model learns such $L(\mathcal{G})$.

**Models.** We denote the GPT2 small architecture (12-layer, 12-head, 768-dimensions) as GPT (Radford et al., 2019) and implemented its two modern variants. We denote GPT with relative positional attention (He et al., 2020) as $\text{GPT}_{\text{rel}}$, and GPT with rotary attention (Su et al., 2021; Black et al., 2022) as $\text{GPT}_{\text{rot}}$. For purposes in later sections, we introduce two weaker variants. $\text{GPT}_{\text{pos}}$ replaces the attention matrix with a matrix based solely on tokens' relative positions, while $\text{GPT}_{\text{uni}}$ uses a constant, uniform average of past tokens from various window lengths as the attention matrix. Detailed explanations of these variants are in Section A.2.

We quickly summarize our findings and then elaborate them in details.

**generation acc (%)**

| | GPT | | GPT_rel | | GPT_rot | | GPT_pos | | GPT_uni | |
|---|---|---|---|---|---|---|---|---|---|---|
| cfg3b | 99.8 | 99.8 | 99.8 | 99.9 | 99.8 | 99.9 | 99.9 | 99.9 | 99.9 | 100.0 |
| cfg3i | 99.5 | 99.5 | 99.8 | 99.8 | 99.4 | 99.5 | 99.8 | 99.8 | 99.6 | 99.7 |
| cfg3h | 96.8 | 96.9 | 99.7 | 99.6 | 99.6 | 99.5 | 99.0 | 99.0 | 98.9 | 98.8 |
| cfg3g | 84.1 | 83.8 | 99.1 | 99.2 | 98.6 | 98.4 | 97.0 | 96.9 | 96.7 | 96.9 |
| cfg3f | 57.1 | 57.3 | 98.8 | 98.8 | 97.6 | 97.7 | 93.9 | 93.8 | 92.8 | 92.9 |
| cfg3e1 | 98.1 | 98.9 | 98.4 | 99.0 | 98.2 | 98.9 | 98.3 | 98.9 | 98.6 | 99.0 |
| cfg3e2 | 99.3 | 99.5 | 99.6 | 99.7 | 99.6 | 99.7 | 99.5 | 99.7 | 99.4 | 99.6 |
| | cut0 | cut50 | cut0 | cut50 | cut0 | cut50 | cut0 | cut50 | cut0 | cut50 |

**entropy (bits)**

| | truth | GPT | GPT_rel | GPT_rot | GPT_pos | GPT_uni |
|---|---|---|---|---|---|---|
| cfg3b | 169 | 169 | 169 | 169 | 169 | 169 |
| cfg3i | 185 | 190 | 189 | 189 | 190 | 189 |
| cfg3h | 204 | 203 | 203 | 203 | 202 | 203 |
| cfg3g | 268 | 272 | 267 | 268 | 266 | 267 |
| cfg3f | 268 | 275 | 270 | 272 | 269 | 269 |
| cfg3e1 | 216 | 214 | 213 | 213 | 214 | 213 |
| cfg3e2 | 256 | 252 | 255 | 251 | 253 | 252 |

**KL divergence**

| | GPT | GPT_rel | GPT_rot | GPT_pos | GPT_uni |
|---|---|---|---|---|---|
| cfg3b | 0.00008 | 0.00011 | 0.00009 | 0.00009 | 0.00004 |
| cfg3i | 0.00024 | 0.00014 | 0.00028 | 0.00015 | 0.00021 |
| cfg3h | 0.00078 | 0.00023 | 0.00023 | 0.00027 | 0.00036 |
| cfg3g | 0.00450 | 0.00034 | 0.00047 | 0.00058 | 0.00069 |
| cfg3f | 0.00455 | 0.00043 | 0.00060 | 0.00093 | 0.00112 |
| cfg3e1 | 0.00019 | 0.00014 | 0.00016 | 0.00013 | 0.00011 |
| cfg3e2 | 0.00031 | 0.00025 | 0.00025 | 0.00011 | 0.00011 |

Figure 4: Generation accuracy (left), entropy (middle), KL-divergence (right) across multiple CFG datasets. **Observations:** Less ambiguous CFGs (`cfg3e1`, `cfg3e2`, as they have more NT/T symbols) are easier to learn. Transformers using relative positional embedding ($\text{GPT}_\text{rel}$ or $\text{GPT}_\text{rot}$) are better for learning harder CFGs. The vanilla `GPT` is worse than even $\text{GPT}_\text{uni}$, which is `GPT` with fixed, uniform attentions.

---

**Result 1-3** (Figure 4). *The GPT models (except the original absolute embedding variant) can effectively learn our synthetic CFGs. Given any prefix, they can generate completion strings*

- *that can perfectly adhere to the CFG rules most of the time,* (accuracy)
- *that are sufficiently diverse in the CFG language, and* (diversity)
- *that closely follow the probabilistic distribution of the CFG language.* (probability)

Moreover, one had better use rotary or relative attentions; the original `GPT` (with absolute positional embedding) performs even worse than $\text{GPT}_\text{uni}$ (with uniform attention).

---

**Result 1: Completion accuracy.** We evaluate $F$ by letting it generate completions for prefixes $x_{:c} = (x_1, x_2, \cdots, x_c)$ from strings $x$ freshly sampled from $L(\mathcal{G})$. The *generation accuracy* is measured as $\mathbf{Pr}_{x \sim L(G) \, + \, \text{randomness of } F}[(x_{:c}, F(x_{:c})) \in L(\mathcal{G})]$. We use multinomial sampling without beam search for generation.[5]

Figure 4 (left) shows the generation accuracies for cuts $c = 0$ and $c = 50$. The $c = 0$ result tests the model's ability to generate a sentence in the CFG, while $c = 50$ tests that to complete a sentence.[6] The results show that the pretrained GPT models can often generate strings that perfectly adhere to the CFG rules for the `cfg3` data family.

**Result 2: Generation diversity.** Could it be possible that the pretrained GPT models only memorized a small subset of strings from the CFG? We evaluate this by measuring the diversity of its generated strings. High diversity suggests a better understanding of the CFG rules.

We consider two methods to estimate diversity. One is to estimate the distribution's entropy, which provides a rough estimate of (the $\log_2$ of) the support size, see the middle of Figure 4. The other is to use birthday paradox to theoretically lower bound the support size (Arora and Zhang, 2017). This allows us to make precise claims, such as in the `cfg3f` dataset, there are at least $4 \times 10^8$ distinct sentential forms derivable from a symbol on levels 1 to 5 or levels 2 to 6; not to say from the root to level 7. Details are in Appendix B. Our general conclusion is that the pre-trained model *does not rely on simply memorizing* a small set of patterns to achieve high completion accuracy.

**Result 3: Distribution comparison.** To fully learn a CFG, it is crucial to also learn the probabilistic distribution. One naive approach is to compare the marginal distributions $p(a, i)$, for the probability of symbol $a \in \mathbf{NT}_\ell$ appearing at position $i$. We observe a strong alignment between the generation probabilities and the ground-truth, included in Appendix B.2.

Another approach is to use the standard KL-divergence formula to compare the next-token prediction probability (as predicted by the transformer model) and the ground-truth. Let $p^*$ denote the distribution over strings in the true CFG and $p$ that from the transformer model. Let $S = \left\{x^{(i)}\right\}_{i \in [M]}$ be samples from the

---

[5]The last softmax layer converts the model outputs into a probability distribution over (next) symbols. We follow this distribution to generate the next symbol, reflecting the unaltered distribution learned by the transformer. This is the source of the "randomness of $F$" and is often referred to as using "temperature $\tau = 1$."

[6]`cfg3` family is large enough to ensure a negligible chance of a freshly sampled prefix of length 50 being seen during pretraining.

**predict NT ancestor (%)**

| | GPT | | | | | GPT_rel | | | | | GPT_rot | | | | | GPT_pos | | | | | GPT_uni | | | | | deBERTa | | | | | baseline (GPT_rand) | | | | |
|---|---|---|---|---|---|---|---|---|---|---|---|---|---|---|---|---|---|---|---|---|---|---|---|---|---|---|---|---|---|---|---|---|---|---|---|
| | NT6 | NT5 | NT4 | NT3 | NT2 | NT6 | NT5 | NT4 | NT3 | NT2 | NT6 | NT5 | NT4 | NT3 | NT2 | NT6 | NT5 | NT4 | NT3 | NT2 | NT6 | NT5 | NT4 | NT3 | NT2 | NT6 | NT5 | NT4 | NT3 | NT2 | NT6 | NT5 | NT4 | NT3 | NT2 |
| $cfg_{3b}$ | 100 | 100 | 100 | 100 | 100 | 100 | 100 | 100 | 100 | 100 | 100 | 100 | 100 | 100 | 100 | 100 | 100 | 100 | 100 | 100 | 100 | 100 | 100 | 100 | 100 | 100 | 100 | 100 | 99.7 | 99.9 | 85.0 | 65.7 | 56.8 | 61.5 | 62.7 |
| $cfg_{3i}$ | 99.6 | 99.7 | 99.6 | 99.2 | 99.7 | 99.6 | 99.7 | 99.6 | 99.2 | 99.7 | 99.6 | 99.7 | 99.6 | 99.2 | 99.8 | 99.6 | 99.7 | 99.6 | 99.3 | 99.8 | 99.6 | 99.7 | 99.6 | 99.3 | 99.8 | 99.7 | 99.7 | 99.7 | 99.2 | 99.4 | 84.6 | 71.7 | 64.6 | 66.4 | 65.2 |
| $cfg_{3h}$ | 99.7 | 98.3 | 98.3 | 99.2 | 100 | 99.7 | 98.1 | 97.8 | 99.0 | 100 | 99.7 | 98.4 | 98.2 | 99.3 | 100 | 99.7 | 98.5 | 98.5 | 99.4 | 100 | 99.7 | 98.6 | 98.6 | 99.4 | 100 | 99.9 | 99.8 | 99.8 | 99.7 | 100 | 67.5 | 47.2 | 50.6 | 66.3 | 92.8 |
| $cfg_{3g}$ | 100 | 99.2 | 95.6 | 94.6 | 97.3 | 100 | 99.3 | 96.7 | 97.2 | 99.0 | 100 | 99.3 | 96.6 | 97.2 | 99.0 | 100 | 99.3 | 96.7 | 96.9 | 98.8 | 100 | 99.4 | 97.0 | 97.2 | 98.9 | 100 | 99.5 | 95.5 | 85.6 | 90.5 | 70.8 | 56.4 | 49.4 | 57.0 | 73.1 |
| $cfg_{3f}$ | 100 | 97.6 | 94.3 | 88.4 | 85.9 | 100 | 97.5 | 94.8 | 92.9 | 93.5 | 100 | 97.7 | 95.2 | 93.3 | 94.2 | 100 | 97.9 | 95.6 | 93.5 | 93.9 | 100 | 98.2 | 95.8 | 93.2 | 93.5 | 100 | 99.6 | 96.3 | 84.0 | 77.5 | 71.3 | 49.9 | 44.6 | 59.1 | 68.6 |
| $cfg_{3e_1}$ | 100 | 100 | 100 | 100 | 100 | 100 | 100 | 100 | 100 | 100 | 100 | 100 | 100 | 100 | 100 | 100 | 100 | 100 | 100 | 100 | 100 | 100 | 100 | 100 | 100 | 100 | 100 | 100 | 100 | 99.8 | 45.4 | 27.6 | 34.6 | 47.2 | 76.3 |
| $cfg_{3e_2}$ | 99.9 | 100 | 100 | 100 | 100 | 99.8 | 100 | 100 | 100 | 100 | 99.9 | 100 | 100 | 100 | 100 | 99.9 | 100 | 100 | 100 | 100 | 99.9 | 100 | 100 | 100 | 100 | 100 | 100 | 100 | 100 | 99.9 | 36.0 | 16.6 | 23.5 | 44.6 | 78.3 |

Figure 5: After pre-training, hidden states of generative models implicitly encode NT-ancestor information. The $NT_\ell$ column represents the accuracy of predicting $\mathfrak{s}_\ell$, the NT ancestors on level $\ell$, via linear probing (4.2).

It also encodes NT boundaries (Appendix C.1); and such information is discovered gradually and *hierarchically* across layers and training epochs (Appendix C.2 and C.3). As a control, we apply the same probing method to a randomly-initialized GPT ($\mathsf{GPT}_{\mathsf{rand}}$) and a BERT-style encoder (DeBERTa). Both fail to recover deep NT structure, confirming the probe does not trivially succeed. In particular, BERT-like models are less effective at learning NT information on levels close to the CFG root.

true CFG distribution. Then, the KL-divergence can be estimated as follows:[7]

$$\frac{1}{|S|} \sum_{x \in S} \frac{1}{\mathbf{len}(x)+1} \sum_{i \in [\mathbf{len}(x)+1]} \sum_{t \in \mathbf{T} \cup \{\mathsf{eos}\}} \mathbf{Pr}_{p^*}[t \mid x_1, \ldots, x_{i-1}] \log \frac{\mathbf{Pr}_{p^*}[t|x_1,\ldots,x_{i-1}]}{\mathbf{Pr}_p[t|x_1,\ldots,x_{i-1}]}$$

(Above, $\mathbf{Pr}_p[t \mid x_1, \ldots, x_{i-1}]$ is the next-token distribution predicted by the model, and $\mathbf{Pr}_{p^*}[t \mid x_1, \ldots, x_{i-1}]$ is that from the ground-truth.[8] ) In Figure 4 (right) we compute KL-divergence using $M = 20000$ samples.

**Connection to DP.** Result 1-3 (e.g., learning the CFG's next-token distribution) is *merely a small step* towards showing that the model employs a DP-like approach. Dynamic programming (e.g., the inside-outside algorithm Baker (1979)) can compute next-token distributions of CFGs, and such algorithms can be implemented using nonlinear neural networks like transformers, achieving a global minimum in the auto-regressive training objective.[9] However, the mere existence of a dynamic-programming transformer to obtain the training objective's global minimum is not satisfactory. Does employing an AdamW stochastic optimizer for 100k iterations on the training objective yield such an algorithm? The remainder of this paper will delve deeper to address this question.

**Other Applications of Results 1–3.** While not the focus of this paper, our constructed CFGs also serve as a quick testbed for architecture designs. For instance, the strong performance of uniform attention aligns with the effectiveness of ALiBi (Press et al., 2021) and H-Alibi (Jelassi et al., 2024), and has motivated our follow-up work on modifying Transformer architectures to explicitly leverage short-window uniform attention (Allen-Zhu, 2025a;b). Additional robustness experiments for uniform attention—across data complexities and model sizes—are included in Appendix H.

**Open source.** Our data generators and evaluation tools, including accuracy evaluation and ground-truth distribution computation, are open-sourced as part of the package (Allen-Zhu, 2025b).

## 4 Results 4-5: How Do Transformers Learn CFGs?

In this section, we delve into the learned representation of the transformer to understand *how* it encodes CFGs. We employ various measurements to probe the representation and gain insights.

**Recall classical way to solve CFGs.** Given CFG $\mathcal{G}$, the classical way to reason about if a sequence $x$ satisfies $L(\mathcal{G})$ is to use dynamic programming (DP) (Sakai, 1961; Sipser, 2012). One possible implementation of DP involves using the function $\mathsf{DP}(i, j, a)$, which determines whether or not $x_{i+1}, x_{i+1} \ldots, x_j$ can be generated from symbol $a$ following the CFG rules. From this DP representation, a DP recurrent formula can be easily derived.[10] In the context of this paper, any sequence $x \sim L(\mathcal{G})$ that satisfies the CFG must satisfy

---

[7]Similar formula was also used in DuSell and Chiang (2022).

[8]There are many dynamic programming methods to compute $\mathbf{Pr}_{p^*}[t \mid x_1, \ldots, x_{i-1}]$ exactly; which one to use is irrelevant.

[9]This has been carefully explored for masked language modeling case in Zhao et al. (2023).

[10]For example, one can compute $\mathsf{DP}(i, j, a) = 1$ if and only if there exists $i = i_1 < i_2 < \cdots < i_k = j$ such that $\mathsf{DP}(i_r, i_{r+1}, b_r) = 1$ for all $r \in [k-1]$ and $a \to b_1, b_2, \ldots, b_k$ is a rule of the CFG. Implementing this naively would result in a $O(\mathbf{len}^4)$ algorithm for CFGs with a maximum rule length of 3. However, it can be implemented more efficiently with $O(\mathbf{len}^3)$ time by introducing auxiliary nodes (e.g., via binarization).

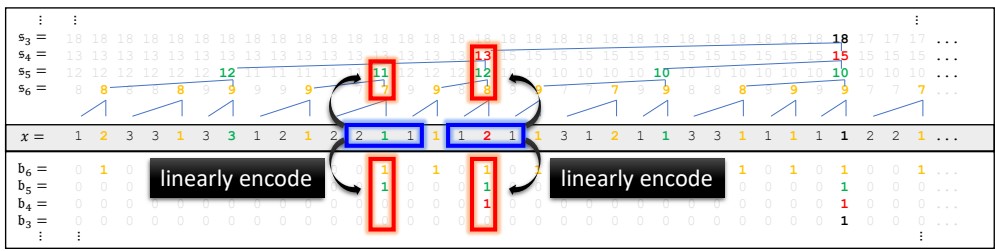

Figure 6: Illustration of Result 5: GPT's last layer hidden states at the **blue** positions linearly encode the NT ancestor/boundary in the **red** boxes. (They may not encode NT ancestors for smaller levels because that may not be information-theoretically possible — that is, reading *only* a prefix may not be enough to tell what such NTs are.)

the following conditions:

$$\mathfrak{b}_\ell(i) = 1, \mathfrak{b}_\ell(j) = 1, \forall k \in (i,j), \mathfrak{b}_\ell(k) = 0 \text{ and } \mathfrak{s}_\ell(j) = a \implies \mathsf{DP}(i,j,a) = 1 \tag{4.1}$$

(recall the NT-boundary $\mathfrak{b}_\ell$ and the NT-ancestor $\mathfrak{s}_\ell$ notions from Section 2.1 and Figure 2). Note that (4.1) is not an "if and only if" condition because there may be a subproblem $\mathsf{DP}(i,j,a) = 1$ that does not lie on the final CFG parsing tree but is still locally parsable by some valid CFG subtree. However, (4.1) provides a "backbone" of subproblems, where verifying all $\mathsf{DP}(i,j,a) = 1$ values in this backbone *certifies* that the sentence $x$ is a valid string from $L(\mathcal{G})$. It is worth mentioning that there are **exponentially many** implementations of the same DP algorithm[11] and **not all** $(i,j,a)$ tuples need to be computed in $\mathsf{DP}(i,j,a)$. Only those in the "backbone" are necessary.

**Connecting to transformer.** In this section, we investigate whether pre-trained transformer $F$ also implicitly encodes the NT ancestor and boundary information, which forms the basis for its structure reasoning capabilities. If so, it suggests the model contains sufficient information to support all the $\mathsf{DP}(i,j,a)$ values in the backbone. This is a significant finding, considering that transformer $F$ is trained solely on the auto-regressive task without any exposure to NT information. If the model encodes NT ancestor and boundary information after pretraining (as demonstrated in Results 4-5), this means it internally possesses the structural knowledge necessary not only for generation but also to *certify* the grammatical correctness of sentences according to the CFG. That is, its internal states effectively represent the parse tree.

### 4.1 Result 4: Transformer's Last Layer Encodes NT Ancestors/Boundaries

Let $l$ be the *last layer* of the transformer (other layers are studied in Appendix C.2). Given an input string $x$, we denote the hidden state of the transformer at layer $l$ and position $i$ as $E_i(x) \in \mathbb{R}^d$. We first investigate whether a linear function can predict $\left(\mathfrak{b}_1(i), \ldots, \mathfrak{b}_L(i)\right)_{i \in [\mathbf{len}(x)]}$ and $\left(\mathfrak{s}_1(i), \ldots, \mathfrak{s}_L(i)\right)_{i \in [\mathbf{len}(x)]}$ using the full $\left(E_i(x)\right)_{i \in [\mathbf{len}(x)]}$. If so, it implies that the last-layer hidden states *encode the CFG's structural information up to a linear transformation*, and the linear transformation does not depend on $x$ (a.k.a. *linear probing*).

**Multi-head linear probing (full).** Due to the high dimensionality of this linear function (e.g., $\mathbf{len}(x) = 300$ and $d = 768$ yield $300 \times 768$ dimensions) and *variable string lengths*, we propose a multi-head linear function for efficient learning. We consider a set of linear functions $f_r \colon \mathbb{R}^d \to \mathbb{R}^{|\mathbf{NT}|}$, where $r \in [H]$ and $H$ is the number of "heads". To predict any $\mathfrak{s}_\ell(i)$, we apply:

$$G_i(x) = \sum_{r \in [H], k \in [\mathbf{len}(x)]} w_{r, i \to k} \cdot f_r(E_k(x)) \in \mathbb{R}^{|\mathbf{NT}|} \tag{4.2}$$

where $w_{r, i \to k} \overset{\text{def}}{=} \frac{\exp(\langle P_{i,r}, P_{k,r} \rangle)}{\sum_{k' \in [\mathbf{len}(x)]} \exp(\langle P_{i,r}, P_{k',r} \rangle)}$ for trainable parameters $P_{i,r} \in \mathbb{R}^{d'}$. In words, $G_i$ is as a "multi-head attention" over linear functions: $f_r$ is the linear probing function mapping each hidden state (of dimension $d$) to NT-symbol logits; $P_{i,r}$ is the feature vector for position $i$ under head $r$, defining position-dependent weights $w_{r, i \to k}$ (independent of the input $x$), and they form matrices $w_{r, i \to k}$ that specify how much the hidden feature at position $k$ contributes to linearly predicting the NT symbol at position $i$.

---

[11]Each inner loop of the dynamic programming can proceed in any arbitrary order, not limited to $k = i..j$ or $k = j..i$, and the algorithm can prune and break early. This gives a safe estimate of at least $(n!)^{\Omega(n^2)}$ possible implementations. Furthermore, there are at least $2^{\Omega(n)}$ ways to perform binarization, meaning to break length-3 rules to length-2 ones. This is just to detect if a given string of length $n$ belongs to the CFG.

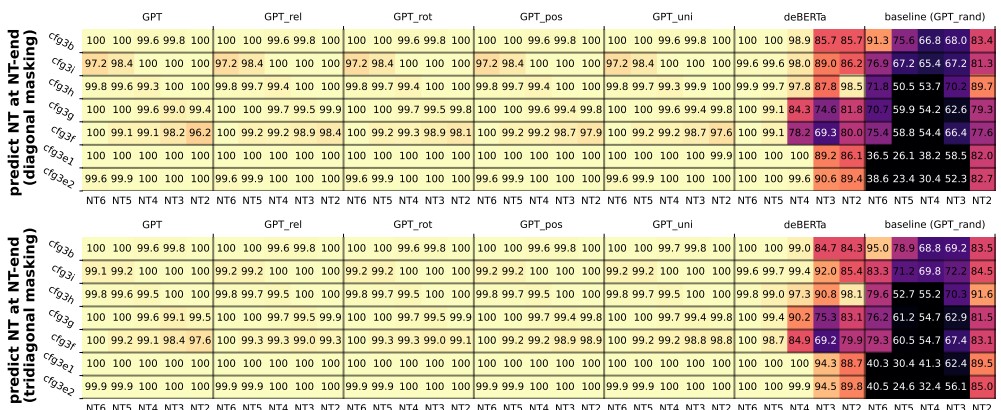

Figure 7: Generative models encode NT ancestors **almost exactly at** NT boundaries. The $NT_\ell$ column represents the accuracy to predict $\mathfrak{s}_\ell(i)$ at locations $i$ with $\mathfrak{b}_\ell(i) = 1$, via diagonal multi-head linear probing (4.3).

**Observation.** When applying the same probe to a random GPT or to DeBERTa (a BERT-style encoder trained with MLM), we find NT ancestor recovery fails at deeper levels—especially at NT boundaries—highlighting that our results reflect meaningful learned structure, not probing artifacts.

We train $G_i(x) \in \mathbb{R}^{|\mathbf{NT}|}$ using the cross-entropy loss to predict $\big(\mathfrak{s}_\ell(i)\big)_{\ell \in [L]}$. Despite having multiple heads,

$$G_i(x) \text{ is still a linear function over } (E_k(x))_{k \in [\mathbf{len}(x)]}$$

as the linear weights $w_{r, i \to k}$ depend only on positions $i$ and $k$, not on $x$ (except it depends on $\mathbf{len}(x)$). Similarly, we train $G'_i(x) \in \mathbb{R}^L$ using the logistic loss to predict the binary values $\big(\mathfrak{b}_\ell(i)\big)_{\ell \in [L]}$. In this process, the transformer is fixed after pretraining, only the linear weights are newly trained; and evaluations are performed using fresh new samples — never seen in pretraining or linear-weight training. Details are in Section A.4.

Using such multi-head linear probing, we discover that:

**Result 4** (Figure 5)**.** *Pre-training allows GPT models to* almost perfectly encode *the NT ancestor* $\mathfrak{s}_\ell(i)$ *and NT boundary* $\mathfrak{b}_\ell(i)$ *information in the last transformer layer's hidden states* $(E_k(x))_{k \in [\mathbf{len}(x)]}$, *up to a* linear *transformation.*

*(See Figure 5 for comparison against randomly-initialized* GPT_rand *or encoder model* deBERTa, *which fail to recover deep NT structure.)*

But, do we need this full layer for linear probing? We explore next.

## 4.2 Result 5: NT Ancestors are Encoded At NT Boundaries

In Result 4, we used the *full* hidden layer, $\big(E_i(x)\big)_{i \in [\mathbf{len}(x)]}$, to predict $\big(\mathfrak{s}_\ell(i)\big)_{\ell \in [L]}$ for *each* position $i$. This is essential since it's information-theoretically impossible to extract **all of $i$'s NT ancestors** by only reading $E_i(x)$ or even all hidden states to its *left*, especially if $x_i$ is the start of a string or a subtree in the CFG. But, how about those ones information-theoretically possible? In particular, how about predicting $\mathfrak{s}_\ell(i)$ at locations $i$ with $\mathfrak{b}_\ell(i) = 1$ — i.e., at the end of the CFG subtrees.

**Multi-head linear probing (diagonal).** We consider a neighborhood of position $i$ in the hidden states, say $E_{i \pm 1}(x)$, and use that for linear probing. In symbols, we replace $w_{r, i \to k}$ in (4.2) with zeros for $|i - k| > 1$ (tridiagonal masking), or with zeros for $i \neq k$ (diagonal masking).

$$G_i(x) = \sum_{r \in [H], k \in [\mathbf{len}(x)], |i-k| \le \delta} w_{r, i \to k} \cdot f_r(E_k(x)) \in \mathbb{R}^{|\mathbf{NT}|} \qquad \text{where } \delta = 0 \text{ or } 1 \qquad (4.3)$$

---

[11]deBERTa is a modern variant of BERT, equipped with relative attentions. It is expected that encoder models may not learn deep NT information, because in a masked-language modeling (MLM) task, the model only needs to figure out the missing token from its surrounding, say, 20 tokens. This can be done by pattern matching, as opposed to global planning like dynamic programming.

> **Result 5** (Figure 6+7)**.** *For GPT models, the information of position $i$'s NT ancestor/boundary is* locally encoded around position $i \pm 1$ *when $i$ is on the NT boundary. This is because:*
>
> - *At NT boundaries (i.e., $\mathfrak{b}_\ell(x) = 1$), we discover that diagonal or tridiagonal multi-head linear probing (4.3) is adequate for accurately predicting the NT ancestors $\mathfrak{s}_\ell(x)$ (see Figure 7).*
> - *Such masking is also sufficient for accurately predicting NT boundaries $\mathfrak{b}_\ell(i)$ (deferred to Figure 19 in Appendix C.1).*
>
> *In contrast, encoder models like* `deBERTa` *do* not *store deep NT information at the NT boundaries.*

**Related work.** Linear probing at least traces back to Hewitt and Manning (2019), who examines the correlation between BERT's hidden states and the parse tree distance metric (similar to NT-distance in our language). Subsequent studies (Shi et al., 2022; Zhao et al., 2023; Maudslay and Cotterell, 2021; Manning et al., 2020; Vilares et al., 2020; Wu et al., 2020; Arps et al., 2022) also explored probing techniques to suggest that BERT-like transformers can approximate CFGs from *natural languages*.

Our approach differs not only in the multi-head probing formula that we proposed; also that we use *synthetic* data to demonstrate that linear probing can *almost perfectly* recover NT ancestors and boundaries, even for complex and ambiguous CFG strings exceeding hundreds of tokens (c.f. English CFG has an average length of 28, see Appendix G). We focus on training *generative decoder-only* models; an encoder-based model like BERT (Kenton and Toutanova, 2019) or its modern variant `deBERTa` (He et al., 2020) may not learn *deep* (i.e., close to the CFG root) NT information very well, as shown in Result 4-5.

Our results, along with Section 5 next, shall provide evidence that generative language models like GPT-2 employ a DP-like approach to generate CFGs, while encoder-based models trained via MLM struggle to learn more complex/deeper CFGs.

## 5 Results 6-9: How Do Transformers Learn NTs?

We now delve into the attention patterns, which reveal the model's reasoning mechanisms. We demonstrate that these patterns mirror the CFG's syntactic structure and rules, with the transformer employing different attention heads to reason with NTs at different CFG levels.

### 5.1 Result 6: Position-Based Attention

We first note that the transformer's attention weights are primarily influenced by the tokens' relative distance. This holds true *even when* trained on the CFG data with *absolute* positional embedding. This implies that the transformer learns the CFG's regularity and periodicity through positional information, which it then uses for generation.

Formally, let $A_{l,h,j \to i}(x)$ for $j \geq i$ represent the attention weight for positions $j \to i$ at layer $l$ and head $h$ of the transformer, on input sequence $x$. For each layer $l$, head $h$, and distance $p \geq 0$, we compute the average of the partial sum $\sum_{1 \leq i' \leq i} A_{l,h,j \to i'}(x)$ over all data $x$ and pairs $i, j$ with $j - i = p$. We plot this cumulative sum for $l, h, p$ in Figure 8. We observe a strong correlation between the attention pattern and the relative distance $p = j - i$. The attention pattern is also *multi-scale*, with some attention heads focusing on shorter distances and others on longer ones.

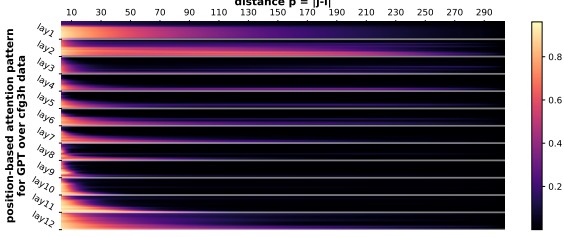

Figure 8: When trained on `cfg3h` using *absolute* positional embedding, `GPT` shows a position-based attention pattern. The 12 rows in each block represent attention heads. See Appendix D.1 for more experiments.

Motivated by this, we explore whether using position-based attention is *sufficient* to learn CFGs. In Figure 4, we find that `GPT`pos (or even `GPT`uni) performs well, surpassing the vanilla `GPT`, but not reaching the full potential of `GPT`rel. This supports the superior practical performance of relative-position based transformer variants (such as `GPT`rel, `GPT`rot, `deBERTa`) over their base

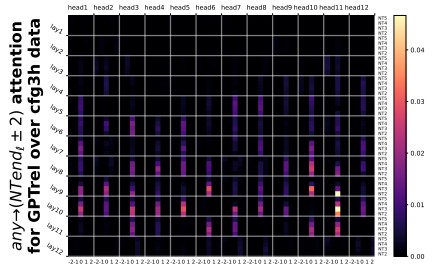
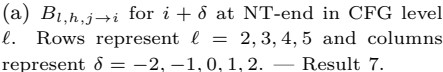
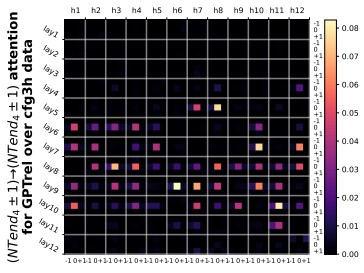
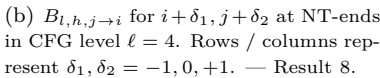
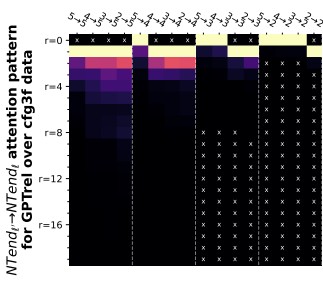

(a) $B_{l,h,j\to i}$ for $i+\delta$ at NT-end in CFG level $\ell$. Rows represent $\ell = 2,3,4,5$ and columns represent $\delta = -2,-1,0,1,2$. — Result 7.

(b) $B_{l,h,j\to i}$ for $i+\delta_1, j+\delta_2$ at NT-ends in CFG level $\ell = 4$. Rows / columns represent $\delta_1, \delta_2 = -1, 0, +1$. — Result 8.

(c) $B_{l,h,\ell'\to\ell,r}^{\text{end}\to\text{end}}$ for NT-ends between CFG levels $\ell' \to \ell$. Rows represent $r$ and columns $\ell' \to \ell$. "$\times$" means n/a entries. — Result 9

Figure 9: After pretrained on our CFG data, GPT model's attention has a strong bias towards " NT-end on level $\ell'$ to the most adjacent NT-end at $\ell$ ", even across different $\ell, \ell'$. For definitions see Section 5.2, more experiments see Appendix D.2, D.3 and D.4. This **provides evidence** for a DP-like approach to learn such hard, synthetic CFGs (discussions in Section 5.3).

models (GPT or BERT). On this other hand, this also indicates that **position-based attention alone is not enough for transformers to learn CFGs.**

## 5.2 Result 7-9: Boundary-Based Attention

Next, our idea is to *remove* the position-bias from the attention to examine the remainder. We discover that the transformer also learns a strong boundary-based attention pattern, where tokens on the NT-end boundaries typically **attend to the "most adjacent" NT-end boundaries**, see Figure 2 for an illustration. This pattern enables the transformer to effectively learn the hierarchical and recursive structure of the CFG, and generate output tokens based on the NT symbols and rules.

Formally, let $A_{l,h,j\to i}(x)$ for $j \geq i$ denote the attention weight for positions $j \to i$ at layer $l$ and head $h$ of the transformer, on input sequence $x$. Given a sample pool $\{x^{(n)}\}_{n\in[N]} \in L(\mathcal{G})$, we compute for each layer $l$, head $h$,[12]

$$\overline{A}_{l,h,p} = Average[\![A_{l,h,j\to i}(x^{(n)}) \mid n \in N, 1 \leq i \leq j \leq \mathbf{len}(x^{(n)}) \text{ s.t. } j - i = p]\!] \ ,$$

which represents the average attention between any token pairs of distance $p$ over the sample pool. To remove position-bias, we focus on $B_{l,h,j\to i}(x) \stackrel{\text{def}}{=} A_{l,h,j\to i}(x) - \overline{A}_{l,h,j-i}$ in this subsection. Our observation can be broken down into three steps.

**Result 7** (Figure 9(a)). $B_{l,h,j\to i}(x)$ *exhibits a strong bias towards* tokens $i$ at NT ends.

This can be seen in Figure 9(a), where we present the average value of $B_{l,h,j\to i}(x)$ over data $x$ and pairs $i, j$ where $i + \delta$ is the deepest NT-end on level $\ell$ (symbolically, $\mathfrak{b}^{\sharp}(i + \delta) = \ell$). The attention weights are highest when $\delta = 0$ and decrease rapidly for surrounding tokens.

While Result 7 already suggests that the transformer performs precise parsing relative to an unseen parse tree (since NT symbols were never revealed to the model), **we go further below**, showing stronger attention-pattern results that connect this behavior more directly to dynamic programming.

**Result 8** (Figure 9(b)). $B_{l,h,j\to i}(x)$ *favors pairs* $i, j$ both at NT ends *on the same level* $\ell$.

This can be seen in Figure 9(b), where we show the average value of $B_{l,h,j\to i}(x)$ over data $x$ and pairs $i, j$ where $\mathfrak{b}_{\ell}(i + \delta_1) = \mathfrak{b}_{\ell}(j + \delta_2) = 1$ for $\delta_1, \delta_2 \in \{-1, 0, 1\}$. It is maximized when $\delta_1 = \delta_2 = 0$.

The connection between Result 8 and DP will be detailed in Section 5.3 and illustrated in Figure 10. Briefly, to certify $\mathsf{DP}(i, j, a) = 1$ using a rule $a \mapsto b, c$, the model must locate a midpoint $k$ such that $i \dots k$ is generated by $b$ and $k + 1 \dots j$ by $c$. This entails reading token $k$ from position $j$, meaning the model must attend NT-ends at the same level — precisely what Result 8 certifies.

---

[12]Throughout this paper, we use $[\![\cdot]\!]$ to denote multi-sets that allow multiplicity, such as $[\![1, 2, 2, 3]\!]$. This allows us to conveniently talk about its set average.

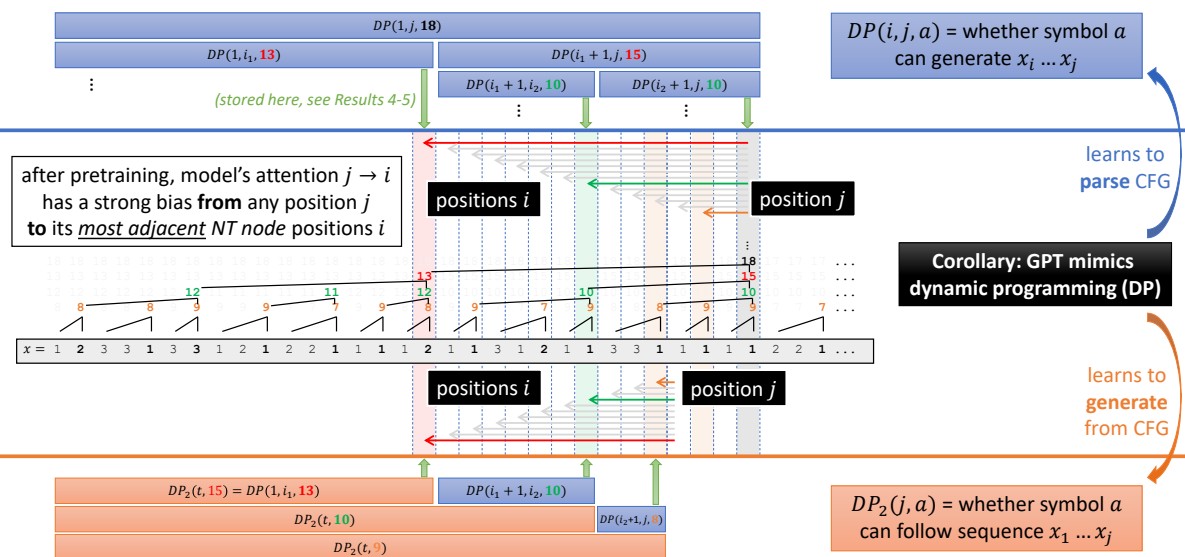

Figure 10: Illustration of how GPTs mimic dynamic programming. See discussions in Section 5.3.

We did not stop here either. To further demonstrate DP-like behavior, the attention should favor *adjacent* rather than arbitrary NT-ends. In the example above, positions $k$ and $j$ are adjacent NT-ends for $b$ and $c$ on the same level; more generally (see Section 5.3), it should also favor connections from $k \to i-1$, representing adjacent NT-ends across levels. This is exactly what Result 9 demonstrates next.

**Result 9** (Figure 9(c)). $B_{l,h,j \to i}(x)$ *favors* "adjacent" NT-end token pairs $i, j$ *across possibly different layers.*

We define "adjacency" as follows. We introduce $B_{l,h,\ell' \to \ell,r}^{\mathrm{end \to end}}$ to represent the average value of $B_{l,h,j \to i}(x)$ over samples $x$ and token pairs $i, j$ that are at the deepest NT-ends on levels $\ell, \ell'$ respectively (symbolically, $\mathfrak{b}^\sharp(i) = \ell \wedge \mathfrak{b}^\sharp(j) = \ell'$), and are at a distance $r \geq 0$ based on the ancestor indices on level $\ell$ (symbolically, $\mathfrak{p}_\ell(j) - \mathfrak{p}_\ell(i) = r$). We observe that $B_{l,h,\ell' \to \ell,r}^{\mathrm{end \to end}}$ is minimized at the smallest possible $r$:

- when $\ell' > \ell$, it is minimized at $r = 0$;
- when $\ell' \leq \ell$, it is minimized at $r = 1$ because $r = 0$ is undefined,[13]

In both cases, this shows NT-ends on level $\ell'$ attend to their closest NT-ends on on every level $\ell$ to their left. We emphasize that this is so *even after removing position bias.*[14]

In the next subsection, we explain why these results are strong supportive evidence that the GPT models have learned to mimic a DP algorithm.

## 5.3 Connecting Results 4,5,7,8,9 to Dynamic Programming (DP)

Dynamic programming involves *storage* of intermediate results and a *recurrent formula* to combine them. While identifying a specific DP implementation within transformers is infeasible due to numerous possibilities (Footnote 11), we can probe for crucial commonalities.

Section 4 demonstrated that transformers encode the DP's *storage* backbone—all necessary $\mathsf{DP}(i, j, a)$ values on the correct CFG parse tree—independent of any specific DP implementation.

For the *recurrent formula* (e.g., $\mathsf{DP}(k, j, a)$ derived from $\mathsf{DP}(k, i, b) \wedge \mathsf{DP}(i, j, c)$ for rule $a \mapsto b, c$), $\mathsf{DP}(k, i, b)$ is stored near $i$, while $\mathsf{DP}(k, j, a)$ and $\mathsf{DP}(i, j, c)$ are near $j$ (Result 5). This necessitates a *memory read* of $i$ at $j$ ($j \to i$), and $i, j$ are **adjacent NT-ends** at the same level. Result 8 shows that GPT models exhibit exactly such $j \to i$ attention pattern, suggesting an information flow consistent with DP. See Figure 10 (top).

**Further reading for DP/CFG experts: a two-step DP.** Transformers are both parsing and generative

---

[13]For any token pair $j \to i$ with $\ell = \mathfrak{b}^\sharp(i) \geq \mathfrak{b}^\sharp(j) = \ell'$ — meaning $i$ is at an NT-end closer to the root than $j$ — it satisfies $\mathfrak{p}_\ell(j) - \mathfrak{p}_\ell(i) \geq 1$ so their distance $r$ is strictly positive.
[14]Without removing position-bias, such a statement may be meaningless as the position bias may favor "adjacent" anything, including NT-end pairs.

algorithms. While the above DP is rather naive, it only allows parsing. CFG experts (or participants in competitions like IOI/USACO/ACM-ICPC) may recognize that the generative process requires a second DP:

$$\text{let } \mathsf{DP}_2(j, a) \text{ denote if prefix } x_1, \ldots, x_j \text{ can be followed by symbol } a \in \mathbf{NT} \cup \mathbf{T}.$$

If a rule $b \mapsto c, a$ holds and $\mathsf{DP}(i, j, c) \wedge \mathsf{DP}_2(i, b)$ are true, then $\mathsf{DP}_2(j, a)$ is also true. This is similar to the inside-outside algorithm (Baker, 1979). The model must perform a *memory read* from position $j$ to $i$, where $i$ is the **nearest NT-end to $j$ at a different level**. Unlike parsing DP; the generative $\mathsf{DP}_2$ uses information about the end of a prior constituent (at $i$) to inform the valid start (at $j$) for symbol $a$. The attention patterns (Result 9 and Figure 10 bottom), indicative of the model's reasoning process, is *also consistent with* this directional information flow.

Finally, to generate according to the CFG distribution, $\mathsf{DP}(k, j, a)$ and $\mathsf{DP}_2(j, a)$ must be converted into probabilities to compute the final conditional probability of the next token. The full pseudocode is provided in Algorithm 1 for reference, and it involves precisely the memory reads described above.

In sum, while pinpointing a specific DP implementation is impractical (recall Footnote 11), the DP backbone, including storage states and recurrent formulas, is evident in pretrained models' hidden states and attention patterns. This suggests that pretrained (decoder-only) transformers largely mimic dynamic programming, regardless of the specific DP implementation.

---

**Algorithm 1** the two-step DP to compute the next-token conditional probability

⋄ *For cleanness, shown for CFG rules of length 2; general case uses binarization (e.g., $a \mapsto b, c, d$ needs to be split into length-2 rules) to keep $O(n^3)$ complexity; code to be released on GitHub upon acceptance.*

---

**Input:** a (probabilistic) CFG and a (valid) prefix $x = x_1, x_2, \ldots, x_n$.
1: initialize $\mathsf{DP}(i, j, a) \leftarrow 0$ for all $i, j, a$ except $\mathsf{DP}(i, i, x_i) \leftarrow 1$ for $i = 1, 2, \ldots, n$
2: **for** $\ell = 2$ to $n$ **do**
3:      **for** $i = 1$ to $n - \ell + 1$ **do**
4:          $j \leftarrow i + \ell - 1$
5:          **for** $k = i$ to $j - 1$ **do**
6:              **for all** CFG rules $a \mapsto b, c$ with prob. $p$ **do**
7:                  $\mathsf{DP}(i, j, a) \leftarrow \mathsf{DP}(i, j, a) + p \cdot \mathsf{DP}(i, k, b) \cdot \mathsf{DP}(k+1, j, c)$      ⋄ *see Figure 10(top)*
8:                       ⋄ *now $\mathsf{DP}(i, j, a) = $ the probability that symbol $a \in \mathbf{NT} \cup \mathbf{T}$ can generate $x_{i:j}$*
9: initialize $\mathsf{DP}_2(k, a) \leftarrow 0$ for all $k, a$ except $\mathsf{DP}_2(0, root) \leftarrow 1$.
10: **for** $k = 0$ to $n$ **do**
11:      **for all** rules $a \mapsto b, c$ with prob. $p$ (from top to bottom levels in order) **do**
12:          $\mathsf{DP}_2(k, b) \leftarrow \mathsf{DP}_2(k, b) + p \cdot \mathsf{DP}_2(k, a)$
13:          **for** $m = k + 1$ to $n$ **do**
14:              $\mathsf{DP}_2(m, c) \leftarrow \mathsf{DP}_2(m, c) + p \cdot \mathsf{DP}_2(k, a) \cdot \mathsf{DP}(k + 1, m, b)$      ⋄ *see Figure 10(bottom)*
15:          ⋄ *now $\mathsf{DP}_2(k, a) = $ the probability that CFG generates (from root) a prefix $x_{1:k}$ followed by symbol $a \in \mathbf{NT} \cup \mathbf{T}$*
**Output:** for all $i \in [n]$: $\mathbf{Pr}[x_{i+1} = \mathsf{eos} \mid x_{1:i}] = \frac{\mathsf{DP}(1, i, root)}{\mathsf{DP}_2(i-1, x_i)}$ and $\mathbf{Pr}[x_{i+1} = t \mid x_{1:i}] = \frac{\mathsf{DP}_2(i, t)}{\mathsf{DP}_2(i-1, x_i)}$

---

# 6 Results 10-13: Extensions of CFGs

## 6.1 Result 10: Implicit CFGs

In an *implicit CFG*, terminal symbols represent bags of tokens with shared properties. For example, a terminal symbol like *noun* corresponds to a distribution over a bag of nouns, while *verb* corresponds to a distribution over a bag of verbs. These distributions can be non-uniform and overlapping, allowing tokens to be shared between different terminal symbols. During pre-training, the model learns to associate tokens with their respective syntactic or semantic categories, without prior knowledge of their specific roles in the CFG.

Formally, we consider a set of *observable tokens* $\mathbf{OT}$, and each terminal symbol $t \in \mathbf{T}$ in $\mathcal{G}$ is associated with a subset $\mathbf{OT}_t \subseteq \mathbf{OT}$ and a probability distribution $\mathcal{D}_t$ over $\mathbf{OT}_t$. The sets $(\mathbf{OT}_t)_t$ can be overlapping. To

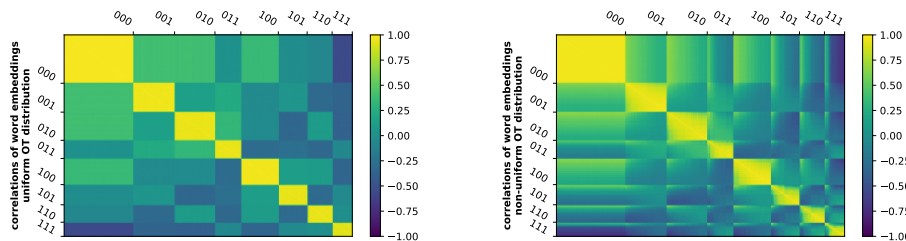

Figure 11: Language models learn implicit CFGs by using word embeddings to encode the (hidden) terminal symbol.

> We present word embedding correlations for GPT pre-trained on an implicit CFG with $|\mathbf{T}| = 3$ and vocabulary size $|\mathbf{OT}| = 300$. 300 rows/columns represent observable tokens $a \in \mathbf{OT}$. Label $ijk \in \{0,1\}^3$ in the figure indicates whether $a$ is in $\mathbf{OT}_t$ for the three choices $t \in \mathbf{T}$. Details are in Section 6.1.

generate a string from this implicit CFG, after generating $x = (x_1, x_2, \ldots, x_m) \sim L(\mathcal{G})$, for each terminal symbol $x_i$, we independently sample one element $y_i \sim \mathcal{D}_{x_i}$. After that, we observe the new string $y = (y_1, y_2, \cdots, y_m)$, and let this new distribution be called $y \sim L_O(\mathcal{G})$

We pre-train language models using samples from the distribution $y \sim L_O(\mathcal{G})$. During testing, we evaluate the success probability of the model generating a string that belongs to $L_O(\mathcal{G})$, given an input prefix $y_{:c}$. Or, in symbols,

$$\mathbf{Pr}_{y \sim L_O(\mathcal{G}) + \text{randomness of } F} \left[ (y_{:c}, F(y_{:c})) \in L_O(\mathcal{G}) \right] ,$$

where $F(y_{:c})$ represents the model's generated completion given prefix $y_{:c}$. (We again use dynamic programming to determine whether the output string is in $L_O(\mathcal{G})$.)

We summarize our finding below and deferring details to Appendix E.

> **Result 10** (Figure 11). *Generative language models can learn implicit CFGs very well. In particular, after pretraining, the token embeddings from the same subset $\mathbf{OT}_t$ are grouped together, indicating they use token embedding layer to encode the hidden terminal symbol information.*

## 6.2 Results 11-13: Robustness on Corrupted CFG

One may wish to pre-train a transformer *robust* against errors and inconsistencies in the input. For example, if input is a prefix with some tokens being corrupted or missing, then one may hope the transformer to correct the errors and still complete the sentence following the correct CFG rules. Robustness is an important property, as it reflects the generalization and adaptation ability of the transformer to reason effectively with real-world training data, which may not always follow the CFG perfectly (such as having grammar errors).

To test robustness, for each input prefix $x_{:c}$ of length $c$ that belongs to the CFG, we randomly select a set of positions $i \in [c]$ in this prefix — each with probability $\rho$ — and flip them i.i.d. with a random symbol in $\mathbf{T}$. Call the resulting prefix $\widetilde{x}_{:c}$. Next, we feed the *corrupted prefix* $\widetilde{x}_{:c}$ to the transformer $F$ and compute its generation accuracy in the uncorrupted CFG: $\mathbf{Pr}_{x \sim L(\mathcal{G}), F}[(x_{:c}, F(\widetilde{x}_{:c})) \in L(\mathcal{G})]$.

We not only consider clean pre-training, but also some versions of *robust pre-training*. That is, we randomly select $\gamma \in [0, 1]$ fraction of the training data and perturb them before feeding into the pre-training process. We compare three types of data perturbations.[15]

- (T-level random perturbation). Each $x_i$ w.p. 0.15 we replace it with a random symbol in $\mathbf{T}$.

- (NT-level random perturbation). Let $\ell = L - 1$ and recall $s_\ell = \left(s_{\ell,1}, s_{\ell,2}, \ldots, s_{\ell,m_{L-1}}\right)$ is the sequence of symbols at NT-level $\ell$. For each $s_{\ell,i}$, w.p. 0.10 we perturb it to a random symbol in $\mathbf{NT}_\ell$; and then generate $x = s_L$ according to this perturbed sequence.

- (NT-level deterministic perturbation). Let $\ell = L - 1$ and fix a permutation $\pi$ over symbols in $\mathbf{NT}_\ell$. For each $s_{\ell,i}$, w.p. 0.05 we perturb it to its next symbol in $\mathbf{NT}_{L-1}$ according to $\pi$; and then generate $x = s_L$ according to this perturbed sequence.

---

[15]One can easily extend our experiments by considering other types of data corruption (for evaluation), and other types of data perturbations (for training). We refrain from doing so because it is beyond the scope of this paper.

**generation acc (%) for cfg3b**

| | NT-level 0.1 random perturbation | | | | | | | | | | T-level 0.15 random perturbation | | | | | | | | | | NT-level 0.05 deterministic permutation | | | | | | | | | | |
|---|---|---|---|---|---|---|---|---|---|---|---|---|---|---|---|---|---|---|---|---|---|---|---|---|---|---|---|---|---|---|---|
| cut0 τ=0.1 | 100 | 100 | 100 | 100 | 100 | 100 | 100 | 100 | 100 | 100 | 100 | 100 | 100 | 100 | 100 | 100 | 100 | 100 | 100 | 100 | 99.8 | 100 | 100 | 100 | 100 | 100 | 100 | 100 | 100 | 100 | 100 |
| cut0 τ=0.2 | 98.7 | 100 | 100 | 100 | 100 | 100 | 100 | 100 | 100 | 100 | 99.2 | 99.9 | 100 | 100 | 100 | 99.9 | 100 | 100 | 100 | 100 | 98.5 | 100 | 100 | 100 | 100 | 100 | 100 | 100 | 100 | 100 | 100 |
| cut0 τ=1 | 0.0 | 14.3 | 24.7 | 39.8 | 44.4 | 55.7 | 64.5 | 73.5 | 82.6 | 91.8 | 0.0 | 14.1 | 22.8 | 35.3 | 44.9 | 58.2 | 65.4 | 75.5 | 83.6 | 92.5 | 0.0 | 14.7 | 26.9 | 38.5 | 49.8 | 56.8 | 65.5 | 75.2 | 81.5 | 91.8 | 99.8 |
| corrupted cut50 τ=0.1 | 78.3 | 78.9 | 80.6 | 78.0 | 79.1 | 78.6 | 79.5 | 78.6 | 76.4 | 77.9 | 82.6 | 80.4 | 80.6 | 80.4 | 81.7 | 82.6 | 81.4 | 81.7 | 80.8 | 80.8 | 60.4 | 58.3 | 56.5 | 58.1 | 60.4 | 59.1 | 60.6 | 57.5 | 58.9 | 56.9 | 30.0 |
| corrupted cut50 τ=0.2 | 77.4 | 78.7 | 80.0 | 76.6 | 77.8 | 78.2 | 78.3 | 77.3 | 74.9 | 77.9 | 81.1 | 81.1 | 80.5 | 79.6 | 81.2 | 82.0 | 81.4 | 80.7 | 80.0 | 80.4 | 59.5 | 57.7 | 55.9 | 57.6 | 59.2 | 58.8 | 59.7 | 57.2 | 57.8 | 57.1 | 30.3 |
| corrupted cut50 τ=1 | 0.0 | 0.5 | 0.5 | 0.6 | 0.5 | 0.3 | 0.6 | 0.4 | 0.5 | 0.7 | 0.0 | 0.4 | 0.5 | 0.8 | 0.2 | 0.3 | 0.5 | 0.6 | 0.7 | 0.6 | 0.0 | 0.1 | 0.4 | 0.4 | 0.4 | 0.5 | 0.9 | 0.5 | 0.3 | 0.3 | 29.6 |
| cut50 τ=0.1 | 100 | 100 | 100 | 100 | 100 | 100 | 100 | 100 | 100 | 100 | 100 | 100 | 100 | 100 | 100 | 100 | 100 | 100 | 100 | 100 | 99.4 | 100 | 100 | 100 | 100 | 100 | 100 | 100 | 100 | 100 | 100 |
| cut50 τ=0.2 | 99.2 | 100 | 100 | 100 | 100 | 100 | 100 | 100 | 100 | 100 | 99.6 | 100 | 100 | 100 | 100 | 100 | 100 | 100 | 100 | 100 | 98.4 | 100 | 100 | 100 | 100 | 100 | 100 | 100 | 100 | 100 | 100 |
| cut50 τ=1 | 0.0 | 91.5 | 95.7 | 97.1 | 98.1 | 98.7 | 99.2 | 99.0 | 99.5 | 99.4 | 0.0 | 92.8 | 96.2 | 97.6 | 98.2 | 99.1 | 99.3 | 99.4 | 99.5 | 99.7 | 0.0 | 83.4 | 90.6 | 94.0 | 96.2 | 97.2 | 98.1 | 98.7 | 99.2 | 99.3 | 99.9 |
| | 1.0 | 0.9 | 0.8 | 0.7 | 0.6 | 0.5 | 0.4 | 0.3 | 0.2 | 0.1 | 1.0 | 0.9 | 0.8 | 0.7 | 0.6 | 0.5 | 0.4 | 0.3 | 0.2 | 0.1 | 1.0 | 0.9 | 0.8 | 0.7 | 0.6 | 0.5 | 0.4 | 0.3 | 0.2 | 0.1 | clean |

**--------pre-training method--------**

**------------pre-training data perturbation ratio $\gamma$ OR clean data------------**

Figure 12: Generation accuracies for models pre-trained cleanly VS pre-trained over perturbed data, on clean or corrupted prefixes with cuts $c = 0$ or $c = 50$, using generation temperatures $\tau = 0.1, 0.2, 1.0$.

> **Observation.** In Rows 4/5, by comparing against the last column, we see it is *beneficial* to include low-quality data (e.g. grammar mistakes) during pre-training. The amount of low-quality data could be little ($\gamma = 0.1$ fraction) or large (*every training sentence may have grammar mistake*). The transformer also learns a "mode switch" between the "correct mode" or not; details in Section 6.2. (More datasets, see Figure 28.)

We focus on $\rho = 0.15$ with a wide range of perturbation rate $\tau = 0.0, 0.1, \ldots, 0.9, 1.0$. We present our findings in Figure 12. The main message is:

> **Result 11** (Figure 12, rows 4/5). *When pretrained over clean data, GPT models are* not so robust *to "grammar mistakes." It is* beneficial *to include corrupted or low-quality pretrain data.*

Specifically, GPT models achieve only $\sim 30\%$ accuracy when pretrained over clean data $x \sim L(\mathcal{G})$. If we pretrain from perturbed data — *both* when $\gamma = 1.0$ so all data are perturbed, *and* when $\gamma = 0.1$ so we have a small fraction of perturbed data — GPT can achieve $\sim 79\%, 82\%$ and $60\%$ robust accuracies respectively using the three types of data perturbations (rows 4/5 of Figure 12).

Next, we take a closer look. If we use temperature $\tau = 1$ for generation:

> **Result 12** (Figure 12, rows 3/6/9). *Pre-training on corrupted data teaches model a* mode switch*.*
>
> - *Given a correct prefix, it mostly completes with a correct string in the CFG (Row 9);*
> - *Given a corrupted prefix, it* always *completes sentences with grammar mistakes (Row 6);*
> - *When given no prefix, it generates corrupted strings with probability close to $\gamma$ (Row 3).*

By comparing the generation accuracies across different $\tau$ and $\gamma$, we observe:

> **Result 13** (Figure 12, rows 4/5/6). High robust accuracy *is achieved when generating* using low temperatures $\tau$,[16] *and is* not sensitive *to $\gamma$ –* even when *model is trained* totally on corrupted data *($\gamma = 1.0$).*

This should not be surprising given that the language model learned a "mode switch." Using low temperature encourages the model to, for each next token, pick a more probable solution. This allows it to achieve good robust accuracy *even when* the model is trained totally on corrupted data ($\gamma = 1.0$). Note this is consistent with practice: when feeding a pre-trained completion model (such as Llama or GPT-3-davinci003) with prompts of grammar mistakes, it tends to produce texts also with (even new!) grammar mistakes when using a large temperature.

Our experiments suggest that, additional instruct fine-tuning may be necessary, if one wants the model to *always* stay in the "correct mode" even for high temperatures. This is beyond the scope of this paper.

# 7 Related Work and Conclusion

**Related Works.** Transformers can encode some CFGs, particularly those related to human languages (Hewitt and Manning, 2019; Shi et al., 2022; Zhao et al., 2023; Maudslay and Cotterell, 2021; Manning et al., 2020; Vilares et al., 2020; Wu et al., 2020; Arps et al., 2022). Deletang et al. (2023) explored transformers'

---

[16]Recall, when temperature $\tau = 0$ the generation is greedy and deterministic; when $\tau = 1$ it reflects the unaltered distribution learned by the transformer; when $\tau > 0$ s small it encourages the transformer to output "more probable" tokens.

learnability on languages within the Chomsky hierarchy, including CFGs. However, the *inner mechanisms* of how transformers solve these tasks remain unclear.

Some works can *precisely* interpret each neuron's function but focus on simpler tasks and architectures. For example, Nanda et al. (2023) studied 1- or 2-layer transformers with context length 3 for arithmetic addition. We focus on the 100M-sized GPT-2 model with a context length over 300. While we cannot determine each neuron's function, we have identified roles of some heads and hidden states that correlate with DP.

Murty et al. (2023) explored methods beyond linear probing to deduce tree structures learned by transformers. They designed a score to quantify a transformer's "tree-like" nature, showing it becomes more tree-like during training. Our Figure 21 in Appendix C.3 supports these findings. *(This paper first appeared in May 2023, so we focus on related works before that.)*

**Conclusion.** This paper demonstrates how transformers learn to reason over challenging synthetic context-free grammars (CFGs), revealing a close alignment between their internal representations and the information flow characteristic of dynamic programming computations underlying parsing and generation. Our primary aim was to provide a controlled interpretability setting that enables mechanistic insights into attention, hierarchical representations, and algorithmic reasoning in transformer models.

Our approach differs from prior interpretability work (e.g., (Elhage et al., 2021; Olsson et al., 2022; Wang et al., 2022)), which primarily analyzes large models trained on real-world corpora where noise and uncontrolled correlations can obscure underlying algorithmic structure. In contrast, we adopt a complementary methodology: leveraging clean, synthetic data to expose and mechanistically analyze how transformers implement well-defined, computationally deep reasoning processes.

Beyond the core analysis, we introduced multi-head linear probing as a diagnostic tool for analyzing complex internal computations, potentially enabling deeper studies of larger models on similarly structured tasks. We also derived several corollary findings, including: (i) absolute positional embeddings are inferior to relative and rotary embeddings for deep hierarchical reasoning; (ii) uniform attention alone is surprisingly effective, motivating our follow-up work on Canon layers (Allen-Zhu, 2025a;b); (iii) encoder-only models (e.g., BERT, DeBERTa) struggle with *deep* structural reasoning on CFGs compared to autoregressive models (e.g., GPT); and (iv) injecting structural or syntactic noise into pretraining data substantially improves robustness to corrupted language prompts.

While synthetic CFGs provide well-defined benchmarks for isolating compositional and hierarchical behavior and exposing fundamental computational mechanisms, they do not capture the full diversity of natural language or intelligence—much like other synthetic algorithmic tasks such as sorting or ListOps. Accordingly, we view CFGs as an abstraction rather than a substitute for real-world evaluation. In complementary work, we extend this synthetic-data approach to grade-school math and reasoning (Parts 2.1+2.2 (Ye et al., 2025a;b)), knowledge storage, extraction, and manipulation (Parts 3.1+3.2+3.3 (Allen-Zhu and Li, 2024; 2025a;b)), and leverage these insights for architecture design within a unified synthetic pretraining playground (Parts 4.1+4.2 (Allen-Zhu, 2025a;b)).

## Acknowledgements

The title *Physics of Language Models* was jointly conceived and designed by ZA and Xiaoli Xu. We would like to thank Lin Xiao, Sida Wang and Hu Xu for many helpful conversations. We would like to extend special thanks to Ian Clark, Gourab De, Anmol Mann, and Max Pfeifer from W&B, as well as Nabib Ahmed, Giri Anantharaman, Lucca Bertoncini, Henry Estela, Liao Hu, Caleb Ho, Will Johnson, Apostolos Kokolis, and Shubho Sengupta from Meta FAIR NextSys; without their invaluable support, the experiments in this paper would not have been possible.

# Appendix

## A    Experiment Setups

### A.1    Dataset Details

We construct seven synthetic CFGs of depth $L = 7$ with varying levels of learning difficulty. It can be inferred that the greater the number of T/NT symbols, the more challenging it is to learn the CFG. For this reason, to push the capabilities of language models to their limits, we primarily focus on cfg3b, cfg3i, cfg3h, cfg3g, cfg3f, which are of sizes $(1, 3, 3, 3, 3, 3, 3)$ and present increasing levels of difficulty. Detailed information about these CFGs is provided in Figure 13:

- In cfg3b, we construct the CFG such that the degree $|\mathcal{R}(a)| = 2$ for every NT $a$. We also ensure that in any generation rule, consecutive pairs of T/NT symbols are distinct.

  The 25%, 50%, 75%, and 95% percentile string lengths are $251, 278, 308, 342$ respectively.

- In cfg3i, we set $|\mathcal{R}(a)| = 2$ for every NT $a$. We remove the requirement for distinctness to make the data more challenging than cfg3b.

  The 25%, 50%, 75%, and 95% percentile string lengths are $276, 307, 340, 386$ respectively.

- In cfg3h, we set $|\mathcal{R}(a)| \in \{2, 3\}$ for every NT $a$ to make the data more challenging than cfg3i.

  The 25%, 50%, 75%, and 95% percentile string lengths are $202, 238, 270, 300$ respectively.

- In cfg3g, we set $|\mathcal{R}(a)| = 3$ for every NT $a$ to make the data more challenging than cfg3h.

  The 25%, 50%, 75%, and 95% percentile string lengths are $212, 258, 294, 341$ respectively.

- In cfg3f, we set $|\mathcal{R}(a)| \in \{3, 4\}$ for every NT $a$ to make the data more challenging than cfg3g.

  The 25%, 50%, 75%, and 95% percentile string lengths are $191, 247, 302, 364$ respectively.

*Remark* A.1. From the examples in Figure 13, it becomes evident that for grammars $\mathcal{G}$ of depth 7, proving that a string $x$ belongs to $L(\mathcal{G})$ is highly non-trivial, even for a human being, and even when the CFG rules are known. The standard method of demonstrating $x \in L(\mathcal{G})$ is through dynamic programming. We further discuss what we mean by a CFG's "difficulty" in Appendix G, and provide additional experiments beyond the cfg3 data family.

*Remark* A.2. The dataset cfg3f lies at the difficulty threshold that GPT2-small can master under our pretraining setup (see Figure 31; training details in later sections). Although deeper, more complex CFGs are possible (as we explore in (Allen-Zhu, 2025a)), they would require a larger model and extended training. We focus on cfg3f because it already provides compelling evidence for our findings.

Simultaneously, to illustrate that transformers can learn CFGs with larger $|\mathbf{NT}|$ or $|\mathbf{T}|$, we construct datasets cfg3e1 and cfg3e2 respectively of sizes $(1, 3, 9, 27, 81, 27, 9)$ and $(1, 3, 9, 27, 27, 9, 4)$. They are too lengthy to describe so we include them in an attached txt file in Appendix G.2 and our repo Allen-Zhu (2025b).

### A.2    Model Architecture Details

We define `GPT` as the standard GPT2-small architecture (Radford et al., 2019), which consists of 12 layers, 12 attention heads per layer, and 768 ($=12 \times 64$) hidden dimensions. We pre-train `GPT` on the aforementioned datasets, starting from random initialization. For a baseline comparison, we also implement DeBERTa (He et al., 2020), resizing it to match the dimensions of GPT2 — thus also comprising 12 layers, 12 attention heads, and 768 dimensions.

**Architecture size.**    We have experimented with models of varying sizes and observed that their learning capabilities scale with the complexity of the CFGs. To ensure a fair comparison and enhance reproducibility, we primarily focus on models with 12 layers, 12 attention heads, and 768 dimensions. The transformers constructed in this manner consist of 86M parameters.

**Modern GPTs with relative attention.**    Recent research (He et al., 2020; Su et al., 2021; Black et al.,

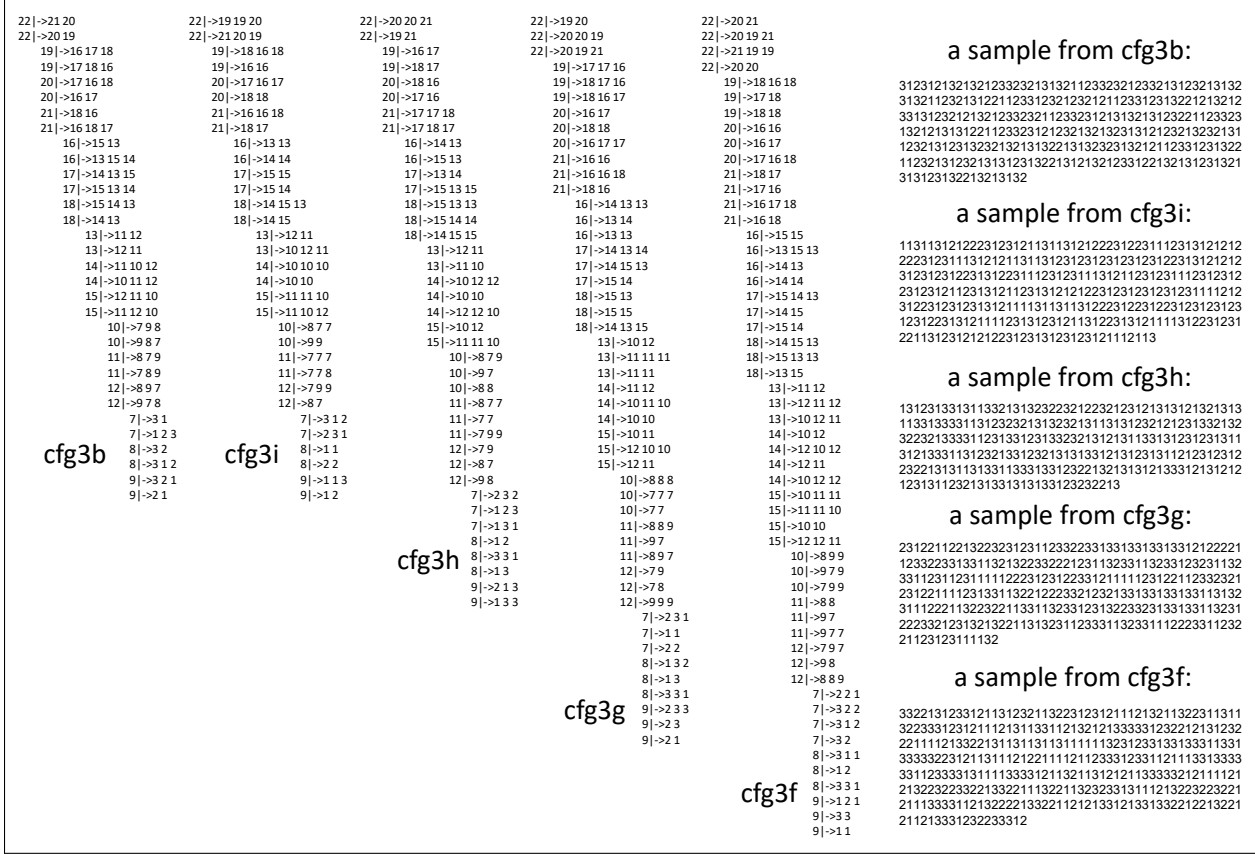

Figure 13: The context-free grammars cfg3b, cfg3i, cfg3h, cfg3g, cfg3f that we primarily use in this paper, together with a sample string from each of them. (Follow-up work (Allen-Zhu, 2025a) has further extended these datasets.)

**Observation.** Although those CFGs are only of depth 7, they are capable of generating sufficiently long and hard instances; after all, even when the CFG rules are given, the typical way to decide if a string $x$ belongs to the CFG language $x \in L(\mathcal{G})$ may require dynamic programming.

2022) has demonstrated that transformers can significantly improve performance by using attention mechanisms based on the *relative* position differences of tokens, as opposed to the absolute positions used in the original GPT2 (Radford et al., 2019) or BERT (Kenton and Toutanova, 2019). There are two main approaches to achieve this. The first is to use a "relative positional embedding layer" on $|j-i|$ when calculating the attention from $j$ to $i$ (or a bucket embedding to save space). This approach is the most effective but tends to train slower. The second approach is to apply a rotary positional embedding (RoPE) transformation (Su et al., 2021) on the hidden states; this is known to be slightly less effective than the relative approach, but it can be trained much faster.

We have implemented both approaches. We adopted the RoPE implementation from the GPT-NeoX-20B project (along with the default parameters), but downsized it to fit the GPT2 small model. We refer to this architecture as GPT$_{\text{rot}}$. Since we could not find a standard implementation of GPT using relative attention, we re-implemented GPT2 using the relative attention framework from DeBERTa (He et al., 2020). (Recall, DeBERTa is a variant of BERT that effectively utilizes relative positional embeddings.) We refer to this architecture as GPT$_{\text{rel}}$.

**Weaker GPTs utilizing only position-based attention.** For the purpose of analysis, we also consider two significantly weaker variants of GPT, where the attention matrix *exclusively depends* on the token positions, and not on the input sequences or hidden embeddings. In other words, the attention pattern remains *constant* for all input sequences.

We implement GPT$_{\text{pos}}$, a variant of GPT$_{\text{rel}}$ that restricts the attention matrix to be computed solely using the

(trainable) relative positional embedding. This can be perceived as a GPT variant that *maximizes the use of position-based attention.* We still choose the 12-layer, 12-head, 768-dim structure.

We also implement $\mathtt{GPT_{uni}}$, a 12-layer, 8-head, 1024-dimensional Transformer where the attention matrices are *fixed.* Specifically, for each $h \in [8]$, the $h$-th head consistently applies a uniform average over the previous $2^h - 1$ tokens. This can be viewed as a GPT variant that *uses the simplest form of position-based attention.* Since $\mathtt{GPT_{uni}}$ lacks key and value matrices, its parameter count differs from standard GPT variants. A GPT2-small-sized $\mathtt{GPT_{uni}}$—i.e., one with 12 layers and 840 hidden dimensions—matches its parameter count. As we show in Appendix H, this smaller version performs similarly to the 1024-dimensional $\mathtt{GPT_{uni}}$.

*Remark* A.3. It should not be surprising that $\mathtt{GPT_{pos}}$ or $\mathtt{GPT_{uni}}$ perform much worse than other GPT models on real-life wikibook pre-training. However, once again, we use them only for *analysis purpose* in this paper, as we wish to demonstrate what is the maximum power of GPT when only using position-based attention to learn CFGs, and what is the marginal effect when one goes *beyond* position-based attention.

**Features from random transformer.** Finally we also consider a randomly-initialized $\mathtt{GPT_{rel}}$, and use those random features for the purpose of predicting NT ancestors and NT ends. This serves as a baseline, and can be viewed as the power of the so-called (finite-width) neural tangent kernel (Allen-Zhu et al., 2019). We call this $\mathtt{GPT_{rand}}$.

### A.3 Pre-Training Details

For each sample $x \sim L(\mathcal{G})$ we append it to the left with a BOS token and to the right with an EOS token. Then, following the tradition of language modeling (LM) pre-training, we concatenate consecutive samples and randomly cut the data to form sequences of a fixed window length 512.

As a baseline comparison, we also applied DeBERTa on a masked language modeling (MLM) task for our datasets. We use standard MLM parameters: 15% masked probability, in which 80% chance of using a masked token, 10% chance using the original token, and 10% chance using a random token.

We use standard initializations from the huggingface library. For GPT pre-training, we use AdamW with $\beta = (0.9, 0.98)$, weight decay 0.1, learning rate 0.0003, and batch size 96. We pre-train the model for 100k iterations, with a linear learning rate decay.[17] For DeBERTa, we use learning rate 0.0001 which is better and 2000 steps of learning rate linear warmup.

Throughout the experiments, for both pre-training and testing, we only use **fresh samples** from the CFG datasets (thus using 4.9 billion tokens $= 96 \times 512 \times 100k$). We have also tested pre-training with a finite training set of $100m$ tokens; and the conclusions of this paper stay similar. To make this paper clean, we choose to stick to the infinite-data regime in this version of the paper, because it enables us to make negative statements (for instance about the vanilla GPT or DeBERTa, or about the learnability of NT ancestors / NT boundaries) without worrying about the sample size. Please note, given that our CFG language is very large (e.g., length 300 tree of length-2/3 rules and degree 4 would have at least $4^{300/3}$ possibility), there is *almost no chance that training/testing hit the same sentence.*

As for the reproducibility of our result, we did not run each pre-train experiment more than once (or plot any confidence interval). This is because, rather than repeating our experiments identically, it is obviously more interesting to use the resources to run it against different datasets and against different parameters. We pick the best model using the perplexity score from each pre-training task. When evaluating the generation accuracy in Figure 4, we have generated more than 20000 samples for each case, and present the diversity pattern accordingly in Figure 14.

We test our results using a mixture of V100 and A100 GPUs (on A100, pretraining a model takes less than a day using 4GPUs), even when using float32.

---

[17]We have slightly tuned the parameters to make pre-training go best. We noticed for training GPTs over our CFG data, a warmup learning rate schedule is not needed.

### A.4 Predict NT ancestor and NT boundary

Recall from Section 4.1 that we have proposed to use a multi-head linear function to probe whether or not the hidden states of a transformer, implicitly encodes the NT ancestor and NT boundary information for each token position. Since this linear function can be of dimension $512 \times 768$ — when having a context length 512 and hidden dimension 768 — recall in (4.2), we have proposed to use a multi-head attention to construct such linear function for efficient learning purpose. This significantly reduces sample complexity and makes it much easier to find the linear function.

In our implementation, we choose $H = 16$ heads and hidden dimension $d' = 1024$ when constructing this position-based attention in (4.2). We have also tried other parameters but the NT ancestor/boundary prediction accuracies are not very sensitive to such architecture change. We again use AdamW with $\beta = (0.9, 0.98)$ but this time with learning rate 0.003, weight decay 0.001, batch size 60 and train for 30k iterations.

Once again we use *fresh new samples* when training such linear functions. When evaluating the accuracies on predicting the NT ancester / boundary information, we also use fresh new samples. Recall our CFG language is sufficiently large so there is negligible chance that the model has seen such a string during training.

## B More Experiments on Results 2-3 (Generation)

Diversity can be estimated through entropy. Given a distribution $p$ over strings and a sampled subset $S = \{x^{(i)}\}_{i \in [M]}$ from $p$, for any string $x \in S$, denote by $\textbf{len}(x)$ its length so $x = (x_1, \ldots, x_{\textbf{len}(x)})$, and denote by $x_{\textbf{len}(x)+1} = \textsf{eos}$. The entropy in bits for $p$ can be estimated by

$$-\tfrac{1}{|S|} \sum_{x \in S} \sum_{i \in [\textbf{len}(x)+1]} \log_2 \textbf{Pr}_p \left[ x_i \mid x_1, \ldots, x_{i-1} \right]$$

We compare the entropy of the true CFG distribution and the transformer's output distribution using $M = 20000$ samples in Figure 4 (middle).

Diversity can also be estimated using the birthday paradox to lower bound the support size of a distribution (Arora and Zhang, 2017). Given a distribution $p$ over strings and a sampled subset $S = \{x^{(i)}\}_{i \in [M]}$ from $p$, if every pair of samples in $S$ are distinct, then with good probability the support of $p$ is of size at least $\Omega(M^2)$. In Appendix B.1, we conducted an experiment with $M = 20000$. We performed a birthday paradox experiment from every symbol $a \in \textbf{NT}_{\ell_1}$ to some other level $\ell_2 > \ell_1$, comparing that with the ground truth. For instance, we confirmed for the cfg3f dataset, there are at least $\Omega(M^2)$ distinct sentential forms that can be derived from a symbol in level 1 to level 5, or from level 2 to level 6, etc. — not to mention from the root in $\textbf{NT}_1$ to the leaf on level 7. In particular, $M^2$ is already more than the number of parameters in the model.

From both experiments, we conclude that the pre-trained model **does not rely on simply memorizing** a small set of patterns to learn the CFGs.

### B.1 Generation Diversity via Birthday Paradox

Since "diversity" is influenced by the length of the input prefix, the length of the output, and the CFG rules, we want to carefully define what we measure.

Given a sample pool $x^{(1)}, \ldots, x^{(M)} \in L(\mathcal{G})$, for every symbol $a \in \textbf{NT}_{\ell_1}$ and some later level $\ell_2 \geq \ell_1$ that is closer to the leaves, we wish to define a *multi-set* $\mathcal{S}_{a \to \ell_2}$ that describes *all possible generations from $a \in \textbf{NT}_{\ell_1}$ to $\textbf{NT}_{\ell_2}$* in this sample pool. Formally,

**Definition B.1.** *For $x \in L(\mathcal{G})$ and $\ell \in [L]$, we use $\mathfrak{s}_\ell(i..j)$ to denote the sequence of NT ancestor symbols on level $\ell \in [L]$ from position $i$ to $j$ with distinct ancestor indices:*[18]

$$\mathfrak{s}_\ell(i..j) = (\mathfrak{s}_\ell(k))_{k \in \{i, i+1, \ldots, j\} \ s.t. \ \mathfrak{p}_\ell(k) \neq \mathfrak{p}_\ell(k+1)}$$

---

[18]With the understanding that $\mathfrak{p}_\ell(0) = \mathfrak{p}_\ell(\textbf{len}(x) + 1) = \infty$.

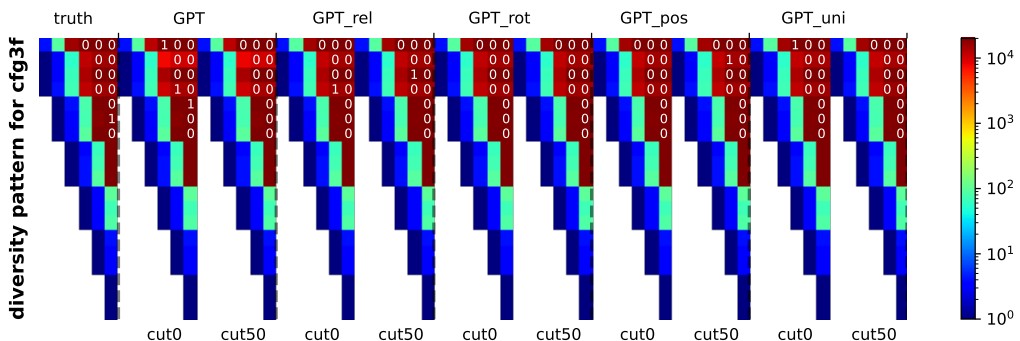

Figure 14: Comparing the generation diversity $\mathcal{S}^{\text{truth}}_{a\rightarrow\ell_2}$ and $\mathcal{S}^{F}_{a\rightarrow\ell_2}$ across different learned GPT models ($c = 0$ or $c = 50$). Rows correspond to NT symbols $a$ and columns correspond to $\ell_2 = 2, 3, \dots, 7$. Colors represent the number of distinct elements in $\mathcal{S}^{\text{truth}}_{a\rightarrow\ell_2}$, and the white numbers represent the collision counts (if not present, meaning there are more than 5 collisions). More experiments in Figure 15, 16, and 17

---

**Observation.** We use $M = 20000$ samples. The diversity pattern from the pre-trained transformer matches that of the ground-truth. For instance, from the root one can generate $\Omega(M^2)$ distinct sequences to level $\ell_2 = 5$ using the CFG rules, and from every $a \in \mathbf{NT}_2$ one can generate $\Omega(M^2)$ to level $\ell_2 = 6$ (not to say to the T-level $\ell_2 = 7$); this is already more than the number of parameters in the model. Therefore, we conclude that the pre-trained model **does not rely on simply memorizing** a small set of patterns to learn the CFGs.

**Definition B.2.** *For symbol $a \in \mathbf{NT}_{\ell_1}$ and some layer $\ell_2 \in \{\ell_1, \ell_1 + 1, \dots, L\}$, define multi-set*[19]

$$\mathcal{S}_{a\rightarrow\ell_2}(x) = \left[\!\!\left[ \mathfrak{s}_{\ell_2}(i..j) \,\middle|\, \forall i, j, i \leq j \text{ such that } \mathfrak{p}_{\ell_1}(i-1) \neq \mathfrak{p}_{\ell_1}(i) = \mathfrak{p}_{\ell_1}(j) \neq \mathfrak{p}_{\ell_1}(j+1) \wedge a = \mathfrak{s}_{\ell_1}(i) \right]\!\!\right]$$

*and we define the multi-set union $\mathcal{S}_{a\rightarrow\ell_2} = \bigcup_{i\in[M]} \mathcal{S}_{a\rightarrow\ell_2}\big(x^{(i)}\big)$, which is* the multiset of all sentential forms that can be derived from NT symbol $a$ to depth $\ell_2$.

(Above, when $x \sim L(\mathcal{G})$ is generated from the ground-truth CFG, then the ancestor indices and symbols $\mathfrak{p}, \mathfrak{s}$ are defined in Section 2.1. If $x \in L(\mathcal{G})$ is an output from the transformer $F$, then we let $\mathfrak{p}, \mathfrak{s}$ be computed using dynamic programming, breaking ties lexicographically.)

We use $\mathcal{S}^{\text{truth}}_{a\rightarrow\ell_2}$ to denote the ground truth $\mathcal{S}_{a\rightarrow\ell_2}$ when $x^{(1)}, \dots, x^{(M)}$ are i.i.d. sampled from the real distribution $L(\mathcal{G})$, and denote by

$$\mathcal{S}^{F}_{a\rightarrow\ell_2} = \bigcup_{i\in[M'] \text{ and } x^{(i)}_{:c}, F(x^{(i)}_{:c})\in L(\mathcal{G})} \mathcal{S}_{a\rightarrow\ell_2}\big(x^{(i)}_{:c}, F(x^{(i)}_{:c})\big)$$

that from the transformer $F$. For a fair comparison, for each $F$ and $p$, we pick an $M' \geq M$ such that $M = \big|\{i \in [M'] \mid x^{(i)}_{:p}, F(x^{(i)}_{:p}) \in L(\mathcal{G})\}\big|$ so that $F$ is capable of generating exactly $M$ sentences that nearly-perfectly satisfy the CFG rules.[20]

Intuitively, for $x$'s generated by the transformer model, the larger the number of distinct sequences in $\mathcal{S}^{F}_{a\rightarrow\ell_2}$ is, the more diverse the set of NTs on level $\ell_2$ (or Ts if $\ell_2 = L$) the model can generate starting from NT $a$. Moreover, in the event that $\mathcal{S}^{F}_{a\rightarrow\ell_2}$ has only distinct sequences (so collision count = 0), then we know that the generation from $a \rightarrow \ell_2$, with good probability, should include at least $\Omega(M^2)$ possibilities using a birthday paradox argument. [21]

For such reason, it can be beneficial if we compare the *number of distinct sequences* and the *collision counts* between $\mathcal{S}^{F}_{a\rightarrow\ell_2}$ and $\mathcal{S}^{\text{truth}}_{a\rightarrow\ell_2}$. Note we consider all $\ell_2 \geq \ell_1$ instead of only $\ell_2 = L$, because we want to better capture model's diversity at all CFG levels.[22] We present our findings in Figure 14 with $M = 20000$ samples for the cfg3f dataset.

---

[19]Throughout this paper, we use $[\![\cdot]\!]$ to denote multi-sets that allow multiplicity, such as $[\![1, 2, 2, 3]\!]$. This allows us to conveniently talk about its collision count, number of distinct elements, and set average.

[20]Please note $M$ and $M'$ are roughly the same, given

[21]A CFG of depth $L$, even with constant degree and constant size, can generate $2^{2^{\Omega(L)}}$ distinct sequences.

[22]A model might generate a same NT symbol sequence $s_{L-1}$, and then generate different Ts randomly from each NT. In this way, the model still generates strings $x$'s with large diversity, but $\mathcal{S}^{F}_{a\rightarrow L-1}(x)$ is small. If $\mathcal{S}^{F}_{a\rightarrow\ell_2}$ is large for every $\ell_2$ and $a$, then the generation from the model is *truely diverse at any level of the CFG*.

In Figure 15 we present that for cfg3b, cfg3i, cfg3h, cfg3g, in Figure 16 for cfg3e1, and in Figure 17 for cfg3e2. We note that not only for hard, ambiguous datasets, also for those less ambiguous (cfg3e1, cfg3e2) datasets, language models are capable of generating very diverse outputs.

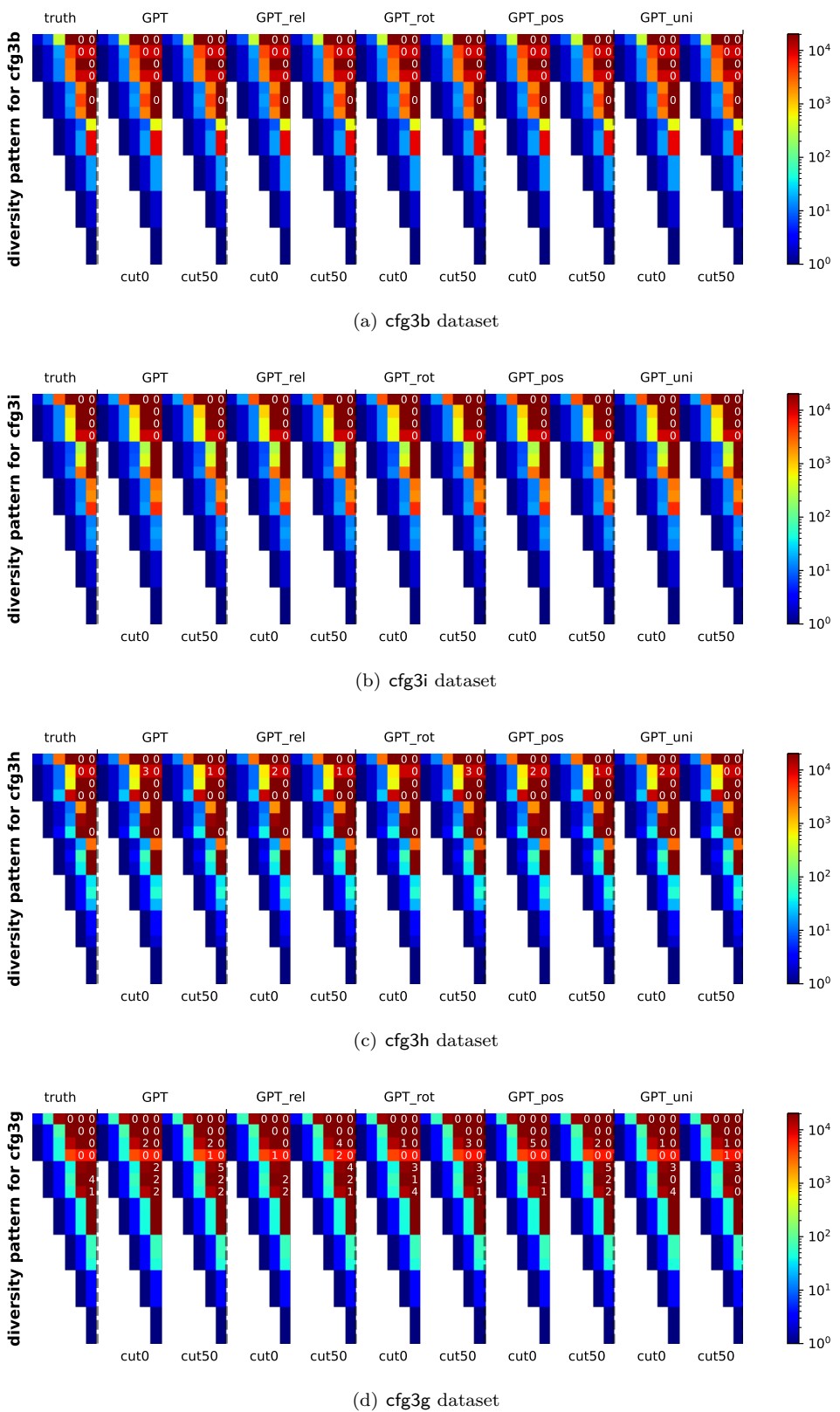

Figure 15: Comparing the generation diversity $\mathcal{S}^{\text{truth}}_{a\to\ell_2}$ and $\mathcal{S}^{F}_{a\to\ell_2}$ across different learned GPT models (and for $c = 0$ or $c = 50$). Rows correspond to NT symbols $a$ and columns correspond to $\ell_2 = 2, 3, \ldots, 7$. Colors represent the number of distinct elements in $\mathcal{S}^{\text{truth}}_{a\to\ell_2}$, and the white numbers represent the collision counts (if not present, meaning there are more than 5 collisions).

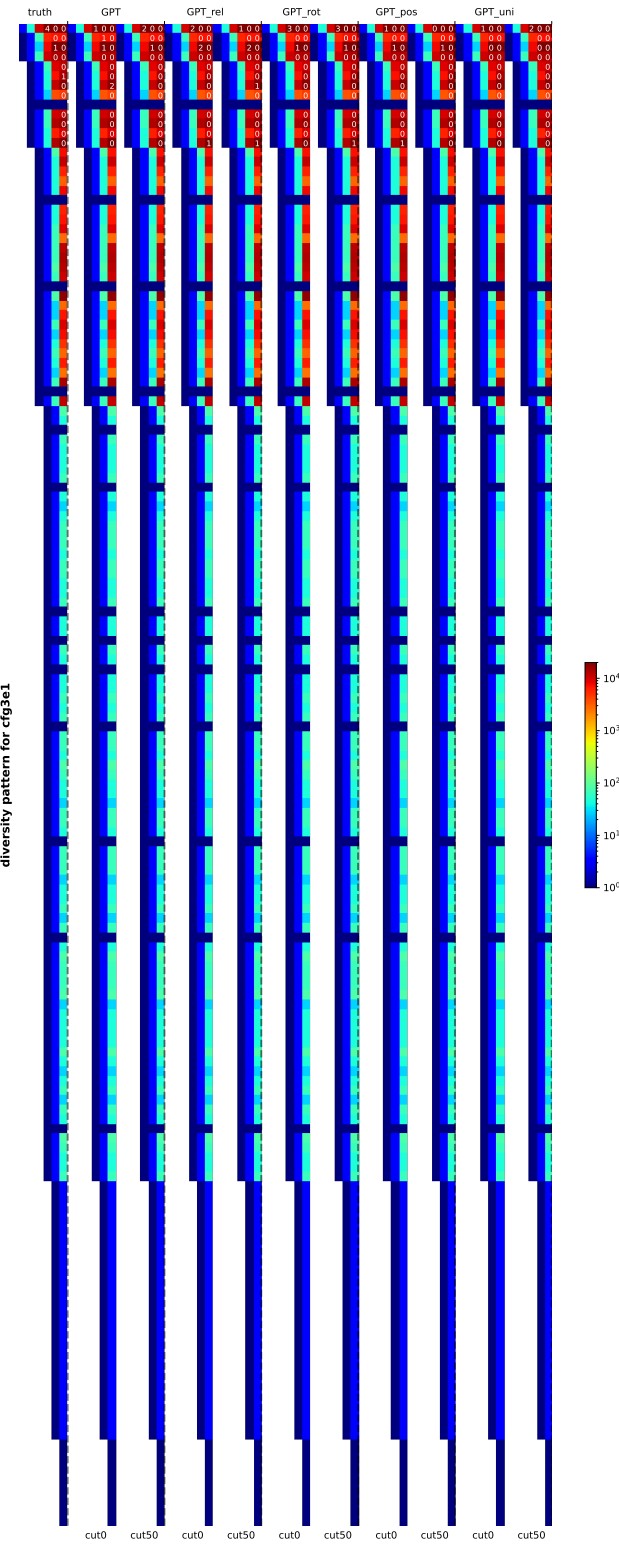

Figure 16: Comparing the generation diversity $\mathcal{S}_{a\to\ell_2}^{\mathrm{truth}}$ and $\mathcal{S}_{a\to\ell_2}^{F}$ across different learned GPT models (and for $c = 0$ or $c = 50$). Rows correspond to NT symbols $a$ and columns correspond to $\ell_2 = 2, 3, \ldots, 7$. Colors represent the number of distinct elements in $\mathcal{S}_{a\to\ell_2}^{\mathrm{truth}}$, and the white numbers represent the collision counts (if not present, meaning there are more than 5 collisions). This is for the cfg3e1 dataset.

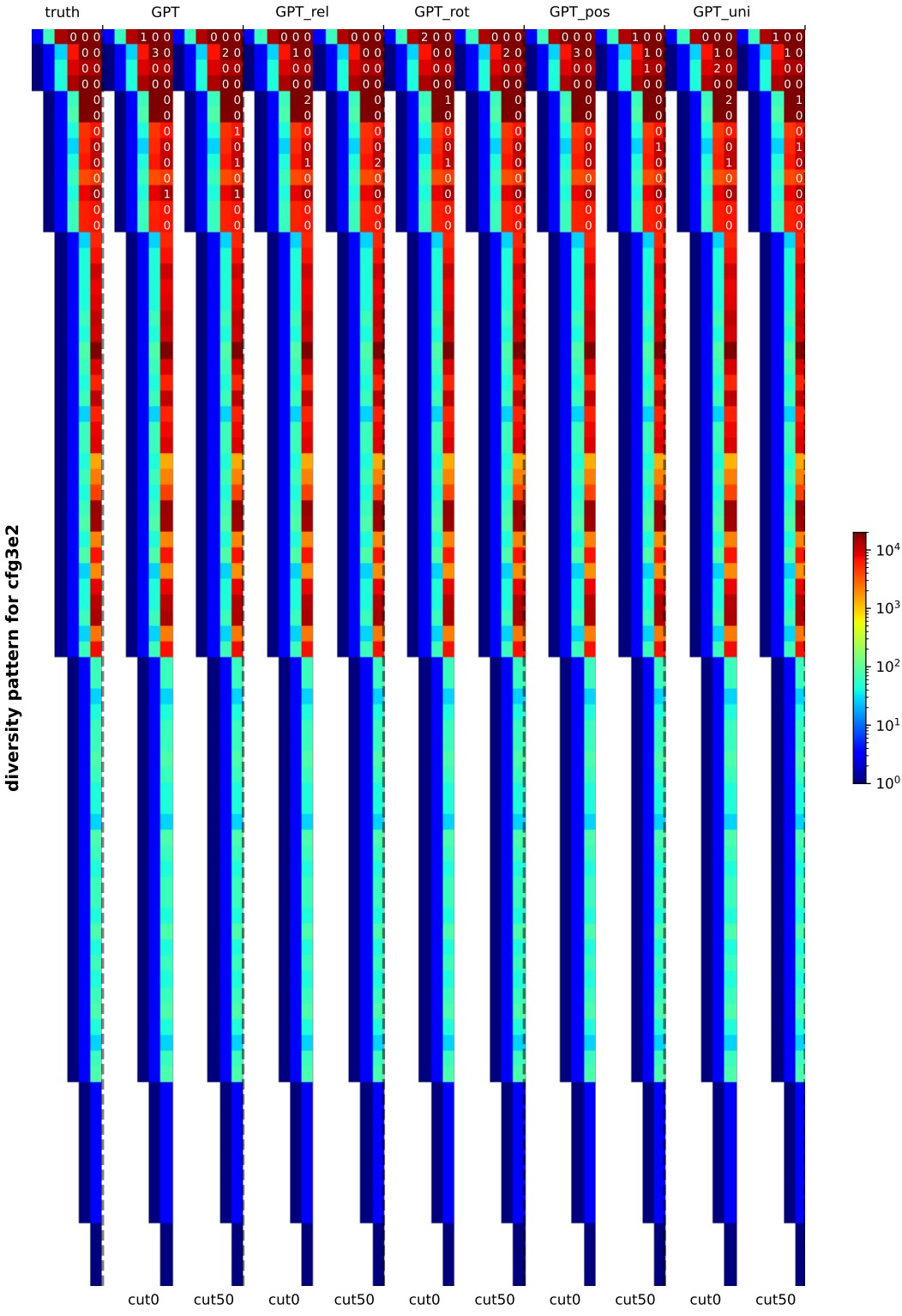

Figure 17: Comparing the generation diversity $\mathcal{S}^{\mathsf{truth}}_{a\to\ell_2}$ and $\mathcal{S}^{F}_{a\to\ell_2}$ across different learned GPT models (and for $c = 0$ or $c = 50$). Rows correspond to NT symbols $a$ and columns correspond to $\ell_2 = 2, 3, \ldots, 7$. Colors represent the number of distinct elements in $\mathcal{S}^{\mathsf{truth}}_{a\to\ell_2}$, and the white numbers represent the collision counts (if not present, meaning there are more than 5 collisions). This is for the cfg3e2 dataset.

## B.2 Marginal Distribution Comparison

In order to effectively learn a CFG, it is also important to match the distribution of generating probabilities. While measuring this can be challenging, we have conducted at least a simple test on the marginal distributions $p(a, i)$, which represent the probability of symbol $a \in \mathbf{NT}_\ell$ appearing at position $i$ (i.e., the probability that $\mathfrak{s}_\ell(i) = a$). We observe a strong alignment between the generated probabilities and the ground-truth distribution. See Figure 18.

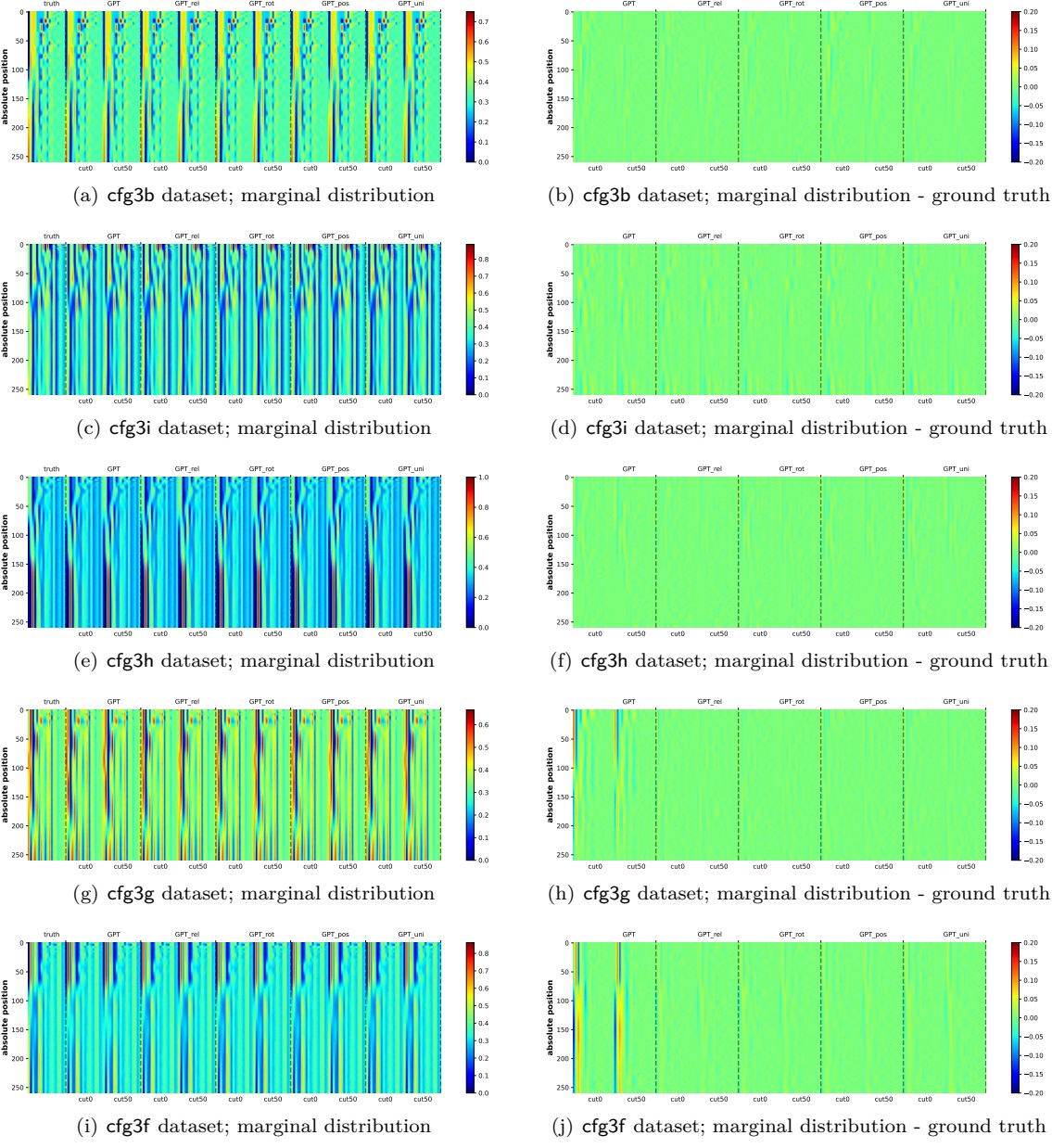

(a) cfg3b dataset; marginal distribution

(b) cfg3b dataset; marginal distribution - ground truth

(c) cfg3i dataset; marginal distribution

(d) cfg3i dataset; marginal distribution - ground truth

(e) cfg3h dataset; marginal distribution

(f) cfg3h dataset; marginal distribution - ground truth

(g) cfg3g dataset; marginal distribution

(h) cfg3g dataset; marginal distribution - ground truth

(i) cfg3f dataset; marginal distribution

(j) cfg3f dataset; marginal distribution - ground truth

Figure 18: Marginal distribution $p(a, i)$ difference between a trained model and the ground-truth, for an NT/T symbol $a$ (column) at position $i$ (row). Figures on the left compare the marginal distribution of the ground-truth against those generated from 5 models × 2 cut positions ($c = 0/c = 50$). Figures on the right showcase the marginal distribution *difference* between them and the ground-truth. It is noticeable from the figures that GPT did not learn cfg3g and cfg3f well. This is consistent with the generation accuracies in Figure 4.

## C  More Experiments on Results 4-5 (NT Ancestor and Boundary Probing)

### C.1  NT Ancestor and NT Boundary Probing

Earlier, as confirmed in Figure 5, we established that the hidden states (of the final transformer layer) have implicitly encoded the NT ancestor symbols $\mathfrak{s}_\ell(i)$ for each CFG level $\ell$ and token position $i$ using a linear transformation. In Figure 19(a) in this section, we also demonstrate that the same conclusion applies to the NT-end boundary $\mathfrak{b}_\ell(i)$. This completes Result 4.

More importantly, for $\mathfrak{b}_\ell(i)$, we also show that this information is *stored locally*, very close to position $i$ (such as at $i \pm 1$). Details can be found in Figure 19. In particular, note as shown in Figure 7, we confirmed that at any NT boundary position $i$ where $\mathfrak{b}_\ell(i) = 1$, the transformer has also locally encoded clear information about the NT ancestor symbol $\mathfrak{s}_\ell(i)$, either exactly at $i$ or at $i \pm 1$. To be precise, this is a conditional statement — given that it is an NT boundary, NT ancestors can be predicted. Therefore, in principle, one must also verify that the prediction task for the NT boundary is successful to begin with. Such missing experiments are, in fact, included in Figure 19(b) and Figure 19(c).

**predict NT-end boundary (%)**

| | GPT | | | | | GPT_rel | | | | | GPT_rot | | | | | GPT_pos | | | | | GPT_uni | | | | | baseline (GPT_rand) | | | | |
|---|---|---|---|---|---|---|---|---|---|---|---|---|---|---|---|---|---|---|---|---|---|---|---|---|---|---|---|---|---|---|
| cfg3b | 100 | 100 | 100 | 100 | 100 | 100 | 100 | 100 | 100 | 100 | 100 | 100 | 100 | 100 | 100 | 100 | 100 | 100 | 100 | 100 | 100 | 100 | 100 | 100 | 100 | 96.5 | 88.0 | 95.5 | 98.5 | 99.6 |
| cfg3i | 99.7 | 99.8 | 99.0 | 99.5 | 99.9 | 99.7 | 99.8 | 99.1 | 99.5 | 99.9 | 99.7 | 99.8 | 99.1 | 99.5 | 99.9 | 99.8 | 99.8 | 99.1 | 99.6 | 99.9 | 99.8 | 99.8 | 99.1 | 99.6 | 99.9 | 87.5 | 88.6 | 94.9 | 97.9 | 99.3 |
| cfg3h | 99.7 | 99.3 | 99.5 | 99.8 | 99.9 | 99.7 | 99.4 | 99.5 | 99.8 | 99.9 | 99.7 | 99.4 | 99.5 | 99.8 | 99.9 | 99.7 | 99.4 | 99.6 | 99.9 | 100 | 99.7 | 99.4 | 99.6 | 99.9 | 100 | 88.1 | 86.8 | 94.0 | 97.9 | 99.4 |
| cfg3g | 99.8 | 98.0 | 98.2 | 99.2 | 99.7 | 99.8 | 98.3 | 98.5 | 99.4 | 99.8 | 99.8 | 98.2 | 98.5 | 99.4 | 99.8 | 99.7 | 98.3 | 98.6 | 99.4 | 99.8 | 99.8 | 98.3 | 98.6 | 99.4 | 99.8 | 92.1 | 85.6 | 93.6 | 97.7 | 99.3 |
| cfg3f | 100 | 98.3 | 98.8 | 99.3 | 99.7 | 100 | 98.8 | 99.0 | 99.5 | 99.8 | 100 | 98.8 | 99.1 | 99.5 | 99.8 | 100 | 98.9 | 99.2 | 99.6 | 99.8 | 100 | 98.8 | 99.1 | 99.5 | 99.8 | 91.7 | 85.6 | 94.8 | 98.1 | 99.4 |
| cfg3e1 | 100 | 100 | 100 | 100 | 100 | 100 | 100 | 100 | 100 | 100 | 100 | 100 | 100 | 100 | 100 | 100 | 100 | 100 | 100 | 100 | 100 | 100 | 100 | 100 | 100 | 71.7 | 84.2 | 94.0 | 97.8 | 99.3 |
| cfg3e2 | 99.5 | 99.9 | 100 | 100 | 100 | 99.6 | 100 | 100 | 100 | 100 | 99.6 | 100 | 100 | 100 | 100 | 99.7 | 100 | 100 | 100 | 100 | 99.7 | 100 | 100 | 100 | 100 | 73.1 | 84.6 | 94.2 | 98.0 | 99.3 |
| | NT6 | NT5 | NT4 | NT3 | NT2 | NT6 | NT5 | NT4 | NT3 | NT2 | NT6 | NT5 | NT4 | NT3 | NT2 | NT6 | NT5 | NT4 | NT3 | NT2 | NT6 | NT5 | NT4 | NT3 | NT2 | NT6 | NT5 | NT4 | NT3 | NT2 |

(a) Predicting NT boundaries: the column $NT_\ell$ for $\ell = 2, 3, 4, 5, 6$ represents the accuracy of predicting $\mathfrak{b}_\ell$ using the multi-head linear probing function described in (4.2).

**predict NT-end boundary (%) (diagonal masking)**

| | GPT | | | | | GPT_rel | | | | | GPT_rot | | | | | GPT_pos | | | | | GPT_uni | | | | | baseline (GPT_rand) | | | | |
|---|---|---|---|---|---|---|---|---|---|---|---|---|---|---|---|---|---|---|---|---|---|---|---|---|---|---|---|---|---|---|
| cfg3b | 95.7 | 100 | 99.6 | 99.5 | 99.9 | 95.8 | 100 | 99.6 | 99.5 | 99.9 | 95.8 | 100 | 99.6 | 99.5 | 99.9 | 95.7 | 100 | 99.6 | 99.5 | 99.9 | 95.8 | 100 | 99.6 | 99.5 | 99.9 | 96.5 | 88.0 | 95.5 | 98.5 | 99.6 |
| cfg3i | 96.5 | 96.9 | 97.7 | 98.5 | 99.4 | 96.6 | 97.1 | 97.8 | 98.5 | 99.4 | 96.6 | 97.0 | 97.8 | 98.5 | 99.4 | 96.5 | 97.0 | 97.7 | 98.5 | 99.4 | 96.6 | 97.1 | 97.8 | 98.5 | 99.4 | 87.5 | 88.6 | 94.9 | 97.9 | 99.3 |
| cfg3h | 91.3 | 95.0 | 97.8 | 99.1 | 99.6 | 91.5 | 95.2 | 97.9 | 99.1 | 99.6 | 91.5 | 95.2 | 97.9 | 99.1 | 99.6 | 91.5 | 95.2 | 97.9 | 99.1 | 99.6 | 91.5 | 95.2 | 97.9 | 99.1 | 99.6 | 88.1 | 86.8 | 94.0 | 97.9 | 99.4 |
| cfg3g | 86.7 | 92.6 | 95.0 | 98.0 | 99.1 | 86.9 | 92.8 | 95.2 | 98.1 | 99.2 | 86.9 | 92.8 | 95.3 | 98.1 | 99.2 | 86.9 | 92.8 | 95.2 | 98.1 | 99.2 | 86.9 | 92.8 | 95.2 | 98.1 | 99.2 | 92.1 | 85.6 | 93.6 | 97.7 | 99.3 |
| cfg3f | 89.1 | 92.7 | 96.5 | 98.2 | 99.2 | 89.4 | 93.2 | 96.7 | 98.4 | 99.3 | 89.4 | 93.2 | 96.7 | 98.4 | 99.3 | 89.3 | 93.2 | 96.6 | 98.3 | 99.2 | 89.3 | 93.2 | 96.6 | 98.3 | 99.2 | 91.7 | 85.6 | 94.8 | 98.1 | 99.4 |
| cfg3e1 | 98.2 | 99.6 | 99.9 | 99.9 | 99.8 | 98.2 | 99.6 | 99.9 | 99.9 | 99.8 | 98.2 | 99.6 | 99.9 | 99.9 | 99.8 | 98.2 | 99.6 | 99.9 | 99.9 | 99.8 | 98.2 | 99.6 | 99.9 | 99.9 | 99.8 | 71.7 | 84.2 | 94.0 | 97.8 | 99.3 |
| cfg3e2 | 96.0 | 99.0 | 99.9 | 100 | 100 | 96.1 | 99.0 | 99.9 | 100 | 100 | 96.0 | 99.0 | 99.9 | 100 | 100 | 96.0 | 99.0 | 99.9 | 100 | 100 | 96.1 | 99.0 | 99.9 | 100 | 100 | 73.1 | 84.6 | 94.2 | 98.0 | 99.3 |
| | NT6 | NT5 | NT4 | NT3 | NT2 | NT6 | NT5 | NT4 | NT3 | NT2 | NT6 | NT5 | NT4 | NT3 | NT2 | NT6 | NT5 | NT4 | NT3 | NT2 | NT6 | NT5 | NT4 | NT3 | NT2 | NT6 | NT5 | NT4 | NT3 | NT2 |

(b) Predicting NT boundaries with diagonal masking: the column $NT_\ell$ for $\ell = 2, 3, 4, 5, 6$ represents the accuracy of predicting $\mathfrak{b}_\ell$ using (4.2) but setting $w_{r,i \to k} = 0$ for $i \neq k$.

**predict NT-end boundary (%) (tridiagonal masking)**

| | GPT | | | | | GPT_rel | | | | | GPT_rot | | | | | GPT_pos | | | | | GPT_uni | | | | | baseline (GPT_rand) | | | | |
|---|---|---|---|---|---|---|---|---|---|---|---|---|---|---|---|---|---|---|---|---|---|---|---|---|---|---|---|---|---|---|
| cfg3b | 99.9 | 100 | 99.6 | 99.6 | 99.9 | 99.9 | 100 | 99.6 | 99.6 | 99.9 | 99.9 | 100 | 99.6 | 99.6 | 99.9 | 99.9 | 100 | 99.6 | 99.6 | 99.9 | 99.9 | 100 | 99.6 | 99.6 | 99.9 | 96.5 | 88.0 | 95.5 | 98.5 | 99.6 |
| cfg3i | 97.7 | 98.2 | 98.3 | 98.9 | 99.6 | 97.8 | 98.2 | 98.4 | 98.9 | 99.6 | 97.7 | 98.2 | 98.4 | 98.9 | 99.6 | 97.8 | 98.2 | 98.4 | 98.9 | 99.6 | 97.8 | 98.2 | 98.4 | 98.9 | 99.6 | 87.5 | 88.6 | 94.9 | 97.9 | 99.3 |
| cfg3h | 98.0 | 97.2 | 98.7 | 99.4 | 99.8 | 98.1 | 97.3 | 98.8 | 99.4 | 99.8 | 98.1 | 97.3 | 98.8 | 99.4 | 99.8 | 98.1 | 97.4 | 98.7 | 99.4 | 99.8 | 98.1 | 97.4 | 98.7 | 99.4 | 99.8 | 88.1 | 86.8 | 94.0 | 97.9 | 99.4 |
| cfg3g | 96.7 | 96.3 | 96.5 | 98.7 | 99.5 | 96.7 | 96.5 | 96.8 | 98.8 | 99.6 | 96.7 | 96.5 | 96.8 | 98.8 | 99.6 | 96.7 | 96.5 | 96.8 | 98.8 | 99.6 | 96.7 | 96.5 | 96.7 | 98.8 | 99.6 | 92.1 | 85.6 | 93.6 | 97.7 | 99.3 |
| cfg3f | 98.3 | 95.4 | 97.4 | 98.7 | 99.6 | 98.4 | 95.7 | 97.6 | 98.9 | 99.6 | 98.4 | 95.7 | 97.6 | 98.9 | 99.6 | 98.4 | 95.7 | 97.6 | 98.8 | 99.6 | 98.4 | 95.7 | 97.6 | 98.8 | 99.6 | 91.7 | 85.6 | 94.8 | 98.1 | 99.4 |
| cfg3e1 | 99.9 | 100 | 100 | 100 | 99.9 | 99.9 | 100 | 100 | 100 | 99.9 | 99.9 | 100 | 100 | 100 | 99.9 | 99.9 | 100 | 100 | 100 | 99.9 | 99.9 | 100 | 100 | 100 | 99.9 | 71.7 | 84.2 | 94.0 | 97.8 | 99.3 |
| cfg3e2 | 98.7 | 99.7 | 100 | 100 | 100 | 98.8 | 99.7 | 100 | 100 | 100 | 98.8 | 99.7 | 100 | 100 | 100 | 98.8 | 99.7 | 100 | 100 | 100 | 98.9 | 99.7 | 100 | 100 | 100 | 73.1 | 84.6 | 94.2 | 98.0 | 99.3 |
| | NT6 | NT5 | NT4 | NT3 | NT2 | NT6 | NT5 | NT4 | NT3 | NT2 | NT6 | NT5 | NT4 | NT3 | NT2 | NT6 | NT5 | NT4 | NT3 | NT2 | NT6 | NT5 | NT4 | NT3 | NT2 | NT6 | NT5 | NT4 | NT3 | NT2 |

(c) Predicting NT boundaries with tridiagonal masking: the column $NT_\ell$ for $\ell = 2, 3, 4, 5, 6$ represents the accuracy of predicting $\mathfrak{b}_\ell$ using (4.2) but setting $w_{r,i \to k} = 0$ for $|i - k| > 1$.

Figure 19: After pre-training, the NT-end boundary information — i.e., $\mathfrak{b}_\ell(i)$ for position $i$ and NT level $\ell$ — is largely stored *locally* near the hidden state at position $i \pm 1$, up to a linear transformation. This can be compared with the prediction accuracy of the NT ancestor $\mathfrak{s}_\ell(i)$ in Figure 5.

---

**Observation.** This implies, the transformer actually *knows*, with a very good accuracy, that "position $i$ is already the end of NT on level $\ell$", by just reading all the texts until this position (possibly peeking one more to its right).

**Remark 1.** It may be mathematically necessary to peek more than 1 tokens to decide if a position $i$ is at an NT boundary, due to CFG's ambiguity. But, in most cases, that can be decided quite early.

**Remark 2.** Predicting NT boundary is a very *biased* binary classification task. For levels $\ell$ that are close to the CFG root, most symbols are not at NT boundary for that level $\ell$ (see Figure 2). For such reason, in the *heatmap color* of the figures above, we have *normalized* the columns with respect to NT2..NT6 differently, to reflect this bias.

## C.2 NT Probing Across Transformer's Layers

As one may image, the NT ancestor and boundary information for smaller CFG levels $\ell$ (i.e., closer to CFG root) are only learned at those deeper transformer layers $l$. In Figure 20, we present this finding by calculating the *linear* encoding accuracies with respect to all the 12 transformer layers in GPT and GPT$_{\mathsf{rel}}$. We confirm that generative models discover such information *hierarchically*.

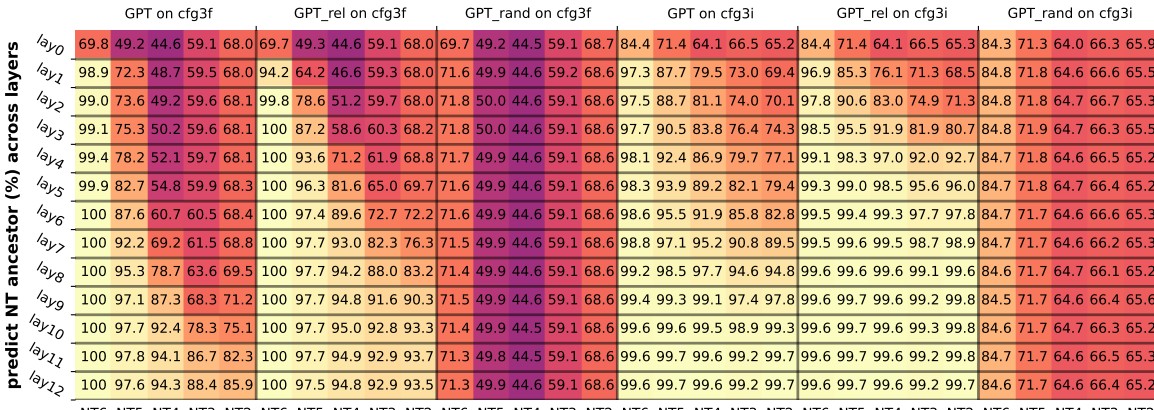

(a) Predict NT ancestors, comparing against the GPT$_{\mathsf{rand}}$ baseline

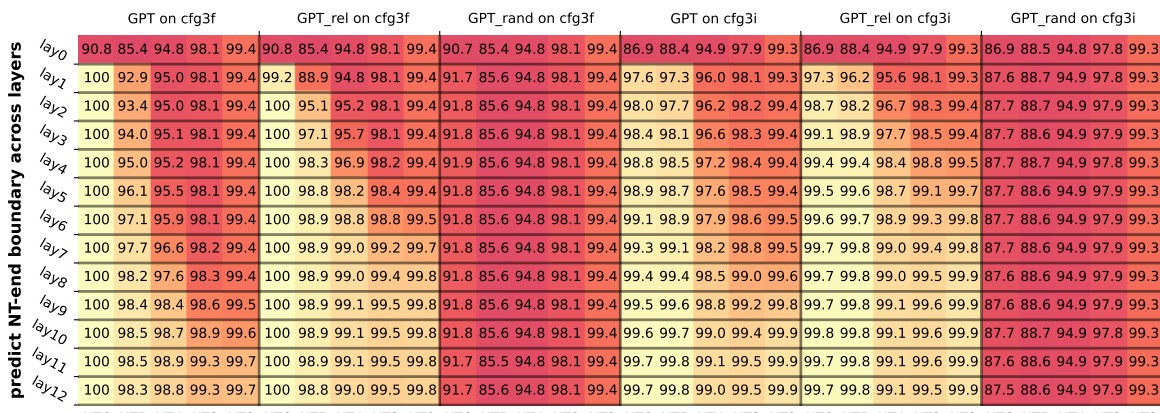

(b) Predict NT boundaries, comparing against the GPT$_{\mathsf{rand}}$ baseline

Figure 20: Generative models discover NT ancestors and NT boundaries hierarchically.

### C.3 NT Predictions Across Training Epochs

Moreover, one may conjecture that the NT ancestor and NT boundary information is learned *gradually* as the number of training steps increase. We have confirmed this in Figure 21. We emphasize that this does not imply layer-wise training is applicable in learning deep CFGs. It is crucial to train all the layers together, as the training process of deeper transformer layers may help backward correct the features learned in the lower layers, through a process called "backward feature correction" (Allen-Zhu and Li, 2023).

predict NT ancestor/boundary (%) across training epochs

| epochs | predict NT (GPT) | | | | | predict NTend (GPT) | | | | | predict NT (GPT_rel) | | | | | predict NTend (GPT_rel) | | | | |
|---|---|---|---|---|---|---|---|---|---|---|---|---|---|---|---|---|---|---|---|---|
| | NT6 | NT5 | NT4 | NT3 | NT2 | NT6 | NT5 | NT4 | NT3 | NT2 | NT6 | NT5 | NT4 | NT3 | NT2 | NT6 | NT5 | NT4 | NT3 | NT2 |
| 5 | 99.5 | 84.2 | 57.2 | 59.9 | 68.7 | 100 | 96.4 | 95.6 | 98.1 | 99.4 | 100 | 96.2 | 86.8 | 68.8 | 70.9 | 100 | 98.5 | 98.5 | 98.7 | 99.5 |
| 10 | 100 | 93.2 | 71.6 | 62.0 | 69.1 | 100 | 98.0 | 97.2 | 98.2 | 99.4 | 100 | 96.8 | 91.7 | 79.7 | 75.5 | 100 | 98.6 | 98.8 | 99.1 | 99.6 |
| 15 | 100 | 95.2 | 79.7 | 64.5 | 69.9 | 100 | 98.2 | 97.9 | 98.4 | 99.4 | 100 | 97.0 | 92.7 | 85.3 | 80.0 | 100 | 98.6 | 98.8 | 99.3 | 99.7 |
| 20 | 100 | 96.1 | 83.4 | 66.1 | 70.3 | 100 | 98.4 | 98.3 | 98.5 | 99.4 | 100 | 97.1 | 93.2 | 87.5 | 83.4 | 100 | 98.7 | 98.9 | 99.4 | 99.7 |
| 25 | 100 | 96.5 | 86.0 | 68.7 | 71.1 | 100 | 98.4 | 98.4 | 98.6 | 99.5 | 100 | 97.2 | 93.6 | 88.9 | 86.0 | 100 | 98.7 | 98.9 | 99.4 | 99.8 |
| 30 | 100 | 96.8 | 87.5 | 70.5 | 71.7 | 100 | 98.4 | 98.5 | 98.7 | 99.5 | 100 | 97.2 | 93.7 | 89.7 | 87.8 | 100 | 98.7 | 98.9 | 99.4 | 99.8 |
| 35 | 100 | 97.0 | 88.5 | 71.9 | 72.6 | 100 | 98.4 | 98.5 | 98.8 | 99.5 | 100 | 97.4 | 94.1 | 90.6 | 89.3 | 100 | 98.7 | 98.9 | 99.4 | 99.8 |
| 40 | 100 | 97.1 | 89.4 | 73.3 | 73.1 | 100 | 98.5 | 98.6 | 98.8 | 99.5 | 100 | 97.3 | 94.0 | 90.8 | 90.1 | 100 | 98.7 | 98.9 | 99.4 | 99.8 |
| 45 | 100 | 97.1 | 90.1 | 74.7 | 73.9 | 100 | 98.4 | 98.6 | 98.9 | 99.5 | 100 | 97.4 | 94.0 | 91.1 | 91.0 | 100 | 98.7 | 98.9 | 99.4 | 99.8 |
| 50 | 100 | 97.2 | 90.6 | 76.3 | 74.4 | 100 | 98.5 | 98.6 | 98.9 | 99.6 | 100 | 97.4 | 94.1 | 91.3 | 91.4 | 100 | 98.7 | 98.9 | 99.4 | 99.8 |
| 55 | 100 | 97.3 | 91.0 | 77.6 | 75.0 | 100 | 98.4 | 98.7 | 99.0 | 99.6 | 100 | 97.4 | 94.2 | 91.5 | 91.7 | 100 | 98.7 | 99.0 | 99.5 | 99.8 |
| 60 | 100 | 97.2 | 91.4 | 78.8 | 76.0 | 100 | 98.4 | 98.7 | 99.0 | 99.6 | 100 | 97.3 | 94.3 | 91.6 | 91.8 | 100 | 98.8 | 99.0 | 99.5 | 99.8 |
| 65 | 100 | 97.3 | 91.8 | 79.8 | 76.9 | 100 | 98.4 | 98.7 | 99.0 | 99.6 | 100 | 97.4 | 94.3 | 91.7 | 92.0 | 100 | 98.7 | 99.0 | 99.5 | 99.8 |
| 70 | 100 | 97.4 | 92.1 | 80.5 | 77.2 | 100 | 98.4 | 98.7 | 99.0 | 99.6 | 100 | 97.5 | 94.4 | 91.7 | 92.3 | 100 | 98.8 | 99.0 | 99.5 | 99.8 |
| 75 | 100 | 97.4 | 92.4 | 81.2 | 77.9 | 100 | 98.4 | 98.7 | 99.1 | 99.6 | 100 | 97.4 | 94.3 | 91.8 | 92.5 | 100 | 98.8 | 99.0 | 99.5 | 99.8 |
| 80 | 100 | 97.5 | 92.7 | 82.2 | 78.5 | 100 | 98.4 | 98.7 | 99.1 | 99.6 | 100 | 97.5 | 94.4 | 91.9 | 92.5 | 100 | 98.8 | 99.0 | 99.5 | 99.8 |
| 85 | 100 | 97.3 | 92.7 | 82.6 | 79.1 | 100 | 98.3 | 98.7 | 99.1 | 99.6 | 100 | 97.5 | 94.5 | 92.1 | 92.5 | 100 | 98.8 | 99.0 | 99.5 | 99.8 |
| 90 | 100 | 97.5 | 92.9 | 83.3 | 79.3 | 100 | 98.4 | 98.7 | 99.1 | 99.7 | 100 | 97.5 | 94.5 | 92.1 | 92.5 | 100 | 98.8 | 99.0 | 99.5 | 99.8 |
| 95 | 100 | 97.5 | 93.0 | 83.9 | 80.3 | 100 | 98.4 | 98.7 | 99.1 | 99.7 | 100 | 97.4 | 94.4 | 92.2 | 93.0 | 100 | 98.7 | 99.0 | 99.5 | 99.8 |
| 100 | 100 | 97.5 | 93.3 | 84.4 | 80.5 | 100 | 98.4 | 98.7 | 99.2 | 99.7 | 100 | 97.5 | 94.5 | 92.3 | 93.0 | 100 | 98.8 | 99.0 | 99.5 | 99.8 |
| 105 | 100 | 97.5 | 93.3 | 84.7 | 80.8 | 100 | 98.4 | 98.8 | 99.2 | 99.7 | 100 | 97.5 | 94.5 | 92.3 | 93.0 | 100 | 98.8 | 99.0 | 99.5 | 99.8 |
| 110 | 100 | 97.5 | 93.3 | 85.0 | 81.6 | 100 | 98.3 | 98.7 | 99.2 | 99.7 | 100 | 97.5 | 94.5 | 92.2 | 92.9 | 100 | 98.7 | 99.0 | 99.5 | 99.8 |
| 115 | 100 | 97.5 | 93.4 | 85.3 | 81.5 | 100 | 98.4 | 98.8 | 99.2 | 99.7 | 100 | 97.4 | 94.4 | 92.2 | 92.8 | 100 | 98.8 | 99.0 | 99.5 | 99.8 |
| 120 | 100 | 97.6 | 93.5 | 85.6 | 82.4 | 100 | 98.4 | 98.8 | 99.2 | 99.7 | 100 | 97.5 | 94.5 | 92.2 | 92.9 | 100 | 98.8 | 99.0 | 99.5 | 99.8 |
| 125 | 100 | 97.6 | 93.8 | 86.2 | 82.8 | 100 | 98.4 | 98.8 | 99.2 | 99.7 | 100 | 97.6 | 94.8 | 92.6 | 93.3 | 100 | 98.8 | 99.0 | 99.5 | 99.8 |
| 130 | 100 | 97.5 | 93.7 | 86.4 | 83.1 | 100 | 98.4 | 98.7 | 99.2 | 99.7 | 100 | 97.4 | 94.6 | 92.6 | 93.1 | 100 | 98.7 | 99.0 | 99.5 | 99.8 |
| 135 | 100 | 97.6 | 93.8 | 86.7 | 83.3 | 100 | 98.4 | 98.8 | 99.2 | 99.7 | 100 | 97.5 | 94.7 | 92.4 | 93.1 | 100 | 98.7 | 99.0 | 99.5 | 99.8 |
| 140 | 100 | 97.5 | 93.6 | 86.5 | 83.6 | 100 | 98.3 | 98.8 | 99.2 | 99.7 | 100 | 97.5 | 94.6 | 92.6 | 93.3 | 100 | 98.7 | 99.0 | 99.5 | 99.8 |
| 145 | 100 | 97.6 | 93.8 | 86.7 | 83.5 | 100 | 98.4 | 98.8 | 99.2 | 99.7 | 100 | 97.5 | 94.7 | 92.9 | 93.4 | 100 | 98.7 | 99.0 | 99.5 | 99.8 |
| 150 | 100 | 97.6 | 93.8 | 87.0 | 83.8 | 100 | 98.4 | 98.8 | 99.2 | 99.7 | 100 | 97.5 | 94.7 | 92.7 | 93.4 | 100 | 98.8 | 99.0 | 99.5 | 99.8 |
| 155 | 100 | 97.6 | 93.9 | 87.1 | 84.7 | 100 | 98.4 | 98.8 | 99.2 | 99.7 | 100 | 97.5 | 94.6 | 92.5 | 93.0 | 100 | 98.8 | 99.0 | 99.5 | 99.8 |
| 160 | 100 | 97.6 | 94.0 | 87.1 | 84.5 | 100 | 98.4 | 98.8 | 99.3 | 99.7 | 100 | 97.6 | 94.7 | 92.5 | 93.0 | 100 | 98.8 | 99.0 | 99.5 | 99.8 |
| 165 | 100 | 97.6 | 94.0 | 87.8 | 85.0 | 100 | 98.4 | 98.8 | 99.3 | 99.7 | 100 | 97.5 | 94.6 | 92.7 | 93.3 | 100 | 98.8 | 99.0 | 99.5 | 99.8 |
| 170 | 100 | 97.5 | 94.1 | 87.8 | 85.3 | 100 | 98.4 | 98.8 | 99.3 | 99.7 | 100 | 97.4 | 94.7 | 92.8 | 93.5 | 100 | 98.7 | 99.0 | 99.5 | 99.8 |
| 175 | 100 | 97.6 | 94.1 | 87.9 | 85.4 | 100 | 98.4 | 98.8 | 99.3 | 99.7 | 100 | 97.5 | 94.7 | 92.6 | 93.2 | 100 | 98.8 | 99.0 | 99.5 | 99.8 |
| 180 | 100 | 97.6 | 94.1 | 87.9 | 85.3 | 100 | 98.4 | 98.8 | 99.3 | 99.7 | 100 | 97.6 | 94.7 | 92.5 | 93.2 | 100 | 98.8 | 99.0 | 99.5 | 99.8 |
| 185 | 100 | 97.6 | 94.2 | 88.1 | 85.5 | 100 | 98.3 | 98.8 | 99.3 | 99.7 | 100 | 97.5 | 94.7 | 92.7 | 93.4 | 100 | 98.8 | 99.0 | 99.5 | 99.8 |
| 190 | 100 | 97.6 | 94.3 | 88.2 | 85.6 | 100 | 98.4 | 98.8 | 99.3 | 99.7 | 100 | 97.5 | 94.8 | 92.8 | 93.6 | 100 | 98.8 | 99.0 | 99.5 | 99.8 |
| 195 | 100 | 97.6 | 94.2 | 88.3 | 86.0 | 100 | 98.4 | 98.8 | 99.3 | 99.7 | 100 | 97.5 | 94.8 | 92.8 | 93.5 | 100 | 98.8 | 99.0 | 99.5 | 99.8 |
| 200 | 100 | 97.7 | 94.2 | 88.2 | 85.7 | 100 | 98.4 | 98.8 | 99.3 | 99.7 | 100 | 97.5 | 94.7 | 92.7 | 93.3 | 100 | 98.8 | 99.0 | 99.5 | 99.8 |

Figure 21: Generative models discover NT ancestors and NT boundaries gradually across training epochs (here 1 epoch equals 500 training steps). CFG levels closer to the leaves are learned faster, and their accuracies continue to increase as deeper levels are being learned, following a principle called "backward feature correction" in deep hierarchical learning (Allen-Zhu and Li, 2023).

# D More Experiments on Results 6-9 (Attention Patterns)

## D.1 Result 6: Position-Based Attention Pattern

Recall from Figure 8 we have shown that the attention weights between any two positions $j \to i$ have a strong bias in the relative difference $p = |j - i|$. Different heads or layers have different dependencies on $p$. Below in Figure 22, we give experiments for this phenomenon in more datasets and for both GPT/GPT$_{rel}$.

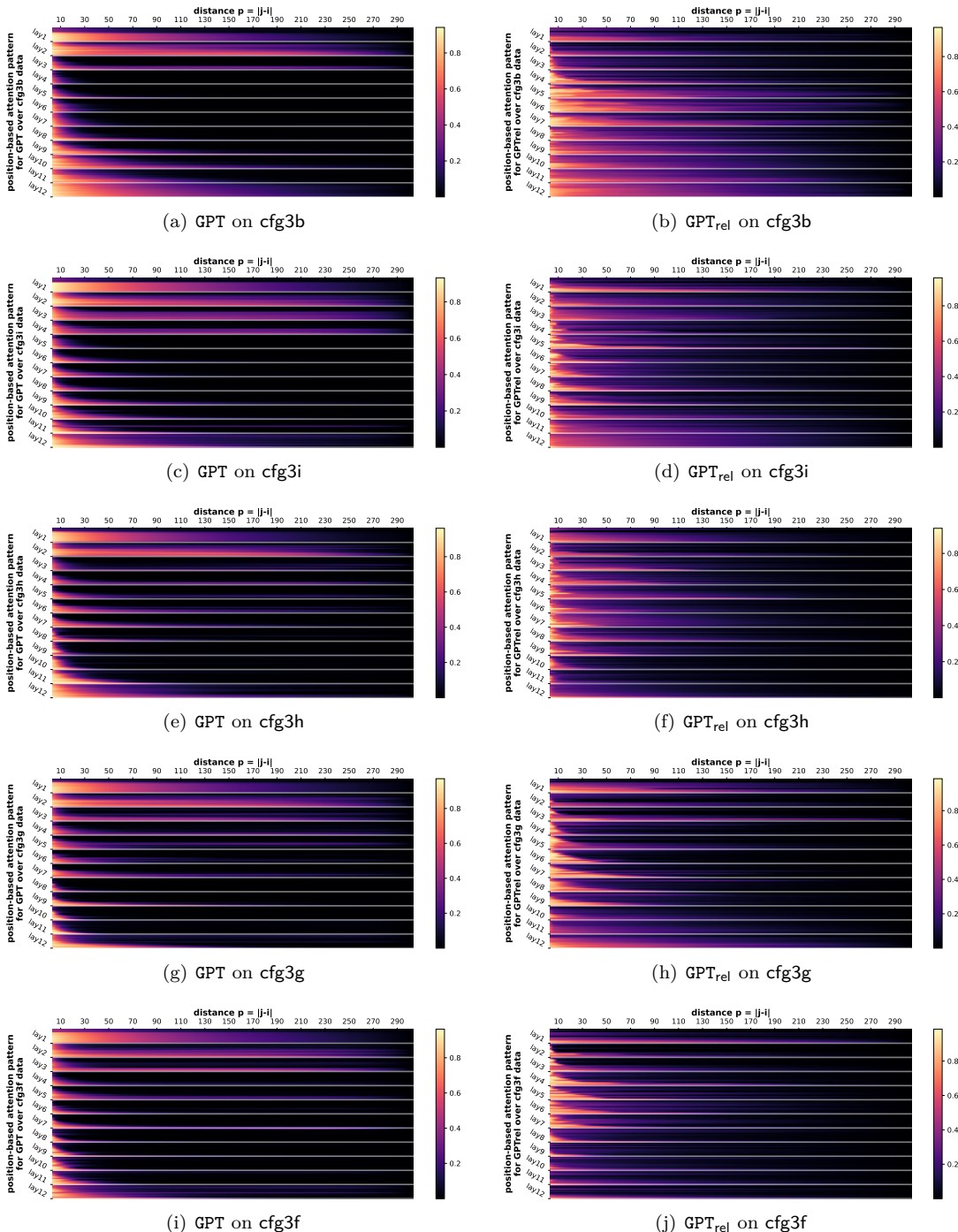

(a) GPT on cfg3b                     (b) GPT$_{rel}$ on cfg3b

(c) GPT on cfg3i                     (d) GPT$_{rel}$ on cfg3i

(e) GPT on cfg3h                     (f) GPT$_{rel}$ on cfg3h

(g) GPT on cfg3g                     (h) GPT$_{rel}$ on cfg3g

(i) GPT on cfg3f                     (j) GPT$_{rel}$ on cfg3f

Figure 22: Position-based attention pattern. The 12 rows in each layer represent 12 heads. **Observations.** The attention pattern is multi-scale: different heads or layers have different dependencies on $p$.

## D.2 Result 7: From Anywhere to NT-ends

Recall from Figure 9(a), we showed that after removing the position-bias $B_{l,h,j\to i}(x) \stackrel{\text{def}}{=} A_{l,h,j\to i}(x) - \overline{A}_{l,h,j-i}$, the attention weights have a very strong bias towards *tokens $i$ that are at NT ends*. In Figure 23 we complement this experiment with more datasets.

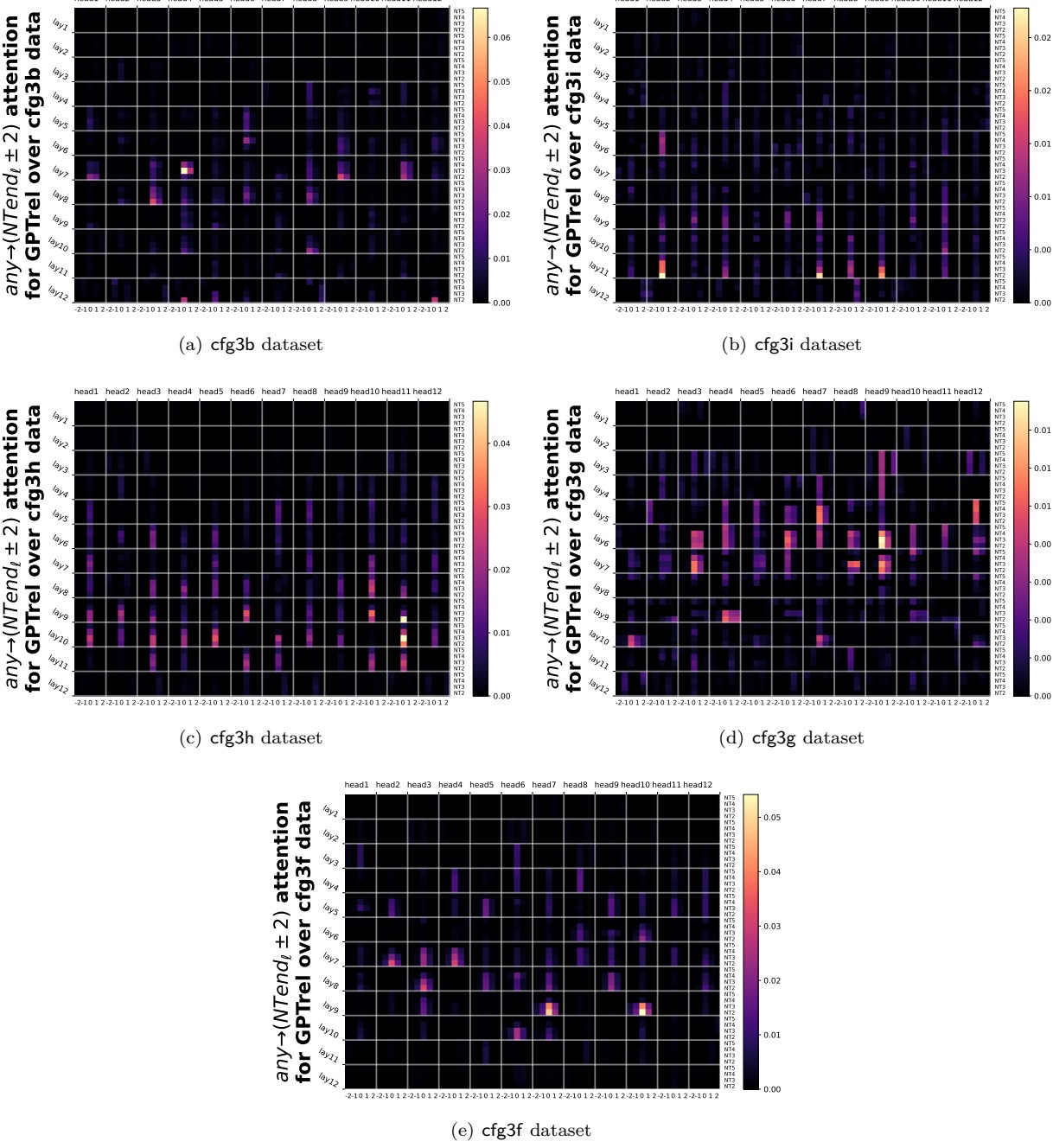

Figure 23: Attention weights $B_{l,h,j\to i}(x)$ averaged over data $x$ and pairs $i,j$ such that $i+\delta$ is at the NT-end in level $\ell$ of the CFG. In each cell, the four rows correspond to levels $\ell = 2, 3, 4, 5$, and the five columns represent $\delta = -2, -1, 0, +1, +2$.

**Observation.** Attention is largest when $\delta = 0$ and drops rapidly to the surrounding tokens of $i$.

### D.3   Result 8: From NT-ends to NT-ends

As mentioned in Section 5.2 and Figure 9(b), not only do tokens generally attend more to NT-ends, but among those attentions, *NT-ends* are also *more likely* to attend to NT-ends. We include this full experiment in Figure 24 for every different level $\ell = 2, 3, 4, 5$, between any two pairs $j \to i$ that are both at NT-ends for level $\ell$, for the cfg3 datasets.

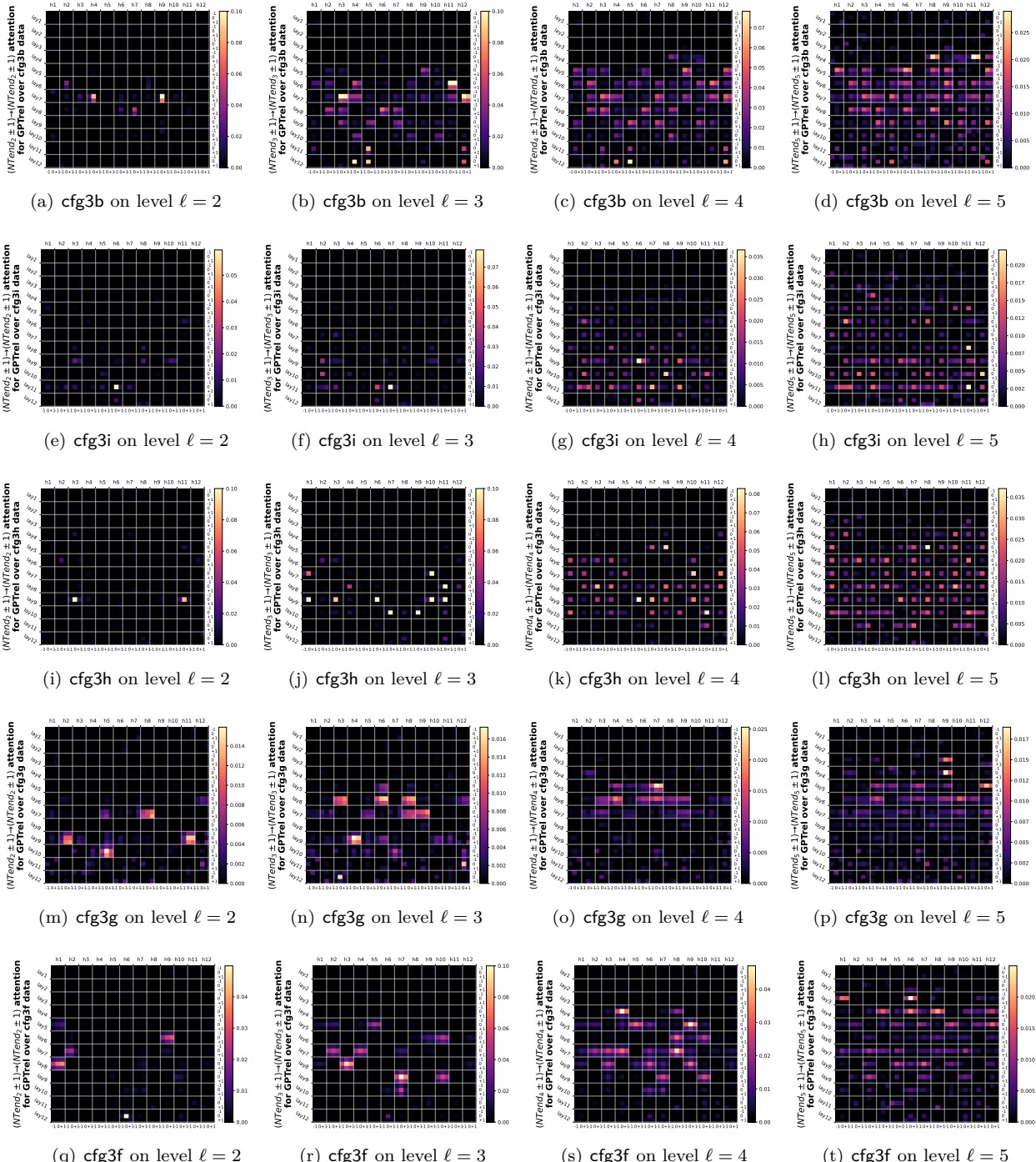

Figure 24: Attention pattern $B_{l,h,j \to i}(x)$ averaged over data $x$ and pairs $i, j$ such that $i + \delta_1$ and $j + \delta_2$ are at the NT-end boundaries in level $\ell$ of the CFG. In each block, the three rows correspond to $\delta_1 = -1, 0, +1$ and the three columns correspond to $\delta_2 = -1, 0, +1$.

**Observation.** Different transformer layer/head may be in charge of attending NT-ends at different levels $\ell$. Also, it is noticeable that the attention value drops rapidly from $\delta_1 = \pm 1$ to $\delta_1 = 0$, but less so from $\delta_2 = \pm 1$ to $\delta_2 = 0$. This should not be surprising, as it may still be ambiguous to decide if position $j$ is at NT-end *until* one reads few more tokens (see discussions under Figure 19).

## D.4 Result 9: From NT-ends to Adjacent NT-ends

In Figure 9(c) we have showcased that $B_{l,h,j\to i}(x)$ has a strong bias towards *token pairs $i,j$ that are "adjacent" NT-ends.* We have defined what "adjacency" means in Section 5.2 and introduced a notion $B_{l,h,\ell'\to\ell,r}^{\mathrm{end}\to\mathrm{end}}$, to capture $B_{l,h,j\to i}(x)$ averaged over samples $x$ and all token pairs $i,j$ such that, they are at deepest NT-ends on levels $\ell,\ell'$ respectively (in symbols, $\mathfrak{b}^\sharp(i)=\ell\wedge\mathfrak{b}^\sharp(j)=\ell'$), and of distance $r$ based on the ancestor indices on level $\ell$ (in symbols, $\mathfrak{p}_\ell(j)-\mathfrak{p}_\ell(i)=r$).

Previously, we have only presented by Figure 9(c) for a single dataset, and averaged over all the transformer layers. In the full experiment Figure 25 we show that for more datasets, and Figure 26 we show that for individual layers.

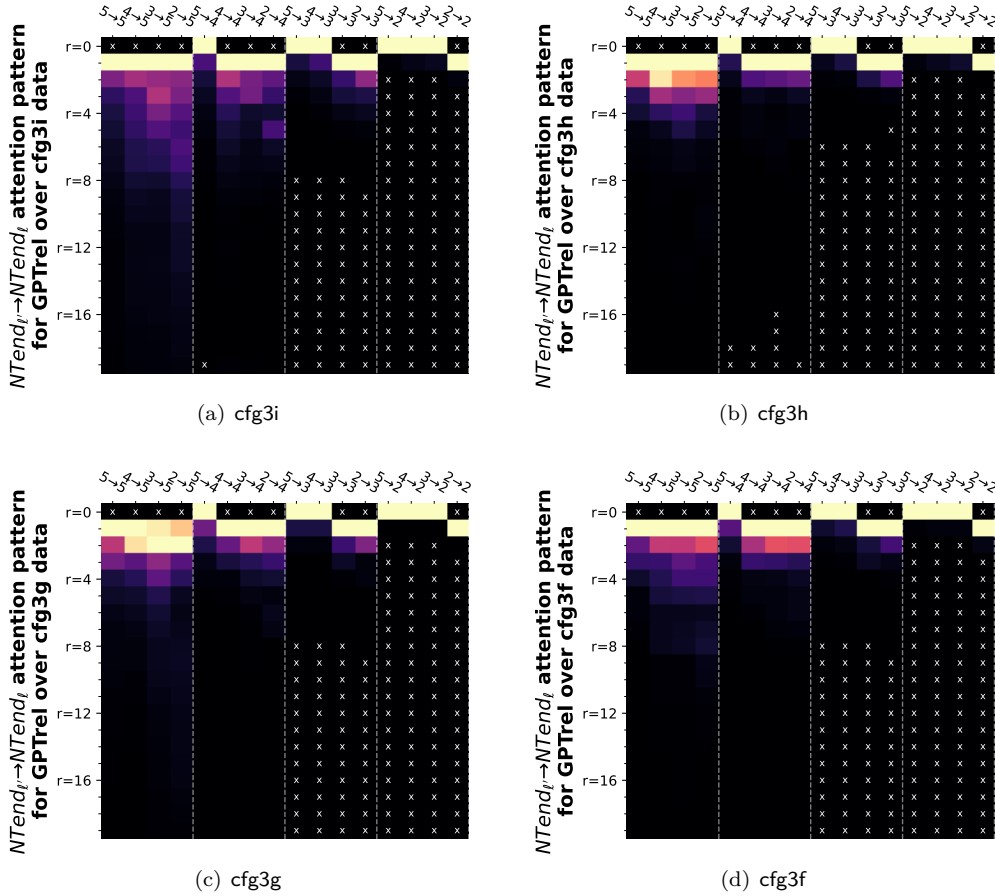

(a) cfg3i

(b) cfg3h

(c) cfg3g

(d) cfg3f

Figure 25: Attention pattern $B_{l,h,\ell'\to\ell,r}^{\mathrm{end}\to\mathrm{end}}(x)$ averaged over layers $l$, heads $h$ and data $x$. The columns represent $\ell'\to\ell$ and the rows represent $r$. "$\times$" means empty entries.

**Remark.** We present this boundary bias by looking at how close NT boundaries on level $\ell'$ attend to any other NT boundary on level $\ell$. For some distances $r$, this "distance" that we have defined may be non-existing. For instance, when $\ell\geq\ell'$ one must have $r>0$. Nevertheless, we see that the attention value, *even after removing the position bias*, still have a large correlation with respect to the smallest possible distance $r$, between every pairs of NT levels $\ell,\ell'$. This is a strong evidence that CFGs are implementing some variant of dynamic programming.

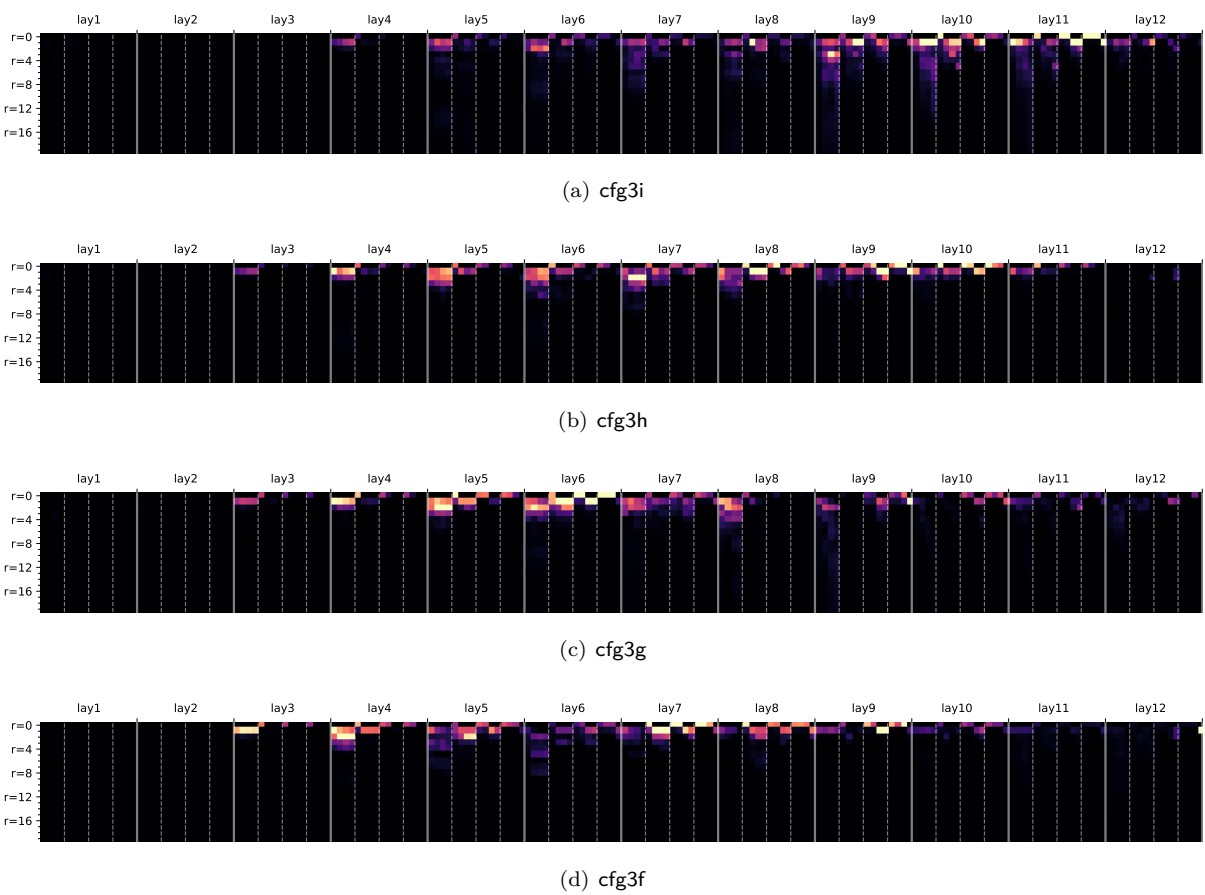

(a) cfg3i

(b) cfg3h

(c) cfg3g

(d) cfg3f

Figure 26: Attention pattern $B^{\text{end}\to\text{end}}_{l,h,\ell'\to\ell,r}(x)$ for each individual transformer layer $l \in [12]$, averaged over heads $h$ and data $x$. The rows and columns are in the same format as Figure 25.

**Observation.** Different transformer layers are responsible for learning "NT-end to most adjacent NT-end" at different CFG levels.

# E  More Experiments on Result 10 (Implicit CFGs)

We study implicit CFGs where each terminal symbol $t \in \mathbf{T}$ is is associated a bag of observable tokens $\mathbf{OT}_t$. For this task, we study eight different variants of implicit CFGs, all converted from the exact same cfg3i dataset (see Section A.1). Recall cfg3i has three terminal symbols $|\mathbf{T}| = 3$:

- we consider a vocabulary size $|\mathbf{OT}| = 90$ or $|\mathbf{OT}| = 300$;
- we let $\{\mathbf{OT}_t\}_{t \in \mathbf{T}}$ be either disjoint or overlapping; and
- we let the distribution over $\mathbf{OT}_t$ be either uniform or non-uniform.

We present the generation accuracies of learning such implicit CFGs with respect to different model architectures in Figure 27, where in each cell we evaluate accuracy using 2000 generation samples. We also present the correlation matrix of the word embedding layer in Figure 11 for the GPT$_{\mathsf{rel}}$ model (the correlation will be similar if we use other models).

| | disjoint \|vocab\|=90 | | | | disjoint \|vocab\|=300 | | | | overlap \|vocab\|=90 | | | | overlap \|vocab\|=300 | | | |
|---|---|---|---|---|---|---|---|---|---|---|---|---|---|---|---|---|
| GPT | 98.7 | 99.4 | 99.0 | 99.2 | 100.0 | 100.0 | 100.0 | 98.1 | 72.7 | 70.4 | 76.2 | 75.4 | 100.0 | 100.0 | 100.0 | 100.0 |
| GPT_rel | 99.3 | 99.7 | 99.0 | 98.9 | 100.0 | 100.0 | 98.9 | 99.1 | 97.8 | 97.9 | 92.9 | 91.9 | 100.0 | 100.0 | 100.0 | 100.0 |
| GPT_rot | 99.2 | 99.5 | 99.0 | 98.4 | 100.0 | 100.0 | 98.6 | 99.0 | 96.4 | 95.9 | 84.9 | 87.8 | 100.0 | 100.0 | 100.0 | 100.0 |
| GPT_pos | 99.2 | 99.4 | 98.4 | 99.2 | 100.0 | 100.0 | 96.6 | 96.4 | 90.1 | 91.3 | 82.6 | 83.6 | 100.0 | 100.0 | 100.0 | 99.7 |
| GPT_uni | 99.7 | 99.6 | 98.4 | 99.0 | 100.0 | 100.0 | 89.5 | 92.9 | 80.5 | 77.3 | 64.4 | 65.4 | 100.0 | 100.0 | 99.9 | 100.0 |
| | cut0 | cut50 | cut0 | cut50 | cut0 | cut50 | cut0 | cut50 | cut0 | cut50 | cut0 | cut50 | cut0 | cut50 | cut0 | cut50 |
| | uniform | | non-uniorm | | uniform | | non-uniorm | | uniform | | non-uniorm | | uniform | | non-uniorm | |

Figure 27: Generation accuracies on eight implicit CFG variants from pre-trained language models.

## F More Experiments on Results 11-13 (Robustness)

Recall that in Figure 12, we have compared clean training vs training over three types of perturbed data, for their generation accuracies given both clean prefixes and corrupted prefixes. We now include more experiments with respect to more datasets in Figure 28. For each entry of the figure, we have generated 2000 samples to evaluate the generation accuracy.

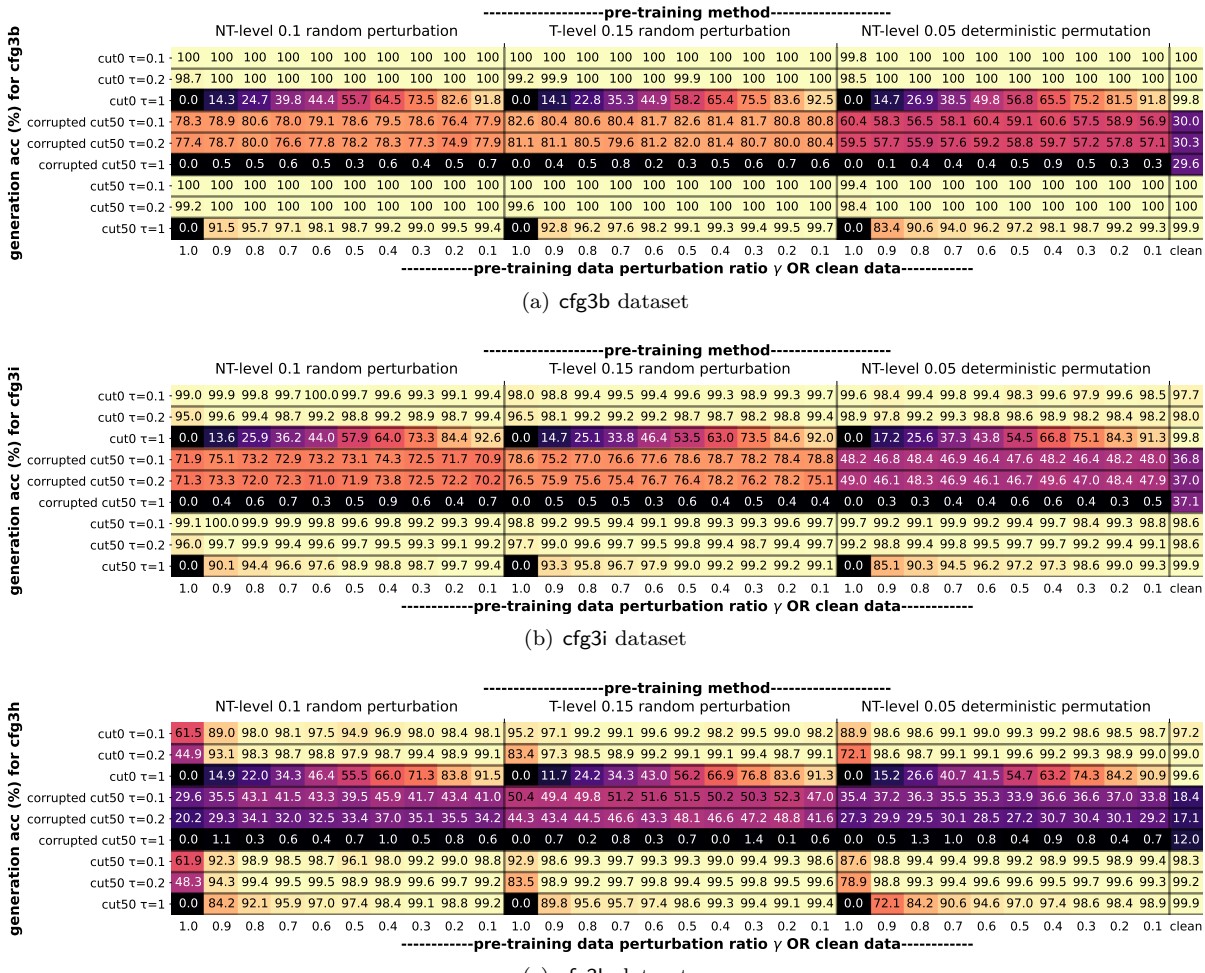

Figure 28: Generation accuracies for models pre-trained cleanly VS pre-trained over perturbed data, on clean or corrupted prefixes with cuts $c = 0$ or $c = 50$, using generation temperatures $\tau = 0.1, 0.2, 1.0$.

**Observation 1.** In Rows 4/5, by comparing against the last column, we see it is *beneficial* to include low-quality data (e.g. grammar mistakes) during pre-training. The amount of low-quality data could be little ($\gamma = 0.1$ fraction) or large (*every training sentence may have grammar mistake*).

**Observation 2.** In Rows 3/6/9 of Figure 12 we see pre-training teaches the model a *mode switch*. When given a correct prefix it is in the *correct mode* and completes with correct strings (Row 9); given corrupted prefixes it *always* completes sentences with grammar mistakes (Row 6); given no prefix it generates corrupted strings with probability $\gamma$ (Row 3).

**Observation 3.** Comparing Rows 4/5 to Row 6 in Figure 12 we see that high robust accuracy is achieved only when generating using low temperatures $\tau$. Using low temperature encourages the model to, for each next token, pick a more probable solution. This allows it to achieve good robust accuracy *even when* the model is trained totally on corrupted data ($\gamma = 1.0$).

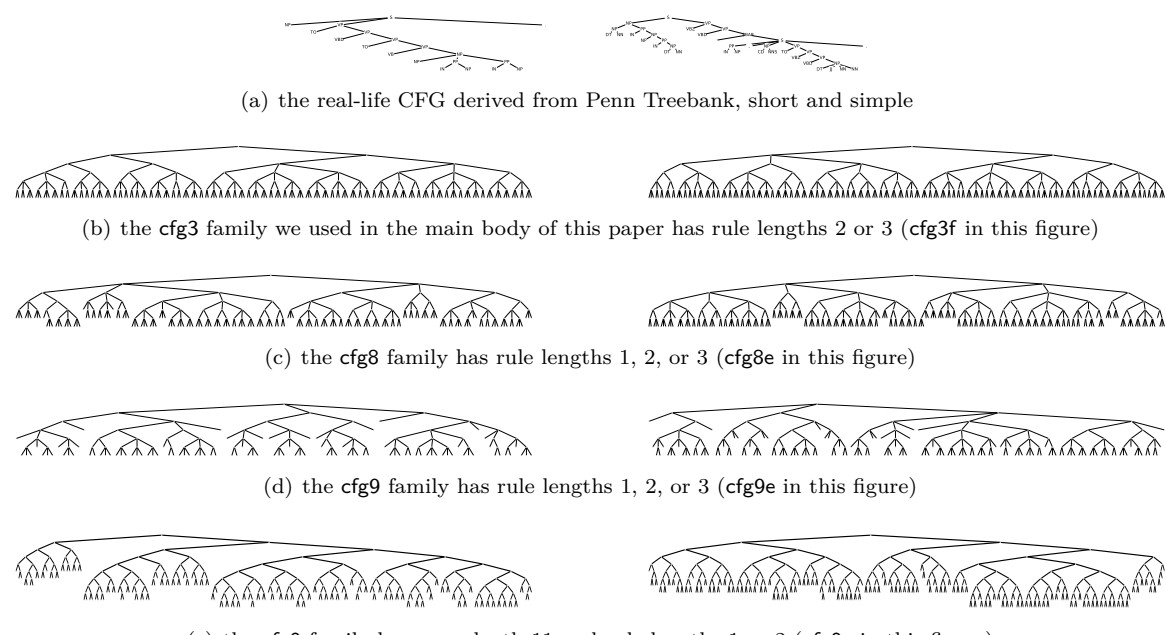

(a) the real-life CFG derived from Penn Treebank, short and simple

(b) the cfg3 family we used in the main body of this paper has rule lengths 2 or 3 (cfg3f in this figure)

(c) the cfg8 family has rule lengths 1, 2, or 3 (cfg8e in this figure)

(d) the cfg9 family has rule lengths 1, 2, or 3 (cfg9e in this figure)

(e) the cfg0 family has max-depth 11 and rule lengths 1 or 2 (cfg0e in this figure)

Figure 29: CFG comparisons: *left* is a medium-length sample and *right* is a 80%-percentile-length sample

# G Beyond the CFG3 Data Family

The primary focus of this paper is on the cfg3 data family, introduced in Section A.1. This paper does not delve into how GPTs parse English or other natural languages. In fact, our CFGs are more "difficult" than, for instance, the English CFGs derived from the Penn TreeBank (PTB) (Marcus et al., 1993). By "difficult", we refer to the ease with which a human can parse them. For example, in the PTB CFG, if one encounters RB JJ or JJ PP consecutively, their parent must be ADJP. In contrast, given a string

```
33221312331211312322113223123121112132113223113113223331231211121311331212321213333312322121312322211112133221311311311
31111113231233133133311331333332231211311121221111211233312331211311331333333112333313111133331211312113121211333332121
112121322322332221132211132221132323311311121322322322121113331121322211332211121213312133133221221322121121331123223331
```

that is in cfg3f, even with all the CFG rules provided, one would likely need a large piece of scratch paper to perform dynamic programming by hand to determine the CFG tree used to generate it.

Generally, the difficulty of CFGs scales with the average length of the strings. For instance, the average length of a CFG in our cfg3 family is over 200, whereas in the English Penn Treebank (PTB), it is only 28. However, the difficulty of CFGs may *inversely scale* with the number of Non-Terminal/Terminal (NT/T) symbols. Having an excess of NT/T symbols can simplify the parsing of the string using a greedy approach (recall the RB JJ or JJ PP examples mentioned earlier). This is why we minimized the number of NT/T symbols per level in our cfg3b, cfg3i, cfg3h, cfg3g, cfg3f construction. For comparison, we also considered cfg3e1, cfg3e2, which have many NT/T symbols per level. Figure 4 shows that such CFGs are extremely easy to learn.

To broaden the scope of this paper, we also briefly present results for some other CFGs. We include the *real-life* CFG derived from the Penn Treebank, and *three new families* of synthetic CFGs (cfg8, cfg9, cfg0). Examples from these are provided in Figure 29 to allow readers to quickly compare their difficulty levels.

## G.1 The Penn TreeBank CFG

We derive the English CFG from the Penn TreeBank (PTB) dataset (Marcus et al., 1993). To make our experiment run faster, we have removed all the CFG rules that have appeared fewer than 50 times in the

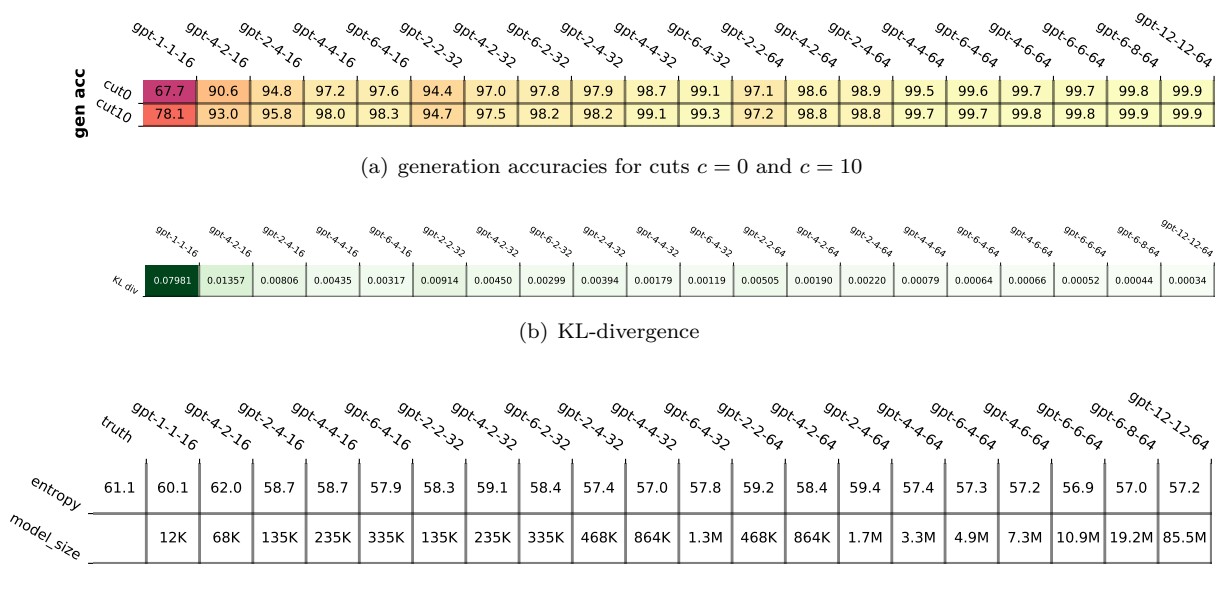

(a) generation accuracies for cuts $c = 0$ and $c = 10$

(b) KL-divergence

(c) entropy and model size

Figure 30: Real-life PTB CFG learned by $\mathtt{GPT_{rot}}$ of different model sizes.

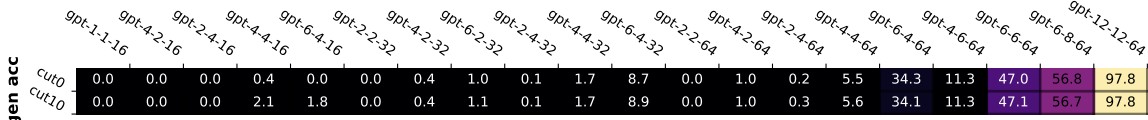

Figure 31: By contrast, small $\mathtt{GPT_{rot}}$ model sizes cannot learn the $\mathtt{cfg3f}$ data (compare to Figure 30(a)).

data.[23] This results in 44 T+NT symbols and 156 CFG rules. The maximum node degree is 65 (for the non-terminal NP) and the maximum CFG rule length is 7 (for $\mathtt{S \rightarrow " S , " NP VP .}$). If one performs binarization (to ensure all the CFG rules have a maximum length of 2), this results in 132 T+NT symbols and 288 rules.

*Remark* G.1. Following the notion of this paper, we treat those symbols such as NNS (common noun, plural), NN (common noun, singular) as *terminal symbols*. If one wishes to also take into consideration the bag of words (such as the word vocabulary of plural nouns), we have called it *implicit CFG* and studied it in Section 6.1. In short, adding bag of words does not increase the learning difficult of a CFG; the (possibly overlapping) vocabulary words will be simply encoded in the embedding layer of a transformer.

For this PTB CFG, we also consider transformers of sizes *smaller* than GPT2-small. Recall GPT2-small has 12 layers, 12 heads, and 64 dimensions for each head. More generally, we let GPT-$\ell$-$h$-$d$ denote an $\ell$-layer, $h$-head, $d$-dim-per-head $\mathtt{GPT_{rot}}$ (so GPT2-small can be written as GPT-12-12-64).

We use transformers of different sizes to pretrain on this PTB CFG. We repeat the experiments in Figure 4 (with the same pretrain parameters described in Appendix A.3), that is, we compute the generation accuracy, completion accuracy (with cut $c = 10$), the output entropy and the KL-divergence. We report the findings in Figure 30. In particular:

- Even a 135K-sized GPT2 (GPT-2-4-16) can achieve generation accuracy $\sim 95\%$ and have a KL divergence less than 0.01. (Note the PTB CFG has 30 terminal symbols so its KL divergence may appear larger

---

[23]These are a large set of rare rules, each appearing with a probability $\leq 0.2\%$. We are evaluating whether the generated sentence belongs to the CFG, a process that requires CPU-intensive dynamic programming. To make the computation time tractable, we remove the set of rare rules.

Note that $\mathtt{cfg3}$ does not contain rare rules either. Including such rules complicates the CFG learning process, necessitating a larger transformer and extended training time. It also complicates the investigation of a transformer's inner workings if these rare rules are not perfectly learned.

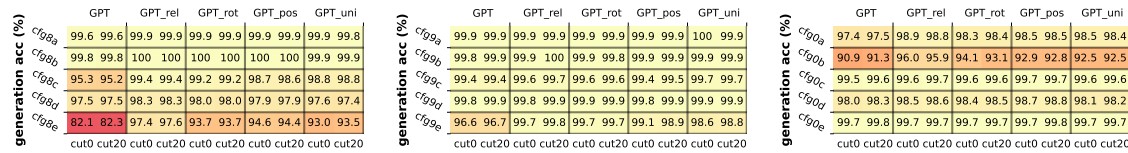

Figure 32: Generation accuracies for cfg8/9/0 data family; suggesting our results *also hold for unbalanced trees with len-1 rules.*

than that of cfg3 in Figure 4.)

- Even a 1.3M-sized GPT2 (GPT-6-4-32) can achieve generation accuracy 99% and have a KL divergence on the order of 0.001.

- Using $M = 10000$ samples, we estimate the entropy of the ground truth PTB CFG is around 60 bits, and the output entropy of those learned transformer models are also on this magnitude.

- By contrast, those small model sizes cannot learn the cfg3f data, see Figure 31.

## G.2 More Synthetic CFGs

Remember that the cfg3 family appears "balanced" because all leaves are at the same depth and the non-terminal (NT) symbols at different levels are disjoint. This characteristic aids our investigation into the *inner workings* of a transformer learning such a language. We introduce three new synthetic data families, which we refer to as cfg8/9/0 (each with five datasets, totaling 15 datasets). These are all "unbalanced" CFGs, which support length-1 rules.[24] Specifically, the cfg0 family has a depth of 11 with rules of length 1 or 2, while the cfg8/9 family has depth 7 with rules of length 1/2/3. In all of these families, we demonstrate in Figure 32 that GPT can learn them with a satisfactory level of accuracy.

For this ICLR submission, we have included all the trees used in the supplementary materials. Below, we provide descriptions of how we selected them.

**CFG8 family.** The cfg8 family consists of five CFGs, namely cfg8a/b/c/d/e. They are constructed similarly to cfg3b/i/h/g/f, with the primary difference being that we sample rule lengths uniformly from $\{1, 2, 3\}$ instead of $\{2, 3\}$. Additionally,

- In cfg8a, we set the degree $|\mathcal{R}(a)| = 2$ for every NT $a$; we also ensure that in any generation rule, consecutive pairs of terminal/non-terminal symbols are distinct. The size is $(1, 3, 3, 3, 3, 3)$.

- In cfg8b, we set $|\mathcal{R}(a)| = 2$ for every NT $a$; we remove the distinctness requirement to make the data more challenging than cfg8a. The size is $(1, 3, 3, 3, 3, 3, 3)$.

- In cfg8c, we set $|\mathcal{R}(a)| \in \{2, 3\}$ for every NT $a$ to make the data more challenging than cfg8b. The size is $(1, 3, 3, 3, 3, 3, 3)$.

- In cfg8d, we set $|\mathcal{R}(a)| = 3$ for every NT $a$. We change the size to $(1, 3, 3, 3, 3, 3, 4)$ because otherwise a random string would be too close (in editing distance) to this language.

- In cfg8e, we set $|\mathcal{R}(a)| \in \{3, 4\}$ for every NT $a$. We change the size to $(1, 3, 3, 3, 3, 3, 4)$ because otherwise a random string would be too close to this language.

A notable feature of this data family is that, due to the introduction of length-1 rules, a string in this language $L(\mathcal{G})$ may be *globally ambiguous.* This means that there can be multiple ways to parse it by the same CFG, resulting in multiple solutions for its NT ancestor/boundary information *for most symbols.* Therefore, it is not meaningful to perform linear probing on this dataset, as the per-symbol NT information is mostly non-unique.[25]

**CFG9 family.** Given the ambiguity issues arising from the cfg8 data construction, our goal is to construct an unbalanced and yet challenging CFG data family where the non-terminal (NT) information is mostly unique, thereby enabling linear probing.

---

[24]When a length-1 CFG rule is applied, we can merge the two nodes at different levels, resulting in an "unbalanced" CFG.

[25]In contrast, the cfg3 data family is only *locally* ambiguous, meaning that it is difficult to determine its hidden NT information by locally examining a substring; however, when looking at the entire string as a whole, the NT information per symbol can be uniquely determined with a high probability (if using for instance dynamic programming).

Figure 33: Same as Figure 5 but for the cfg9 family. After pre-training, hidden states of generative models implicitly encode the NT ancestors information. The $NT_\ell$ column represents the accuracy of predicting $\mathfrak{s}_\ell$, the NT ancestors on level $\ell$. This suggests our probing technique applies more broadly.

To accomplish this, we first adjust the size to $(1, 4, 4, 4, 4, 4, 4)$, then we permit only one NT per layer to have a rule of length 1. We construct five CFGs, denoted as cfg9a/b/c/d/e, and their degree configurations (i.e., $\mathcal{R}(a)$) are identical to those of the cfg8 family. We then employ rejection sampling by generating a few strings from these CFGs and checking if the dynamic programming (DP) solution is unique. If it is not, we continue to generate a new CFG until this condition is met.

Examples from cfg9e are illustrated in Figure 29. We will conduct linear probing experiments on this data family.

**CFG0 family.** Since all the CFGs above support rules of length 3, we have focused on $L = 7$ to prevent the string length from becoming excessively long.[26] In the cfg0 family, we construct five CFGs, denoted as cfg0a/b/c/d/e. All of them have a depth of $L = 11$. Their rule lengths are randomly selected from $\{1, 2\}$ (compared to $\{2, 3\}$ for cfg3 or $\{1, 2, 3\}$ for cfg8/9). Their degree configurations (i.e., $\mathcal{R}(a)$) are identical to those of the cfg8 family. We have chosen their sizes as follows, noting that we have enlarged the sizes as otherwise a random string would be too close to this language:

- We use size $[1, 2, 3, 4, 4, 4, 4, 4, 4, 4, 4]$ for cfg0a/b.
- We use size $[1, 2, 3, 4, 5, 6, 6, 6, 6, 6, 6]$ for cfg0c.
- We use size $[1, 2, 3, 4, 5, 6, 7, 8, 9, 10, 11]$ for cfg0d/e.

Once again, the CFGs generated in this manner are globally ambiguous like the cfg8 family, so we cannot perform linear probing on them. However, it would be interesting to demonstrate the ability of transformers to learn such CFGs.

**Additional experiments.** We present the generation accuracies (or the complete accuracies for cut $c = 20$) for the three new data families in Figure 32. It is evident that the cfg8/9/0 families can be learned almost perfectly by GPT2-small, especially the relative/rotary embedding ones.

As previously mentioned, the cfg9 data family is not globally ambiguous, making it an excellent synthetic data set for testing the encoding of the NT ancestor/boundary information, similar to what we did in Section 4. Indeed, we replicated our probing experiments in Figure 33 and Figure 34 for the cfg9 data family. This suggests that **our probing technique has broader applicability.**

## H More on Uniform Attention

In Result 1, we observed that GPT_uni (uniform attention) performs surprisingly well—significantly outperforming the original GPT with absolute positional embeddings. Although interpretability is the primary focus of this paper, we briefly highlight the robustness and implications of this result.

Since GPT_uni lacks query and key matrices, its per-layer parameter count is approximately $10d^2$ (for hidden size $d$), compared to $12d^2$ for vanilla GPT. Thus, a parameter-matched comparison would be GPT$(12, 768)$ (12 heads, 768 dimensions) versus GPT_uni$(8, 840)$ (8 heads, 840 dimensions). For completeness, we also compare with other natural baselines such as GPT$(8, 1024)$ and GPT$(12, 936)$. All experiments are summarized in Figure 35.

---

[26]Naturally, a larger transformer would be capable of solving such CFG learning tasks when the string length exceeds 1000; we have briefly tested this and found it to be true. However, conducting comprehensive experiments of this length would be prohibitively expensive, so we have not included them in this paper.

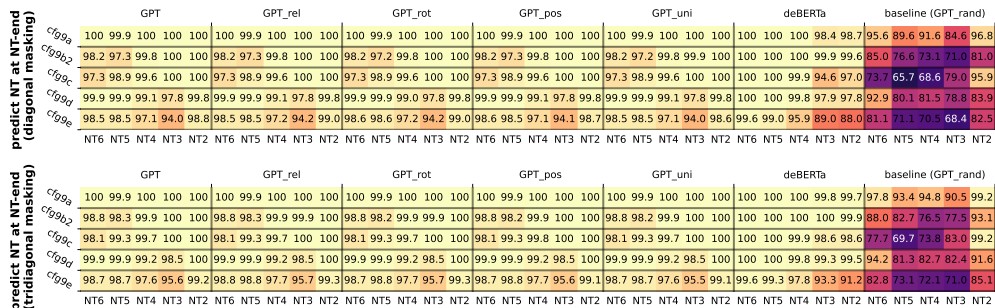

Figure 34: Same as Figure 7 but for the `cfg9` data family. Generative pre-trained transformer encodes NT ancestors almost exactly at NT boundaries. The $NT_\ell$ column represents the accuracy of predicting $\mathfrak{s}_\ell(i)$ at locations $i$ with $\mathfrak{b}_\ell(i) = 1$. This suggests our probing technique applies more broadly.

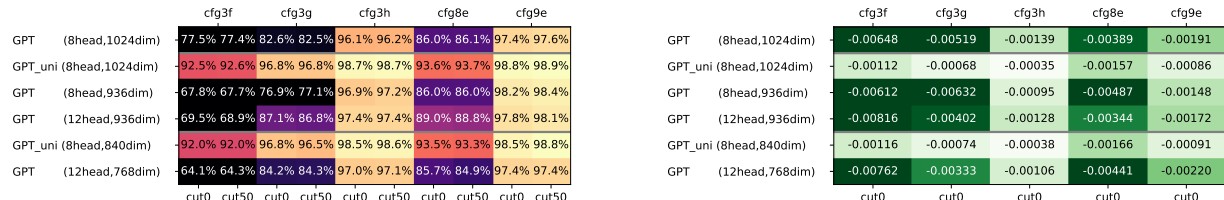

Figure 35: Performance comparison between $\text{GPT}_{\text{uni}}$ (uniform attention) and vanilla GPT (absolute positional embedding). Left: generation accuracy; Right: KL divergence against ground truth. Each model is trained with 5 random seeds. We report the *median* for $\text{GPT}_{\text{uni}}$ and the *best* run for GPT.

At a high level, we group six 12-layer models by parameter scale:

$$\text{GPT}(12, 768) \approx \text{GPT}_{\text{uni}}(8, 840) \ll \text{GPT}_{\text{uni}}(8, 1024) \approx \text{GPT}(12, 936) \approx \text{GPT}(8, 936) \ll \text{GPT}(8, 1024) \ .$$

Yet, across all settings, Figure 35 consistently shows that uniform attention $\text{GPT}_{\text{uni}}$ outperforms vanilla GPT—*even when* the latter has more trainable parameters.

While we do not claim this constitutes a comprehensive architecture benchmark (CFG tasks do not reflect the full spectrum of language abilities), these results reinforce the strength of uniform attention. This connects to prior work such as ALiBi Press et al. (2021) and especially H-Alibi Jelassi et al. (2024), which apply hard attention cutoffs—where each attention head attends only to a fixed-size window. This is structurally similar to our $\text{GPT}_{\text{uni}}$, where the window size varies per head.

These findings also motivate our follow-up work (Allen-Zhu, 2025a;b), where we incorporate short-window uniform attention as a lightweight architectural component to further improve Transformer performance without increasing parameter count significantly.

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
