# OpenReview forum: "Physics of Language Models: Part 1, Learning Hierarchical Language Structures"
_TMLR — Accepted by TMLR_

### Review · Reviewer_vyuo · 2025-09-01

**Summary Of Contributions:**

This paper aims to investigate the learning of hierarchical structures in transformers, with the goal of understanding how large language models solve complex tasks with deep reasoning and computation chains. To that end, the work generates a large number of synthetic context-free grammars (CFGs), which are inherently hierarchical in nature. Parsing such CFGs can require long range reasoning and handling local ambiguity, since the next symbol in a CFG-generated string may only be uniquely determined by tokens that occurred hundreds of tokens on the "past." The paper validates, on a GPT2 small architecture, that a set of synthetic CFGs are almost perfectly learnable, and are likely generalizing rather than memorizing. Further experiments suggest that information about the CFG (i.e., parent "nodes") are extractable from GPT2's internal representations via linear probing, and further analysis of the attention mechanisms suggest that they are critical to learning CFGs.

**Audience:**

No

**Audience Explanation:**

The relevance/significance of this work to the ML community is unclear. While the motivation in the intro is given as: "Is there a setting for us to understand how language models perform hard tasks, involving deep logics / reasoning / computation chains?" It's unclear the extent to which CFG is the correct abstraction for this. In addition, "hard" is underdetermined, as is "deep logic/reasoning," so it is hard to follow formally what question the work is really trying to answer. In other words, why is it interesting that a model can or cannot learn to recognize a CFG? In what way does that give us a "bridge" into "interesting" tasks (as determined by the literature, or other practical justifications)?

The paper also uses a rather outdated model for experiments (GPT2 small), though this is OK for the purposes of building a mechanistic understanding of how models learn CFGs.

Ultimately, without a stronger connection to current LLM problem-solving work it's unclear where this work fits in the ML community.

**Broader Impact Concerns:**

None.

**Claims And Evidence:**

No

**Claims Explanation:**

I see design and clarity issues in the current manuscript. First, on the design side, Appendix B.1 shows that the CFGs are non-recursive: a symbol can never expand to itself. This is subtle, but without "true" recursion the proposed CFGs reduce to finite languages — a much smaller class of languages and I'd expect this to be easier to learn for models in general (cf. Chomsky hierarchy — all finite languages are regular). However, the significance of this design flaw is unclear without a better sense of why CFGs are the right level of analysis.

On clarity — it is very difficult to follow Results 6-9, because it's unclear what we expect to see. The results feel more exploratory and I have trouble parsing what the paper is trying to show, especially Figure 8. Results 10-13 seem a little more orthogonal to the main paper as well and I was confused about their placement.

**Requested Changes:**

**Critical:** Please answer the following questions and revise such that these explanations are readily available in the text.
* **MOST CRITICAL:** Why CFGs at all? Why are they an appropriate abstraction for (L)LM problem solving/reasoning, OR why is it interesting that models can recognize CFGs?
    * I'm looking for something to convince me that a CFG-based abstraction for problem-solving in LLMs is (1) salient/sufficiently faithful, and (2) relevant to understanding problem-solving in LLMs, preferably with concrete examples. It is difficult for me to advocate that this work is relevant to the TMLR community otherwise, so a thorough & convincing answer here could really improve my assessment.
* Does the non-recursive/finite formulation of the CFG affect the conclusions?
* Why anchor to a DP based formulation of problem-solving? This is just one problem-solving strategy; what is its relevance to (1) the learnability of CFGs (i.e., must a model use DP to recognize a CFG) and (2) broader (L)LM problem-solving in general?
* What is an NT "boundary" intuitively, and why are those interesting locations to analyze (cf. Results 4-5)?
* What is the purpose of Results 6-9, and intuitively, what results should I expect to see?

**Important:** It would be nice to get clarity on the following design decisions:
* What's the motivation for choosing the GPT2 architecture, as opposed to a more recent open-weight model (e.g., Llama3 family, Qwen2.5 or 3)?
* All of the synthetic CFGs seem to have 3 terminal symbols with 2-3 expansion rules per non-terminal symbol, with the main difference between CFGs being the "depth" of the CFG. While that is one possible axis of CFG difficulty, why not consider others?
    * E.g., more terminals, more complex expansion rules, recursive rules, or heterogeneity in the number of rules per non-terminal (e.g., perhaps a "3" usually is generated by a "5" but occasionally by a "4" — does the model just rely on distributional statistics and predict the parent is a "5"?).
    * Without justification for why this set of CFGs is sufficient, or extensions as described above, the claims seem to apply to a subclass of specific CFG rather than CFGs in general.
* What's the justification for the noisy CFG experiment? It's unclear why this experiment is central for the main claims of the paper, and I might suggest a move to an Appendix.
* "Thus, human languages may be too easy for our interpretability purpose." — I think this claim rests on a misspecified CFG for human language. I believe this is true if we're using a CFG over grammatical units (e.g., parts of speech, phrases), but not over *words.* Certain words could appear in a variety of phrases (e.g., "run" in an NP or VP) and introduce a similar ambiguity as seen in the synthetic CFGs.
    * (strictly speaking — perhaps the authors mean that human *grammars* are too easy? Grammar != language, so that is somewhat more believable, but I don't have the expertise to verify.)

---

> ### Author Response · Authors · 2025-10-09
> **Response to Reviewer vyuo (1/2)**
>
> We thank the reviewer for their detailed feedback and for taking the time to engage with our paper.
> We **respectfully note** that many of the concerns raised focus on why this class of CFGs is studied, rather than on the technical correctness of our claims and evidence.
> We fully understand that this direction—synthetic CFGs as interpretability testbeds—*may not appeal to all readers*.
> However, as per TMLR’s acceptance criteria, our goal is not to argue for universal significance, but to ensure that
> 1. our claims are supported by accurate evidence, and
> 2. the work is of interest to at least some members of the TMLR community.
>
> In this spirit, we have **revised the paper to further strengthen the clarity and rigor of our technical findings**.
> We also respectfully note that the same CFG framework has been used in our follow-up research (not cited due to anonymity) to design improved Transformer architectures at 8B scale, suggesting that the approach is meaningful to at least part of the community --- not only to interpretability readers, but perhaps to architecture design readers as well.
>
> Below, we address each of the reviewer’s points in detail.
>
> ### 1. Why study CFGs at all?
>
> We appreciate the concern that CFGs may not capture all aspects of LLM reasoning.
> Our motivation is that CFGs do capture key aspects—hierarchical structure, long-range dependencies, and local ambiguity—while providing a controlled, mathematically rigorous setting for interpretability.
> This enables us to probe model internals in a way that is difficult to achieve on natural data.
> We recognize that not all researchers will find this abstraction appealing, but interpretability researchers and model-design practitioners have found it useful for understanding how Transformers represent and manipulate hierarchical structures.
> Following TMLR’s guidelines, we believe this satisfies the criterion of technical correctness and interest to a subset of readers.
>
> ### 2. Finite / non-recursive CFGs
>
> While our grammars are finite, they are doubly exponential in size (for instance, cfg3f generates roughly 10^80 valid strings among ~10^140 possible combinations).
> Thus, they are far from trivial—parsing or generation requires dynamic-programming-level complexity.
> The choice of non-recursive, layered CFGs was intentional: it provides clean interpretability and enables systematic probing across levels of hierarchical representation.
> These grammars therefore form a non-trivial, well-defined synthetic language suitable for studying the internal computations of Transformers.
>
> ### 3. Why anchor the analysis to dynamic programming (DP)?
>
> We appreciate this question.
> Our use of DP is not intended to imply that the model implements a specific algorithm—proving such equivalence would be intractable (unless the algorithm is extremely simple, such as associate recall).
>
> Rather, DP serves as a natural and theoretically grounded reference for reasoning about CFG parsing and generation.
> Our results show that the model’s hidden states and attention patterns align with the information flow characteristic of DP-style parsing, where intermediate results are stored and composed hierarchically.
> The **revised Section 5.2+5.3** elaborate on this correspondence, explaining better how Results 4–9 **collectively provide evidence for DP-like computation within Transformers**.
>
> ### 4. What are NT boundaries, and why analyze them?
>
> Non-terminal (NT) boundaries indicate where subtrees in the CFG complete.
> These are the most informative positions for analyzing hierarchical structure, because they correspond to where the model must “close” a constituent—precisely where DP algorithms perform memory reads or writes.
> We analyze them to test whether the Transformer’s internal representations reflect these compositional transitions.
> This is also consistent with the **newly added** pseudocode (Algorithm 1) and visualizations (Figures 8–9) that explicitly link NT-boundary attention to DP-style information flow.
>
> ### 5. Purpose and expectations of Results 6–9
>
> These results were designed to examine whether attention patterns reflect the expected DP-like propagation of information across NT boundaries.
> We have **improved the writing in Section 5.2 (blue text)** to state the hypotheses and expected outcomes explicitly and to connect them directly to Section 5.3, where the full DP correspondence is discussed.
> We believe this clarification makes the contribution and purpose of these results much clearer.

---

> ### Author Response · Authors · 2025-10-09
> **Response to Reviewer vyuo (2/2)**
>
> ### 6. Model choice (GPT-2 small)
>
> Our goal is to study the core Transformer mechanism, not implementation differences across model families.
> GPT-2 small provides an open, well-understood baseline that is sufficient to solve these difficult CFG tasks.
> Modern models such as LLaMA 3 and Qwen 2.5 differ mainly in tokenizer, group-query attention, or rotary embedding—of which the rotary embedding is already included in our experiments; tokenizer is irrelevant, and group-query attention is not needed for smaller-sized models.
> Given that our study focuses on technical correctness of the interpretability results, rather than model “significance,” we believe GPT-2 small is an appropriate and transparent choice.
>
> ### 7. CFG design axes
>
> We agree that other CFG variants (e.g., recursive or heterogeneous) could be interesting to explore.
> However, we intentionally focused on balanced, layered CFGs for easier interpretability and visualization.
> Some unbalanced CFGs are already included in Appendix H for completeness, but the key findings hold across both settings.
> Again, our emphasis is on providing technically sound and interpretable evidence rather than exhaustive coverage of all CFGs.
>
> ### 8. Noisy CFG experiments
>
> The noisy CFG experiments (Section 6) show how Transformers behave when trained or prompted with data containing grammatical errors—mirroring real-world situations where corpora are imperfect.
> The results reveal distinct “grammar-correct” and “grammar-error” states, an observation that connects to practical model behavior under temperature-controlled generation.
> Some readers may find this aspect particularly relevant for understanding robustness in generative models.
>
> ### 9. On the statement “human languages may be too easy”
>
> We thank the reviewer for catching this phrasing.
> We indeed misspoke—the intended meaning was that CFGs derived from human-language corpora. We **have revised the text** to clearly state this now.
>
> # Closing remark
>
> We again thank the reviewer for their thoughtful comments.
> We fully acknowledge that this line of work may not appeal to all readers.
> Our intent is not to claim broad significance but to **contribute technically correct, carefully evidenced findings** that shed light on how Transformers represent and process **some** hierarchical structures.
> We hope these clarifications and revisions address the reviewer’s concerns and make our work clearer and more accessible to interested members of the TMLR community.
>
> Thanks a lot for your consideration.

---

> > ### Comment · Reviewer_vyuo · 2025-10-09
> >
> > Thanks for the thorough response! This is helpful — I think my concern about broader appeal is anchored to whether CFGs are the correct abstraction to use to analyze problem-solving capabilities and not a write-off of CFGs as a tool. As I said in my review, I wanted to understand why CFGs as an abstraction are "(1) salient/sufficiently faithful, and (2) relevant to understanding problem-solving in LLMs, preferably with concrete examples."
> >
> > **Suggestions on broader significance/interestingness.** The response makes good progress here — in light of the author response, I now see my hesitance about CFGs as a fixable framing issue. I like the explanation "The choice of non-recursive, layered CFGs was intentional: it provides clean interpretability and enables systematic probing across levels of hierarchical representation." This supplements the paper nicely — to make this click 100%, it would be nice to subsequently highlight some examples of practical tasks where "skills" needed to solve CFGs are active; e.g., reasoning under local ambiguity or with long-range dependencies as stated in section 2 (without needing to claim that such tasks reduce to CFGs).
> >
> > As a side note: the explanation for "why noisy CFGs" in the author response (noisy CFGs are like "grammatical errors") is a great example of such a practical anchor. I'm sure many tasks analyzed in the LLM benchmarking literature can be framed as reasoning under local ambiguity or capturing long-range dependencies, such as parsing large codebases, proof-writing, or other tasks that require symbolic reasoning (perhaps AIME arguably counts here) — looking forward to seeing such an argument in more depth.
> >
> > With these updates, I think the chain of reasoning for this paper becomes "Transformer-based models are often used for Task(s) X -> Task(s) X require Skill(s) Y as a prerequisite -> CFGs are a good model for Skill(s) Y" and the utility of this work in the broader community becomes very obvious to me. In my first pass, I interpreted the work as "Consider Skill(s) Y in the abstract -> Skill(s) Y are encoded in CFGs." Even if I agree that Skill Y is non-trivial, without the practical "anchor" I was left a little confused about the place of this work in the community. Since the edit distance is quite small here, I favorably update my prior on relevance to the community.
> >
> > **On the DP-like interpretation/the attention map results** On my remaining concerns/the DP-like interpretation — Figure 9 is extremely helpful in conjunction with the clarifications in Section 5.2. It is now very clear what I expect to see and how we can use attention as a proxy to observe DP-like behaviors, because Figure 9 + the surrounding explanation establishes that (1) one should expect to see patterns at some distance related to the gap between NT-end boundaries, and (2) the DP algorithm for parsing the CFG inherently operates with "awareness" of this gap. I still don't quite follow Figure 8 though — it appears to be some summary of the extent to which we observe this boundary-attention like behavior but I can't quite see how to read that off of the plot. It seems I'm not alone in highlighting concerns about Figure 8; perhaps the authors could explain in plain English what they aim to show and suggest some alternative figure designs for doing so?
> >
> > The other technical concerns are resolved/I'm convinced they're not a reason to reject. Further suggestions:
> >
> > **Clarity.** I think the clarity of the writing and general structure can still be improved. For example, while the added explanations in Section 5.2 are great technically, I don't think there's a need for colloquial expressions such as "we go further below/we did not stop there." The thoroughness of the explanations/results should already speak to that. On general structure: I've seen many papers explicitly list out the core claims/main contributions in the introduction, and why they matter for the paper conclusions (e.g., one sentence per section). I think such a clarifying section would help reduce the risk of misinterpretation of the paper's main goals. I see some of this done at the bottom of page 2; I think, for an intro, the overview can afford to be slightly more high-level.
> >
> > Thanks for the engagement — I find the revision and response to be an objective improvement in quality.

---

### Review · Reviewer_BL2E · 2025-09-04

**Summary Of Contributions:**

The paper investigates whether and how autoregressive Transformers, trained solely on terminal tokens without access to parse-tree supervision, can implicitly learn context-free grammars. It shows that GPT-style models can learn to generate from several synthetic, often locally ambiguous and non-trivial CFG families, producing strings with high validity and close distributional match to the target languages. Probing and interpretability analyses—such as linear probes that recover nonterminal structure and attention patterns consistent with dynamic-programming-like information flow—provide strong evidence that the models internalize aspects of the underlying generative process rather than relying on rote memorization or lookup tables.

**Audience:**

Yes

**Audience Explanation:**

This paper is mostly for readers interested in mechanistic interpretability and algorithmic reasoning in LLMs, best read with familiarity with CFG notation and DP parsing.

**Claims And Evidence:**

Yes

**Claims Explanation:**

The core claims are backed by careful measurements and ablations on well-specified synthetic CFG benchmarks. However, a few conclusions (especially the DP interpretation) are suggestive rather than strictly causal, and external validity to natural language is explicitly limited.

* **Evidence of learning:** High completion accuracy across prefix cuts, strong generative diversity (entropy/birthday bound), and low next-token KL indicate generalization beyond memorization.
* **Internal structure:** Multi-head linear probes (full/diagonal/tridiagonal) nearly perfectly recover nonterminal ancestors and boundaries; encoder baselines fail on deeper NTs.
* **Attention & DP:** After removing position bias, attention concentrates on adjacent NT-end tokens, aligning with DP-style information flow (alignment, not proof).
* **Robustness & extensions:** Noisy-prefix pretraining and low temperatures boost robust accuracy; implicit CFGs yield structured embeddings.
* **Baselines:** Relative/rotary ≥ vanilla; uniform surprisingly strong.

**Requested Changes:**

1. For Result 3, please detail the exact DP used to compute grammar conditionals and consider adding per-prefix likelihood/calibration curves in addition to the KL summary

2. Define NT, “NT boundary,” $s_\ell(i)$, $b_\ell(i)$, and the probe $G_i(x)$ in words and symbols before Section 4; many readers won’t infer these from the equation alone. Link each symbol to where it first appears.

---

> ### Author Response · Authors · 2025-10-09
> **Response to Reviewer BL2E**
>
> We sincerely thank the reviewer for the thoughtful and constructive feedback, and for finding that the paper’s core claims are well supported and potentially interesting to some (certainly not all) readers interested in interpretability and algorithmic reasoning.
>
> ### Result 3 and DP computation.
> Following the reviewer’s suggestion, we added a **pseudocode (Algorithm 1)** detailing the DP procedure used to compute grammar conditionals and **revised Sections 5.2–5.3** (blue highlights) to connect the attention-pattern results more clearly with the DP illustration.
>
> Regarding the suggestion to add per-prefix likelihood or calibration curves:
> our vocabulary size is |V| = 3 (plus EOS), so the per-prefix conditional distribution consists of only four probabilities.
> Unless we misunderstood the question, such curves would be not be very informative?
> Instead, we have reported calibration/KV results averaged over all token positions and input strings, which provides a more stable summary.
>
>
> ### Definitions of NT, NT boundary, and probe.
>
> Thanks and great suggestions. We have improved visibility of these definitions throughout the paper:
>
> * Itemized explanations were added to the caption of Figure 2 to increase visibility **(revised in blue)**;
>
> * Formal definitions remain in Section 2 (unchanged);
>
> * Section 4 more explicitly reminds readers of the notions from Section 2 & Fig 2 at the beginning **(revised in blue)**;
>
> * In Section 4.1, we added a verbal explanation of how the probe Gᵢ(x) operates **(revised in blue)**.
>
> We appreciate these helpful suggestions, which have improved the paper’s clarity and readability.
> We hope these changes make the technical presentation smoother and more accessible to readers less familiar with the notation.
> Thanks again and have a good rest of your day.

---

### Review · Reviewer_CZQJ · 2025-09-28

**Summary Of Contributions:**

In this paper, the authors train a number of transformer models on synthetic fixed-depth (non-recursive) context-free grammars (CFGs), and then present some analyses of what features those models have learned. Specifically, they find that
(1) transformers can be trained to fairly high accuracy on the synthetic CFGs,
(2) a "multi-head linear probing" layer can be trained to recover nonterminal symbols from the learned embeddings (but not from random embeddings),
(3) attention patterns are consistent with the hypothesis that the models have learned to perform a dynamic-programming-like algorithm,
(4) the models can be trained to recover from noise in the CFG generation process.

Overall, this provides evidence that sufficiently-powerful transformer models _sometimes_ learn non-recursive dynamic-programming-like algorithms, and can identify regularities of the underlying CFG based only on terminal token sequences.

**Audience:**

Yes

**Audience Explanation:**

These experiments give some interesting insights into how transformers are capable of tracking hierarchically-structured sequences, and I think some members of the community would find them interesting. (I am unsure how representative these tasks are, but I think it is sufficiently interesting for publication in TMLR.)

**Broader Impact Concerns:**

No broader impact concerns.

**Claims And Evidence:**

No

**Claims Explanation:**

Most of the claims made by the paper seem correct and adequately supported by evidence. (I am not sure how much these results should tell us about the behavior of real-world transformer models, but the authors are clear about these limitations, and this work does provide evidence that transformers can learn these kinds of algorithms given the right data.)

However, I do have a few questions and concerns related to the correctness and clarity of some of the claims:

## Details and interpretation of multi-head linear probing layer
Section 4.1 introduces a "Multi-head linear probing" layer, which appears to be a per-token linear layer projected through some sort of attention matrix. I found this section confusing and am not sure how to interpret the results, or what claims can reasonably be made about what this probe learns.

As I understand it, the attention matrix is _fixed_, and is not supposed to depend at all on the input $x$ (since the authors want it to be "linear"). However, the attention matrix depends on token position, which depends on the length and content of $x$. Does this mean that the attention matrix is directly optimized to achieve high accuracy for a fixed input $x$?

My understanding is that (linear) probes are usually trained on a training set, and then subsequently evaluated on a held-out test set, to avoid overfiting and ensure that they are actually probing for a general concept. But this does not seem possible with this probe, making me question what these results are showing. If you allow the linear mapping to be learned separately for each $x$, it seems like there should _always_ be some linear mapping that achieves 100% training accuracy.

In practice there is clearly _something_ nontrivial happening, since the accuracy for trained transformer models is higher than for untrained ones. However, if the probe is being trained separately for each $x$, it seems plausible to me that this is primarily an artifact of the inductive biases of the specific probe architecture used, rather than being fundamental to the representation learned by the model. It would be good to clarify this.

## Complexity of the synthetic CFGs
Page 4 states "Typically, the learning difficulty of CFGs inversely scales with the number of NT/T
symbols, assuming other factors remain constant, because having **more** NT/T symbols makes the language less ambiguous and more easily parsed using greedy". However, page 5 states "Less ambiguous CFGs (cfg3e1, cfg3e2, as they have **fewer** NT/T symbols) are easier to learn". These statements seem contradictory. Which is correct? (or am I misunderstanding?)

## Attention diagrams

I found Figure 8 to be quite difficult to interpret. It seems that the cells being visualized represent specific offsets where we might expect a dynamic programming algorithm to attend or not attend, and the diagram is showing whether the model actually attends to these locations?

Overall it was not clear to me what claims were being made in this figure and the associated analysis in section 5. My best guess for Figure 8a and 8b is that we should expect a true DP algorithm to attend at $\delta = 0$, and we do actually observe this, which is evidence that the attention may be implementing something like a DP algorithm? But this is not obvious from the way section 5 is written. Additionally, I have no idea what Figure 8c is meant to be evidence of, or whether or not the findings are consistent with what one would expect.

**Requested Changes:**

## Clarify how the multi-head linear probing layer is used
I think the authors should clarify the way that the multi-head probing layer is trained, and in particular whether it is (over)fitting to a specific input or whether it is shared between inputs. This also affects the claims that can be reasonably made based on how well the layer works.

If it is optimized against a single input, I think the paper should state this clearly, and avoid claiming that this is a linear probe in the same sense that "linear probe" is typically used in the literature. (The final transformation being linear is not a meaningful restriction if it is being fit to a specific example; if you include the optimization steps for each example the full extraction process is highly nonlinear.) If it is trained on one set and evaluated on another, the paper should describe this process.

## Explain claims in Section 5 and meaning of figure 8 more clearly
I think two things that would improve this part would be:

1. Explain more clearly what the hypothesis is that you are testing with each of these experiments; what would we expect to see (and why) if the hypothesis is true, and what would we expect to see (and why) if the hypothesis is false / an alternative hypothesis is true?
2. Ground the results in an actual sequence, and show what the attention matrices actually look like, rather than just showing very small offset slices of 3- or 5-token windows out of context (which are difficult to interpret on their own).

## Address some issues of how this work is positioned relative to prior work

A few of the references to prior work in this paper are strangely worded, and some of the statements about prior work seem unreasonable relative to the current submission. I think the authors should reword these to better contextualize this work.

- For induction heads:
  - Why quote "hypothesized"?
  - It also seems inappropriate to quote and emphasize “don’t have a strong framework for mechanistically understanding”, because this work also does not provide such a framework: Olsson et al. (2022) were referring to complex heads learned on real-world data which have more complicated behavior than a simple induction head, and this current work does not have any experiments with real-world data or investigate behavior of any particular heads at all.
- For "interpretability in the wild":
  - I am unsure about the claim "there exists very simple rule-based algorithm to achieve the same" as it relates to the motivation for the rest of the paper. It is not obvious to me that the circuit needed to implement an algorithm for a fixed depth CFG is much more complex than the indirect object identification circuit?
  - The footnote makes a very strange statement "Yet, they also said 'to the best of our knowledge, (this is) the most detailed attempt at reverse-engineering a natural end-to-end behavior in a transformer-based language model.' Our paper appeared online six months after Wang et al. (2022)." Why is this relevant? The current paper tackles a totally different setting than "interpretability in the wild", and does not reverse engineer a _natural_ end-to-end behavior (in the sense of being "in the wild") because it focuses entirely on toy tasks.

## Fix other smaller issues
- Please fix inconsistency about how CFG difficulty scales with the number of NT/T symbols.
- The paper occasionally makes statements like "A coding grammar (like python) can be parsed using greedy without ambiguity." What does it mean to "use greedy"? I assume you mean "using a greedy algorithm"? Please fix this to be explicit.
- What do you mean by "that may not be information-theoretically possible" in Figure 6?

---

> ### Author Response · Authors · 2025-10-09
> **Response to Reviewer CZQJ**
>
> We sincerely thank the reviewer for the thoughtful, detailed, and constructive feedback.
> We are very grateful that the reviewer finds most of the claims correct, the limitations clearly stated, and the overall work potentially interesting to some (certainly not all) readers in the interpretability and algorithmic reasoning community.
> Below we respond to each of the reviewer’s major points and describe the corresponding revisions (highlighted in blue in the updated manuscript).
>
> ### 1. Multi-head linear probing layer
>
> We strongly confirm that the probing function is linear in the model’s hidden states and **not trained per input x**.
> The probe is trained once and then **evaluated on entirely unseen CFG samples**.
> This clarification—previously in the appendix—is now moved to Section 4.1 and explicitly emphasized in the text (see blue edits).
>
> When the sequence length is fixed (but $x$ varies), the probing function $G_i(x)$ is a totally fixed linear function over the hidden-states, regardless of $x$ being seen or unseen during training.
> We **now also expanded the explanation in words** to describe intuitively what the probing function $G_i(x)$ does and why it is linear.
> We hope this makes its operation clearer and addresses the reviewer’s concern about generalization versus per-instance fitting.
>
> ### 2. Complexity of the synthetic CFGs
>
> We thank the reviewer for catching this typo.
> cfg3e1 and cfg3e2 have more (not fewer) NT/T symbols, which makes them less ambiguous and therefore easier to learn.
> We have corrected this typo in Figure 4's caption and also highlighted the corrected statement in Section 2 (in blue text).
> We deeply appreciate the reviewer’s careful reading in spotting this error.
>
> ### 3. Attention diagrams and Section 5 interpretation
>
> We fully agree that the writing around Results 7–9 and Figure 8 was previously unclear. We kind of stated the results in Section 5.2 but deferring all the explanations to Section 5.3. That was a bad choice.
>
> In this revision, we have **substantially revised Section 5.2** (highlighted in blue) to better explain the purpose and hypotheses of these experiments: what we expect to observe if the Transformer indeed learns DP-like reasoning, and why such attention behavior is meaningful. Specifically:
>
> * We have **followed the reviewer’s suggestion** to describe “what we would expect to see (and why)” if the hypothesis were true.
>
> * We have **improved the explanation of Figure 8c (Result 9)**, adding intuitive descriptions of how to read the figure and why it is important.
>
> * We **added Algorithm 1** (a pseudocode DP example) to show exactly how these attention results correspond to DP-like information flow, as illustrated in Figure 9.
>
> *Regarding the suggestion to ground the results in an actual sequence:*
> our sequences are often hundreds of tokens long and difficult to visualize in full, but Figure 9 now presents a truncated real sequence with explicit correspondence to the DP pseudocode, serving the same purpose in a more readable form.
>
> ### 4. Positioning relative to prior work
>
> We thank the reviewer for this careful feedback and have revised the writing accordingly.
>
> Our use of the word “hypothesized” for Olsson et al. (2022) was intentional but we recognize it needed clarification. They paper focus primarily on token-level induction heads (e.g., pattern matching, copying) and said
>
> > "What about the induction heads we saw in Argument 2 with more complex behavior? Can we also reverse engineer them? Do they operate on the same mechanisms? Presently, fully reverse engineering them is beyond us, since they exist in large models with MLPs, which we don't have a strong framework for mechanistically understanding."
>
> We did not solve this either. We have now clarified our work is complementary: we trade naturalistic settings for controlled synthetic grammars that allow us to interpret deeper, hierarchical algorithmic structures.
> We have clarified this distinction in the conclusion, where we explicitly state that our study does not claim to mechanistically reverse-engineer real-world models, but instead aims to provide a controlled environment for algorithmic interpretability.
>
> Additionally, following the reviewer’s advice, we have removed the sentence “Our paper appeared online six months after Wang et al. (2022)”
>
>
> ### 5. Other smaller issues
>
> We have fixed all minor issues mentioned:
>
> * Changed phrasing to “using a greedy algorithm.”
>
> * Clarified the meaning of “information-theoretically possible” in Figure 6.
>
> * Corrected the inconsistencies about CFG difficulty scaling.
>
> ## Closing remark
>
> We thank the reviewer again for the insightful comments and constructive suggestions, which have greatly improved the clarity and presentation of the paper.
> We hope the revisions address the reviewer’s concerns and make the manuscript clearer and more accessible to readers interested in hierarchical reasoning and interpretability. Have a good rest of your day.

---

> > ### Comment · Reviewer_CZQJ · 2025-10-28
> >
> > Thank you for the revision, and for clarifying the training of the multi-head linear probing layer. The explanation of section 5 is also much clearer in the revised version of the paper, and I believe my main concerns have been addressed.

---

### Decision · Action_Editor_R4g6 · 2025-11-08

**Recommendation:** Accept as is

**Additional Comments:**

My main recommendation for the author for the final version is to think how to advance the clarify of the paper, which is already quite dense. If the goal is for people to read and enjoy I think pushing on that front thinking how to simplify wherever possible and improving visualizations as much as possible will aid in this goal -- I don't have concrete proposals but it's clear the paper is not easy to digest.

**Audience:**

Yes

**Audience Explanation:**

Both understanding the practical expressivity of transformers and the mechanistic way by which they implement computations such as CFG parsing that require understanding hierarchical structure and resolving local ambiguities with global structure can be relevant to members of the TMLR audience

**Claims And Evidence:**

Yes

**Claims Explanation:**

The authors convincingly show that transformers can learn synthetic CFGs and provide interpretability methods that show that the hidden states are consistent with a DP algorithm that correctly identifies constituent boundaries.